# PARAMETER AND MEMORY EFFICIENT PRETRAINING VIA LOW-RANK RIEMANNIAN OPTIMIZATION

**Zhanfeng Mo**[†]   **Long-Kai Huang** [§]   **Sinno Jialin Pan**[†‡]

[†] Nanyang Technological University, Singapore  [§] Tencent AI Lab

[‡] The Chinese University of Hong Kong

zhanfeng001@ntu.edu.sg;    hlongkai@gmail.com;    sinnopan@cuhk.edu.hk

## ABSTRACT

Pretraining large language models often requires significant computational resources and memory due to their vast parameter amount. An effective approach to enhance parameter efficiency in both training and inference is to parameterize each full-size weight as the product of two trainable low-rank factors. While low-rank fine-tuning has achieved great success, low-rank pretraining remains challenging as it requires learning extensive knowledge from scratch under the restrictive low-rank parameterization. During standard low-rank pretraining, separately optimizing the low-rank factors introduces redundant information from the full gradient, which hinders the learning process. To achieve efficient yet effective low-rank pretraining, we propose a **Lo**w-rank **R**iemannian **O**ptimizer (**LORO**). At each LORO update step, the low-rank factor pairs are jointly updated to ensure their full-size product moves along the steepest descent direction on the low-rank manifold, without the need to compute any memory-intensive full-size matrices or gradients. Hence, our LORO finds low-rank models that achieve high performance comparable to full-size pretrained models, while significantly reducing memory usage and accelerating both training and inference. A LLaMA 1B model pretrained with LORO achieves a perplexity score of $2\%$ better than the full-size baseline, with a $54\%$ reduction in model memory, a $\times 1.8$ speedup in training, and a $\times 2.2$ speedup in inference. The code is available on GitHub[1].

## 1 INTRODUCTION

In recent years, large language models (LLMs) (Brown et al., 2020; Touvron et al., 2023a;b; Team et al., 2024a;b), whose parameter sizes often reach hundreds of billions, have obtained remarkable performance across a wide range of applications. Typically, a pretraining and fine-tuning paradigm is adopted for building powerful LLMs from scratch: a model is first pretrained on large-scale unsupervised corpora (Devlin et al., 2019; Brown et al., 2020) to acquire general semantics and extensive knowledge, and then fine-tuned on a smaller downstream dataset to enhance its domain-specific capabilities (Hu et al., 2022; Team et al., 2024b; Ramesh et al., 2024).

An extensive amount of trainable parameters results in a substantial memory footprint and high computational costs during pretraining and fine-tuning. To alleviate these issues, various memory-efficient training techniques have been proposed to reduce memory usage and accelerate training. These approaches stem from either an engineering perspective, such as gradient checkpointing (Chen et al., 2016), memory offloading (Rajbhandari et al., 2020), or an optimization perspective, including low-rank optimizers (Hu et al., 2022; Zhang et al., 2023; Hayou et al., 2024; Lialin et al., 2024) and low-precision optimizers (Dettmers et al., 2021; Li et al., 2023; Zhang et al., 2024b). Since low-rank adaptation (LoRA) (Hu et al., 2022) has gained increasing attention, becoming one of the most popular parameter-efficient fine-tuning methods. At each layer, LoRA replaces the full-size pretrained weight $\mathbf{W} \in \mathbb{R}^{d \times d}$ with $\mathbf{W} + \mathbf{BA}$, where $\mathbf{B} \in \mathbb{R}^{d \times r}$ and $\mathbf{A} \in \mathbb{R}^{r \times d}$ are two trainable low-rank factors, while $\mathbf{W}$ is frozen during fine-tuning to reduce training memory footprint. However, these methods depend on full-size pretrained weights, making them inapplicable to

---

[1]https://github.com/mzf666/LORO-main

pretraining scenarios. Furthermore, they ultimately produce models with full-size weights, which do not result in any memory reduction or speedup in the inference phase.

In this work, we aim to enhance efficiency during both the training and inference by exploring the pretraining of low-rank parameterized language models, where each weight matrix is parameterized as $\mathbf{W} \triangleq \mathbf{BA}$, the product of two rank-$r$ matrices. In practice, such low-rank parameterization enjoys great potential in improving both training and inference efficiency, as it reduces the parameter size at each layer from $d^2$ to $2rd$ and decreases the FLOPS from $O(d^3)$ to $O(rd^2)$. Theoretically, a low-rank model can achieve a $\times(2r/d)$ memory reduction and a $\times(d/2r)$ inference speedup.

However, pre-training low-rank language models from scratch using standard stochastic gradient descent methods is much more challenging. As shown in (Kamalakara et al., 2022; Zhao et al., 2024), separately optimizing each low-rank factor of the model may result in a significant performance drop compared to full-size models. Therefore, as a compromise, recent works consider pretraining language models with full-size weights while utilizing low-rank or sparse updates to enhance training efficiency: ReLoRA (Lialin et al., 2024) periodically merges a low-rank update into the full-size weight after a full-size warm-up training phase; GaLore (Zhao et al., 2024) updates the full-size model weights using low-rank yet full-size gradients; SLTrain (Han et al., 2024) proposes sparse plus low-rank pretraining, where the full-size weights are decomposed into sum of sparse and low-rank matrices. However, these methods can merely produce full-size models, leading to limited or no memory reduction or acceleration during inference.

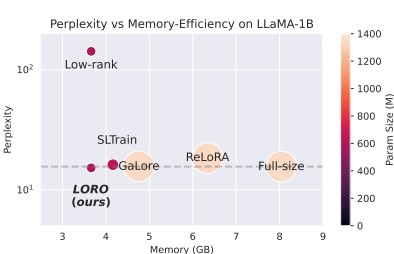

Figure 1: Visualization of the perplexity (log scaled) and memory-efficiency trade-off of pretraining LLaMA-1B on C4 (Table 1). Methods closer to the bottom-left corner and associated with smaller bubble sizes exhibit superior performance and memory efficiency.

While recent studies (Zhao et al., 2024; Han et al., 2024) observe that the rank of the resulting weight matrix $\mathbf{W}$ obtained from full-size pretraining methods is typically high (close to $d$), this does not necessarily mean that low-rank pretraining approaches (learning $\mathbf{BA}$ instead of $\mathbf{W}$) are incapable of achieving satisfactory performance in learning foundation models. We argue that with careful design of the learning process for the low-rank factors $\mathbf{A}$ and $\mathbf{B}$, it is possible to train a model with performance comparable to full-size models. Therefore, this paper aims to address the research question: *Can we train a low-rank parameterized language model from scratch to achieve performance comparable to the full-size baseline?* In essence, low-rank pretraining is more challenging than fine-tuning. Given the pretrained full-size weights, fine-tuning only requires capturing the minor domain-specific knowledge through a low-rank adaptation. In contrast, learning extensive pretraining knowledge from scratch under the restrictive low-rank parameterization requires a more careful optimization process. As noted in the discussions of (3) in Section 3, optimizing the low-rank factors separately introduces redundant information from the full gradient, which hinders the feature learning process. In this scenario, the separate optimization of $\mathbf{B}$ and $\mathbf{A}$ fails to direct their product $\mathbf{BA}$ along the steepest descent direction of the loss on the rank-$r$ matrix manifold. This necessitates considering the intricate geometry of the manifold, prompting us to optimize the low-rank factors jointly through Riemannian optimization. While recent works (Savostianova et al., 2023; Schotthöfer et al., 2022) have attempted to train low-rank vision classifiers using Riemannian gradient methods, they either require evaluating full-size gradients or rely on SVD-like parameterization, which introduces $O(r^2)$ additional parameters, making them less preferable for pretraining language models. To the best of our knowledge, this paper is the first to explore the pretraining of low-rank language models through the lens of Riemannian optimization.

**Contributions.** We propose the **Lo**w-rank **R**iemannian **O**ptimizer (**LORO**) for effective low-rank pretraining. At each LORO update step, each pair of low-rank factors is jointly updated to ensure their product is transported along the Riemannian gradient of the rank-$r$ matrix manifold to pursue effective descent in the model loss. All LORO's updates are performed directly within the low-rank parameterization, without the need to evaluate the costly full-size weights or gradients. To our knowledge, we are the first to show that pretraining low-rank language models with Riemannian gradients achieve perplexity scores comparable to full-size pretraining, based on extensive experiments conducted on LLaMA models ranging from 60M to 1B in size. On LLaMA-1B, LORO achieves a

perplexity score of $2\%$ better than the full-size baseline, with $54\%$ less model memory cost and offers a $\times 1.8$ speedup in training and a $\times 2.2$ speedup in inference. Within the same parameter budget, we further show that LORO outperforms its full-size counterparts, highlighting LORO's capacity to unblock the potential of low-rank parameterization for enhanced representation learning.

## 2 RELATED WORKS

**Low-rank fine-tuning.** Pioneered by LoRA (Hu et al., 2022), which achieves parameter-efficient fine-tuning by restricting optimization exclusively to the low-rank factors $\mathbf{BA}$, various low-rank fine-tuning methods have been proposed to enhance downstream adaptation (Zhang et al., 2023; Ding et al., 2023; Hayou et al., 2024; Zhang & Pilanci, 2024; Xia et al., 2024), improve cross-task generalization (Agiza et al., 2024; Huang et al., 2023; Buehler & Buehler, 2024), and increase fine-tuning efficiency (Chen et al., 2023; Dettmers et al., 2023; Gamal & Rabusseau, 2023; Koohpayegani et al., 2024; Valipour et al., 2023). However, all these low-rank fine-tuning methods require access to the full-size pretrained weights so the model can be adapted within a low-rank parameter space. When directly applied to pretraining, the absence of pretrained full-size weights—where most knowledge resides—often results in significant performance drops.

**Low-rank pretraining.** While a few works have attempted to train low-rank vision models (Sui et al., 2023; Khodak et al., 2021; Savostianova et al., 2023), training low-rank parameterized language models from scratch has been observed to be challenging and remains an open problem (Kamalakara et al., 2022; Zhao et al., 2024). Existing methods circumvent these challenges by employing low-rank or sparse updates while still using full-size weights, such as updating full-size weights with low-rank gradients (Zhao et al., 2024; Zhang et al., 2024b), periodically merging low-rank factors into full-size weights (Lialin et al., 2024), or using a low-rank plus sparse weight parameterization (Han et al., 2024). As a result, these methods provide little to no improvement in inference efficiency. In contrast, LORO is the first to directly tackle language model pretraining using low-rank parameterized weights.

**Optimization on low-rank matrix manifolds.** Abundant theories and algorithms (Absil & Oseledets, 2015; Absil et al., 2009; Mishra et al., 2014) have been proposed to perform Riemannian optimization on low-rank matrix manifolds for traditional applications, such as matrix completion (Vandereycken, 2013; Bian et al., 2023) and regressions (Bleakley & Yamanishi, 2009; Yuan et al., 2007; Amit et al., 2007). Recent works (Savostianova et al., 2023; Schotthöfer et al., 2022) aim to train low-rank vision classifiers using Riemannian gradient flow (Koch & Lubich, 2007). However, these methods either require evaluating full-size gradient and weight matrices or rely on SVD-like parameterization, which introduces parameter overhead, making them incompatible with low-rank parameterized language models. In contrast, LORO operates under low-rank parameterization, avoiding full-size matrix evaluations and enabling efficient pretraining on language models.

## 3 PRELIMINARIES

**Low-rank pretraining.** The rank-$r$ parameterization of a $d$-by-$d$ matrix is defined as $\mathbf{W} = \mathbf{BA}$, where $\mathbf{B} \in \mathbb{R}^{d \times r}$ and $\mathbf{A} \in \mathbb{R}^{r \times d}$ are the associated rank-$r$ factors. When $r < d$, $\mathbf{BA}$ is called the low-rank parameterization of $\mathbf{W}$. In this work, we focus on training the decoder-only transformer-based language models (Vaswani et al., 2017; Touvron et al., 2023a) from scratch using low-rank parameterization. Recall that the two fundamental building blocks of the transformer model are the linear projection $f_{\text{proj}}(\cdot)$ and the self-attention function $f_{\text{attn}}(\cdot)$, which are defined as

$$f_{\text{proj}}(\mathbf{X}) \triangleq \mathbf{X}\mathbf{W}_p, \ f_{\text{attn}}(\mathbf{X}) \triangleq \text{softmax}\left(\mathbf{X}\mathbf{W}_q\mathbf{W}_k^\top\mathbf{X}^\top/\sqrt{d}\right)\mathbf{X}\mathbf{W}_v, \tag{1}$$

where $\mathbf{X} \in \mathbb{R}^{n \times d}$ is the hidden feature, $n$ is the token number, $d$ is the hidden dimension, and $\mathbf{W}_p$, $\mathbf{W}_q$, $\mathbf{W}_k$, $\mathbf{W}_v \in \mathbb{R}^{d \times d}$ denote the trainable projection, query, key, and value matrices, respectively. For notation simplicity, we assume that $d$ remains constant across all layers, though all derivations in this work also apply to scenarios with varying dimensions. As the composition of multiple layers of linear projections, self-attention functions, and various operations (e.g., activation, normalization, and positional embedding), the target language model can be parameterized by the stack of trainable matrices across all $N$ layers, denoted as $(\mathbf{W}_1, \cdots, \mathbf{W}_N)$. Therefore, the rank-$r$ parameterized

language model is implemented by parameterizing each trainable matrix as a rank-$r$ factorization $\mathbf{W}_i = \mathbf{B}_i \mathbf{A}_i$ with $\mathbf{B}_i \in \mathbb{R}^{d \times r}$ and $\mathbf{A}_i \in \mathbb{R}^{r \times d}$, and is represented by $(\mathbf{B}_1 \mathbf{A}_1, \ldots, \mathbf{B}_N \mathbf{A}_N)$. In practice, we set $r < d$ for parameter efficiency. During pretraining, the rank-$r$ model is trained from scratch on unsupervised corpus data for the next token prediction, defined as:

$$\min L(\mathbf{W}_1, \cdots, \mathbf{W}_N), \text{ s.t. } \mathbf{W}_i = \mathbf{B}_i \mathbf{A}_i, \mathbf{B}_i \in \mathbb{R}^{d \times r}, \mathbf{A}_i \in \mathbb{R}^{r \times d}, i = 1, ..., N, \quad (2)$$

where $L(\cdot) : \mathbb{R}^{d \times d} \times \cdots \times \mathbb{R}^{d \times d} \mapsto \mathbb{R}^+$ denotes the smooth next token prediction loss (i.e., the logarithm of perplexity score (Radford & Narasimhan, 2018)). At the $t$-th training step with learning rate $\eta > 0$, the full-size weight is updated by the full gradient $\mathbf{W}_i^{t+1} \leftarrow \eta \nabla_{\mathbf{W}_i^t} L$. In contrast, during the standard low-rank pretraining, the low-rank factors are updated separately, which leads to

$$\mathbf{W}_i^{t+1} \leftarrow (\mathbf{B}_i^t - \eta \nabla_{\mathbf{B}_i^t} L)(\mathbf{A}_i^t - \eta \nabla_{\mathbf{A}_i^t} L) \approx \mathbf{W}_i^t - \eta \left( \mathbf{B}_i^t \mathbf{B}_i^{t\top} \cdot \nabla_{\mathbf{W}_i^t} L + \nabla_{\mathbf{W}_i^t} L \cdot \mathbf{A}_i^{t\top} \mathbf{A}_i^t \right) \quad (3)$$

where the last line is derived from the chain-rule $\mathbf{B}_i^{t\top} \nabla_{\mathbf{W}_i^t} L = \nabla_{\mathbf{A}_i^t} L$ and $\nabla_{\mathbf{W}_i^t} L \cdot \mathbf{A}_i^{t\top} = \nabla_{\mathbf{B}_i^t} L$. The update direction in (3) is the sum of two sketched versions of the full gradient, each projected onto the low-rank subspaces spanned by the columns of $\mathbf{B}_i^t$ and the rows of $\mathbf{A}_i^t$. When these two subspaces overlap, the sketched gradient introduces redundant information from the full gradient, hindering the learning of effective representations. From a geometric perspective, separately optimizing the low-rank factors overlooks the intricate structure of the low-rank parameterization and fails to guide their product toward the steepest loss descent on the low-rank matrix manifold [2]. To address this issue, we recast (2) into a Riemannian optimization problem:

$$\min_{\mathbf{B}_i \in \mathbb{R}^{d \times r}, \ \mathbf{A}_i \in \mathbb{R}^{r \times d}} L(\mathbf{B}_1 \mathbf{A}_1, \cdots, \mathbf{B}_N \mathbf{A}_N) \implies \min_{(\mathbf{B}_i \mathbf{A}_i) \in \mathcal{M}_r} L(\mathbf{B}_1 \mathbf{A}_1, \cdots, \mathbf{B}_N \mathbf{A}_N), \quad (4)$$

where $\mathcal{M}_r$ denotes the manifold of all $d$-by-$d$ rank-$r$ matrices. In essence, this reformulation accounts for the geometric structure of $\mathcal{M}_r$ by **treating the product $(\mathbf{B}_i \mathbf{A}_i)$ as a whole**: it emphasizes that the principle for updating $\mathbf{B}_i$ and $\mathbf{A}_i$ is to guide the product $(\mathbf{B}_i \mathbf{A}_i)$ towards the steepest descent direction within the manifold, rather than optimizing them separately in the ambient space.

**Optimization on low-rank matrix manifold.** We briefly review the retraction-based Riemannian optimization on manifolds and refer interested readers to (Lee, 2019; Absil & Oseledets, 2015) for rigorous definitions and a more comprehensive review. Imagine an ant on the Earth's surface seeking the highest point. Although it wants to climb straight up along the steepest path—an infeasible option that would send it into space—it must navigate the curved surface, choosing the best direction from the feasible tangents at its current location to remain on the Earth. This thought experiment helps illustrate the difference between optimization in the flat full-size matrix space (e.g., the ambient space) and on the curved rank-$r$ matrix manifold $\mathcal{M}_r$ (e.g., the Earth).

As visualized in Figure 2, when optimizing in the ambient space, we evaluate the negative Euclidean gradient (blue vector) at each step by backpropagation, then update the current weight by moving it a small step towards the blue vector. However, this update is infeasible on $\mathcal{M}_r$, as it moves the current weight out of the manifold. Instead, we aim to update along the black dashed curve on the manifold, which guides us toward smaller loss values. To this end, we need to identify the Riemannian gradient (red solid vector) within the tangent space (gray plane) that aligns most closely with the Euclidean gradient. This can be done by projecting the blue vector onto the tangent space. We then update the weight for a small step along the red solid vector and retract it back to the manifold (via the red dashed vector) to avoid violating the rank constraint. This process is known as the retraction-based Riemannian optimization step, involving: 1. Updating with the projected gradient and 2. Retracting back to the manifold.

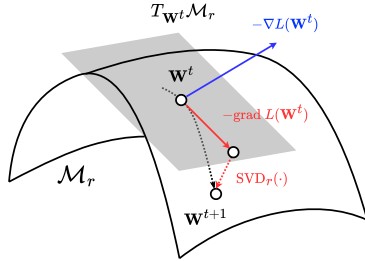

Figure 2: Retraction-based Riemannian optimization, where the grey plane is the tangent space, $\nabla L(\mathbf{W}^t)$ and $\text{grad } L(\mathbf{W}^t)$ denote the Euclidean and Riemannian gradient at $\mathbf{W}^t$, respectively.

Thanks to the embedded manifold theories, Proposition 1 (see (Lee, 2019, Example 8.14) and (Absil et al., 2009, Equation 3.37)) shows how to compute the Riemannian gradient via orthogonal projection onto the tangent space, while Proposition 2 (see (Lewis & Malick, 2008, Lemma 2.1)) shows that retraction to $\mathcal{M}_r$ can be done by rank-$r$ singular value decomposition (SVD).

---

[2]See Appendix C for more detailed discussions on the issues in standard low-rank pretraining.

**Proposition 1** (Tangent Space and Riemannian Gradient on $\mathcal{M}_r$). *$\mathcal{M}_r$ is a smooth submanifold embedded in the ambient space $\mathbb{R}^{d \times d}$. For any base point $\mathbf{W} \in \mathcal{M}_r$ with singular value decomposition $\mathbf{W} = \mathbf{U\Sigma V}^\top$ with $\mathbf{U}, \mathbf{V} \in \mathbb{R}^{d \times r}$ being orthogonal, the tangent space at $\mathbf{W}$ is given by*

$$T_{\mathbf{W}}\mathcal{M}_r \triangleq \left\{ \mathbf{UZV}^\top + \mathbf{U}'\mathbf{V}^\top + \mathbf{UV}'^\top : \mathbf{Z} \in \mathbb{R}^{r \times r}, \mathbf{U}', \mathbf{V}' \in \mathbb{R}^{d \times r}, \mathbf{U}^\top \mathbf{U}' = \mathbf{V}^\top \mathbf{V}' = \mathbf{0} \right\}, \quad (5)$$

*and the orthogonal projection to $T_{\mathbf{W}}\mathcal{M}_r$ is given by $\mathcal{P}(\mathbf{G}) = \mathbf{UU}^\top \mathbf{G} + \mathbf{GVV}^\top - \mathbf{UU}^\top \mathbf{GVV}^\top$. When equipped with the matrix inner-product metric, $\mathcal{M}_r$ becomes a Riemannian manifold. Then, the Riemannian gradient of any smooth function $L(\cdot) : \mathbb{R}^{d \times d} \mapsto \mathbb{R}$ evaluated at $\mathbf{W}$, denoted as $\mathrm{grad}\, L(\mathbf{W})$, can be calculated by projecting the Euclidean gradient $\nabla L(\mathbf{W})$ to $T_{\mathbf{W}}\mathcal{M}_r$:*

$$\mathrm{grad}\, L(\mathbf{W}) = \mathcal{P}(\nabla L(\mathbf{W})) = \mathbf{UU}^\top \nabla L(\mathbf{W}) + \nabla L(\mathbf{W})\mathbf{VV}^\top - \mathbf{UU}^\top \nabla L(\mathbf{W})\mathbf{VV}^\top. \quad (6)$$

**Proposition 2** (Retraction to $\mathcal{M}_r$ via SVD). *Informally, for any base point $\mathbf{W} \in \mathcal{M}_r$ and tangent vector $\mathbf{G} \in T_{\mathbf{W}}\mathcal{M}_r$, a retraction is a function $R(\cdot)$ that maps the shifted base point $(\mathbf{W} + \mathbf{G})$ back to $\mathcal{M}_r$. A feasible retraction on $\mathcal{M}_r$ is $R(\mathbf{W} + \mathbf{G}) \triangleq \mathrm{SVD}_r(\mathbf{W} + \mathbf{G})$, where $\mathrm{SVD}_r(\cdot)$ is the rank-$r$ singular value decomposition, projecting the updated base point from $\mathbb{R}^{d \times d}$ back to $\mathcal{M}_r$.*

## 4 METHODOLOGY

### 4.1 RIEMANNIAN UPDATE UNDER LOW-RANK PARAMETERIZATION

Proposition 1 and Proposition 2 enable us to perform Riemannian low-rank pretraining on $\mathcal{M}_r$ via

$$\mathbf{W}_i^{t+1} \leftarrow R(\mathbf{W}_i^t - \eta\, \mathrm{grad}\, L(\mathbf{W}_i^t)) = \mathrm{SVD}_r\left(\mathbf{W}_i^t - \eta\, \mathcal{P}_i\left(\nabla_{\mathbf{W}_i^t} L\right)\right), \quad (7)$$

where $\mathcal{P}_i(\cdot)$ denotes the orthogonal projection to the tangent space $T_{\mathbf{W}_i}\mathcal{M}_r$ as defined in Proposition 1. For each trainable weight, we: 1. evaluate the Euclidean gradient $\nabla_{\mathbf{W}_i} L$, and derive the tangent space projection $P_i(\cdot)$ from the SVD of base point $\mathbf{W}_i = \mathbf{U}_i \mathbf{\Sigma}_i \mathbf{V}_i^\top$; 2. find the Riemannian gradient $\mathcal{P}_i(\nabla_{\mathbf{W}_i} L)$ by projecting $\nabla_{\mathbf{W}_i} L$ to the tangent space; 3. transport the base point along the negative Riemannian gradient; 4. retract the updated base point back to $\mathcal{M}_r$ via the $\mathrm{SVD}_r(\cdot)$ projection.

However, these update steps encounter several practical issues that render them incompatible with low-rank parameterization: First, the full-size gradient $\nabla_{\mathbf{W}_i} L$ is inaccessible during backpropagation (Rumelhart et al., 1988); we can only obtain $\nabla_{\mathbf{B}_i} L$ and $\nabla_{\mathbf{A}_i} L$, the gradients of the low-rank factors. Second, in steps 1, 2, and 4, we need to manipulate the memory-intensive full-size matrix $\mathbf{W}_i$, either by performing SVD or matrix multiplication, which compromises memory efficiency.

This motivates us to revise the update steps of (7) into a "**low-rank parameterization friendly**" form by avoiding any computations with full-size $d$-by-$d$ matrices (e.g., the weight $\mathbf{W}_i$ and gradient $\nabla_{\mathbf{W}_i} L$) and ensuring all operations can be performed under the low-rank parameterization. To achieve this objective, our focus is on the following two goals: 1. Eliminating the need for the full-size gradient when calculating the Riemannian gradient through tangent space projection; 2. Keeping the Riemannian update in its factorized form to avoid the necessity of performing SVD on full-size matrices. To achieve the 1st goal, we propose Proposition 3, which shows that the Riemannian gradient on $\mathcal{M}_r$ can be computed from $\nabla_{\mathbf{B}_i} L$ and $\nabla_{\mathbf{A}_i} L$.

**Proposition 3** (Riemannian Gradient under Low-rank Parameterization). *Following the previous notation, for any low-rank parameterized weight $\mathbf{W} = \mathbf{BA} \in \mathcal{M}_r$, where $\mathbf{B} \in \mathbb{R}^{d \times r}$ and $\mathbf{A} \in \mathbb{R}^{r \times d}$, the Riemannian gradient of any smooth function $L(\cdot) : \mathbb{R}^{d \times d} \mapsto \mathbb{R}$ evaluated at $\mathbf{W}$ is*

$$\mathrm{grad}\, L(\mathbf{W}) = \mathbf{B}(\mathbf{B}^\top \mathbf{B})^{-1} \cdot \nabla_{\mathbf{A}} L + \nabla_{\mathbf{B}} L \cdot (\mathbf{AA}^\top)^{-1} \mathbf{A} - \mathbf{B}(\mathbf{B}^\top \mathbf{B})^{-1} \cdot \nabla_{\mathbf{A}} L \cdot \mathbf{A}^\top (\mathbf{AA}^\top)^{-1} \mathbf{A}. \quad (8)$$

The proof of Proposition 3 can be obtained from Proposition 1 based on two key observations: 1. Let $\mathbf{U\Sigma V}^\top$ be the SVD of $\mathbf{BA}$, the column space of $\mathbf{B}$ coincides with that of $\mathbf{U}$, and the row space of $\mathbf{A}$ coincides with the column space of $\mathbf{V}$. 2. Backpropagation gives $\nabla_{\mathbf{W}} L \cdot \mathbf{A}^\top = \nabla_{\mathbf{B}} L$ and $\mathbf{B} \cdot \nabla_{\mathbf{W}} L = \nabla_{\mathbf{A}} L$. The proof details are postponed to the Appendix B. In practice, for each weight matrix $\mathbf{W}_i$, (8) involves only multiplications of $r$-by-$d$ and $r$-by-$r$ matrices, as well as the inversion of $r$-by-$r$ matrices, eliminating the need to compute any expensive $d$-by-$d$ matrices or use the $d$-by-$d$ gradient $\nabla_{\mathbf{W}_i} L$. Additionally, all terms in (8) can be efficiently computed from the readily available variables—$\mathbf{B}_i$, $\mathbf{A}_i$, $\nabla_{\mathbf{B}_i} L$, and $\nabla_{\mathbf{A}_i} L$—obtained during backpropagation.

Although Proposition 3 provides a memory-efficient approach for calculating the Riemannian gradient, the resulting $\mathrm{grad}\, L(\mathbf{W})$ remains a full-size $d$-by-$d$ matrix, which compromises the feasibility of the subsequent Riemannian update and retraction. To address this issue and achieve the 2nd goal, we propose Proposition 4, which shows that the successive update steps can be achieved within a factorized form, avoiding the need to compute any full-size weight or gradient.

**Proposition 4** (Riemannian Update under Low-rank Parameterization). *Following the previous notations, for any low-rank parameterized weight* $\mathbf{W} = \mathbf{BA} \in \mathcal{M}_r$, *let* $\mathbf{B} = \mathbf{Q}_B \mathbf{R}_B$ *and* $\mathbf{A}^\top = \mathbf{Q}_A \mathbf{R}_A$ *be the QR-decompositions of* $\mathbf{B}$ *and* $\mathbf{A}^\top$. *Then the Riemannian updated weight* $\mathbf{W}' \triangleq \mathrm{SVD}_r(\mathbf{W} - \eta\, \mathrm{grad}\, L(\mathbf{W}))$ *admits a low-rank factorization* $\mathbf{W}' = \mathbf{B}'\mathbf{A}'$, *with* $\mathbf{B}' = (\mathbf{Q}_B, \overline{\mathbf{Q}}_B)\, \mathbf{U}_* \boldsymbol{\Sigma}_*^{1/2}$ *and* $\mathbf{A}' = \boldsymbol{\Sigma}_*^{1/2}\mathbf{V}_*^\top (\mathbf{Q}_A, \overline{\mathbf{Q}}_A)^\top$. *Specifically,* $\mathbf{U}_*\boldsymbol{\Sigma}_*\mathbf{V}_*^\top$ *is determined by the rank-$r$ SVD of the following $2r$-by-$2r$ matrix, i.e.,*

$$\mathbf{U}_*\boldsymbol{\Sigma}_*\mathbf{V}_*^\top = \mathrm{SVD}_r \begin{pmatrix} \mathbf{S} & -\eta\,\overline{\mathbf{R}}_A^\top \\ -\eta\,\overline{\mathbf{R}}_B & \mathbf{0} \end{pmatrix} \in \mathbb{R}^{2r \times 2r}, \quad \mathbf{S} \triangleq \mathbf{R}_B \mathbf{R}_A^\top - \eta\, \mathbf{R}_B^{-\top}\nabla_\mathbf{A} L \cdot \mathbf{Q}_A \in \mathbb{R}^{r \times r}, \quad (9)$$

*and* $\overline{\mathbf{Q}}_B\overline{\mathbf{R}}_B$ *is the QR decomposition of the $d$-by-$r$ matrix* $(\nabla_\mathbf{B} L \cdot \mathbf{R}_A^{-1} - \mathbf{Q}_B\mathbf{Q}_B^\top \nabla_\mathbf{B} L \cdot \mathbf{R}_A^{-1})$, *while* $\overline{\mathbf{Q}}_A\overline{\mathbf{R}}_A$ *is the QR decomposition of the $d$-by-$r$ matrix* $(\nabla_\mathbf{A} L^\top \mathbf{R}_B^{-1} - \mathbf{Q}_A\mathbf{Q}_A^\top \nabla_\mathbf{A} L^\top \mathbf{R}_B^{-1})$.

The proof of Proposition 4 can be derived from Proposition 3 using the tangent space decomposition (Vandereycken, 2013; Bian et al., 2023) and the proof details are postponed to Appendix B. The key to avoiding SVD on full-size $d$-by-$d$ matrices lies in factorizing the $d$-by-$d$ tangent vector into a product of a $d$-by-$2r$ orthogonal matrix, a $2r$-by-$2r$ full-rank matrix, and a $2r$-by-$d$ orthogonal matrix, enabling SVD to be applied to the smaller $2r$-by-$2r$ matrix instead. Notably, the revised Riemannian update in Proposition 4 only involves QR decomposition of a $d$-by-$r$ matrix, inversion of an $r$-by-$r$ matrix, SVD on a $2r$-by-$2r$ matrix, and multiplication of $d$-by-$2r$ matrices.

## 4.2 Low-rank Riemannian Optimizer

In Proposition 4, we have derived the "low-rank parameterization friendly" Riemannian update:

$$\mathbf{B}_i^{t+1} \leftarrow (\mathbf{Q}_B, \overline{\mathbf{Q}}_B)\mathbf{U}_*\boldsymbol{\Sigma}_*^{1/2}, \quad \mathbf{A}_i^{t+1} \leftarrow \boldsymbol{\Sigma}_*^{1/2}\mathbf{V}_*^\top(\mathbf{Q}_A, \overline{\mathbf{Q}}_A)^\top, \quad (10)$$

where $\mathbf{Q}_B$, $\overline{\mathbf{Q}}_B$, $\mathbf{Q}_A$, $\overline{\mathbf{Q}}_A$, $\mathbf{U}_*$, $\boldsymbol{\Sigma}_*$, and $\mathbf{V}_*$ can be directly computed from $\mathbf{B}_i^t$, $\mathbf{A}_i^t$, $\nabla_{\mathbf{B}_i^t} L$, and $\nabla_{\mathbf{A}_i^t} L$, without evaluating any full-size weight or gradient. Although memory-efficient, the QR decomposition, SVD, and matrix inversion involved in (10) impede the computational efficiency of low-rank pretraining. This motivates us to develop a rough but efficient approximation of the exact Riemannian update, which enables us to amortize the computational cost across multiple steps by periodically applying the exact update while primarily using the approximated update.

To this end, a simple approach is to skip the costly Riemannian SVD retraction and directly update the weight along the Riemannian gradient. Unfortunately, the Riemannian gradient in (8) has a maximum rank of $2r$, making it incompatible with the rank-$r$ parameterization. To establish an approximate update that can be applied to $\mathbf{B}_i$ and $\mathbf{A}_i$ respectively, we highlight the differences between the standard low-rank update (3) and Riemannian update (8) in blue:

Standard low-rank update $= -\eta \left( \mathbf{B}_i^t \cdot \nabla_{\mathbf{A}_i^t} L + \nabla_{\mathbf{B}_i^t} L \cdot \mathbf{A}_i^t \right) + O(\eta^2)$,

Riemannian update $= -\eta\big(\mathbf{B}_i^t(\mathbf{B}_i^{t\top}\mathbf{B}_i^t)^{-1}\nabla_{\mathbf{A}_i^t} L \cdot \underbrace{(\mathbf{I} - \mathbf{A}_i^{t\top}(\mathbf{A}_i^t\mathbf{A}_i^{t\top})^{-1}\mathbf{A}_i^t)}_{\text{Orthogonal projection}} + \nabla_{\mathbf{B}_i^t} L \cdot (\mathbf{A}_i^t\mathbf{A}_i^{t\top})^{-1}\mathbf{A}_i^t\big)$.

In the Riemannian update, the gradients are additionally normalized by multiplying them with either $(\mathbf{B}_i^{t\top}\mathbf{B}_i^t)^{-1}$ or $(\mathbf{A}_i^t\mathbf{A}_i^{t\top})^{-1}$. Additionally, $\nabla_{\mathbf{A}_i^t} L$ is projected onto the orthogonal space of the row space of $\mathbf{A}_i^t$ (the middle blue term), thereby removing the redundant gradient components. This motivates us to approximate this normalization by scaling the gradient terms in proportion to the spectral norm of either $(\mathbf{B}_i^{t\top}\mathbf{B}_i^t)^{-1}$ or $(\mathbf{A}_i^t\mathbf{A}_i^{t\top})^{-1}$. The orthogonal projection can be omitted, as it does not affect the gradient scale. Notice that $\left\|(\mathbf{B}_i^{t\top}\mathbf{B}_i^t)^{-1}\right\|_2 = \sigma_1\left((\mathbf{B}_i^{t\top}\mathbf{B}_i^t)^{-1}\right) = \sigma_r\left(\mathbf{B}_i^{t\top}\mathbf{B}_i^t\right)^{-1} = \sigma_r\left(\mathbf{B}_i^t\right)^{-2}$, where $\|\cdot\|_2$ is the spectral norm and $\sigma_k(\cdot)$ denotes the $k$-th largest singular value of a matrix. We estimate $\sigma_r(\mathbf{B}_i^t)^2$ by the average of the squared singular values, i.e., $(1/r)\sum_{k=1}^r \sigma_k(\mathbf{B}_i^t)^2$, which equals $(1/r)\|\mathbf{B}_i^t\|_F^2$, where $\|\cdot\|_F$ denotes the Frobenius norm. Assuming each row of $\mathbf{B}_i^t$ contains no extreme values and its norm is upper-bounded by a constant, we

obtain $\|\mathbf{B}_i^t\|_{\mathrm{F}}^2 = O(d)$. Thus, we approximate the scaling factor as $r/d$ to avoid explicitly computing $\|\mathbf{B}_i^t\|_{\mathrm{F}}^2$. We set the approximated Riemannian update step as

$$\mathbf{B}_i^{t+1} \leftarrow \mathbf{B}_i^t - \eta \cdot (r/d) \cdot \nabla_{\mathbf{B}_i^t} L, \quad \mathbf{A}_i^{t+1} \leftarrow \mathbf{A}_i^t - \eta \cdot (r/d) \cdot \nabla_{\mathbf{A}_i^t} L. \tag{11}$$

We now propose the **Lo**w-rank **R**iemannian **O**ptimizer): for every $K$ training steps, LORO periodically employs an exact low-rank Riemannian update step (equation 10), while in between, it uses the approximated low-rank Riemannian update (equation 11). In practice, LORO accumulates the gradients of $\mathbf{B}_i^t$ and $\mathbf{A}_i^t$ using the momentum strategy as the Adam optimizer (Kingma & Ba, 2015). A pseudo-code of LORO is outlined in Algorithm 1, and a PyTorch code is attached in Algorithm 2. We refer the readers to Appendix C for more detailed discussions on how LORO benefits low-rank pretraining.

---
**Algorithm 1 Low-rank Riemannian Optimizer**

**Input:** current update step $t$, trainable low-rank factors $\mathbf{B}_i^t, \mathbf{A}_i^{t\top} \in \mathbb{R}^{d \times r}$, backward gradients $\nabla_{\mathbf{B}_i^t} L$, $\nabla_{\mathbf{A}_i^t} L$, learning rate $\eta$, exact update frequency $K$.
**Output:** updated low-rank factors $\mathbf{B}_i^{t+1}, \mathbf{A}_i^{t+1}$.
**if** $t \bmod K \neq 0$ **then**
  $\mathbf{B}_i^{t+1} \leftarrow \mathbf{B}_i^t - \eta(r/d)\nabla_{\mathbf{B}_i^t} L$
  $\mathbf{A}_i^{t+1} \leftarrow \mathbf{A}_i^t - \eta(r/d)\nabla_{\mathbf{A}_i^t} L$
**else**
  $(\mathbf{Q}_B, \mathbf{R}_B) \leftarrow \mathrm{QR}(\mathbf{B}_i^t), (\mathbf{Q}_A, \mathbf{R}_A) \leftarrow \mathrm{QR}(\mathbf{A}_i^{t\top})$
  $\mathbf{Z}_B \leftarrow \nabla_{\mathbf{B}_i^t} L \cdot \mathbf{R}_A^{-1}, \mathbf{Z}_A \leftarrow \nabla_{\mathbf{A}_i^t} L^\top \mathbf{R}_B^{-1}$
  $(\overline{\mathbf{Q}}_B, \overline{\mathbf{R}}_B) \leftarrow \mathrm{QR}(\mathbf{Z}_B - \mathbf{Q}_B \mathbf{Q}_B^\top \mathbf{Z}_B)$
  $(\overline{\mathbf{Q}}_A, \overline{\mathbf{R}}_A) \leftarrow \mathrm{QR}(\mathbf{Z}_A - \mathbf{Q}_A \mathbf{Q}_A^\top \mathbf{Z}_A)$
  $\mathbf{S} \leftarrow \mathbf{R}_B \mathbf{R}_A^\top - \eta \mathbf{R}_B^\top \nabla_{\mathbf{A}_i^t} L \cdot \mathbf{Q}_A$

  $\mathbf{U}_* \mathbf{\Sigma}_* \mathbf{V}_*^\top \leftarrow \mathrm{SVD}_r \begin{pmatrix} \mathbf{S} & -\eta \overline{\mathbf{R}}_A^\top \\ -\eta \overline{\mathbf{R}}_B & \mathbf{0} \end{pmatrix}$

  $\mathbf{B}_i^{t+1} \leftarrow (\mathbf{Q}_B, \overline{\mathbf{Q}}_B) \mathbf{U}_* \mathbf{\Sigma}_*^{1/2}$
  $\mathbf{A}_i^{t+1} \leftarrow \mathbf{\Sigma}_*^{1/2} \mathbf{V}_*^\top (\mathbf{Q}_A, \overline{\mathbf{Q}}_A)^\top$
**end if**
**return** $\mathbf{B}_i^{t+1}, \mathbf{A}_i^{t+1}$

---

**Complexity Analysis.** As shown in Algorithm 1, the algorithmic complexity of the approximate LORO update step is $O(rd)$ as it only involves element-wise operations. On the other hand, the exact update step includes QR decomposition of $d$-by-$r$ matrices with a complexity of $O(r^2 d)$, inversion of $r$-by-$r$ triangular matrices with a complexity of $O(r^2)$, SVD of $2r$-by-$2r$ matrices at $O(r^3)$, and matrix multiplications with a complexity of $O(r^2 d)$. Suppose the frequency of the exact update is $K$, then the average complexity of LORO is $O(r^2(r+d)/K + rd)$. In practice, we set the frequency $K$ in the same order as $r$, which simplifies the LORO complexity to $O(r^2 + rd)$. Therefore, when comparing with the standard Euclidean low-rank pretraining with a $O(rd)$ complexity, LORO does not introduce evident computational overhead.

## 5 Experiment Results

### 5.1 Efficient Low-rank Pretraining

**Experiment setup.** In this section, we evaluate the effectiveness of our proposed LORO in training low-rank language models from scratch. All the experiments are implemented in PyTorch (Paszke et al., 2019) and conducted on NVIDIA 40G A100 GPUs. For a fair comparison with existing baselines, we follow a standard experiment setup used in (Zhao et al., 2024; Lialin et al., 2024). Specifically, we adopt the LLaMA-based language model (Touvron et al., 2023b) equipped with RMSNorm (Zhang & Sennrich, 2019) and SwiGLU activations (Shazeer, 2020). We follow the same model configurations as (Zhao et al., 2024, Table 5) for the LLaMA-60M, -130M, -350M, and -1B models[3] and we conduct all the pretraining experiments using BF16 (bfloat16) format. We train all the models on the C4 (Colossal Clean Crawled Corpus) dataset (Raffel et al., 2019), a large-scale cleaned dataset designed for language models pretraining. The models are trained on a sufficiently large volume of data without repetition to simulate the pretraining setting in practice.

**LORO configurations.** We consider the low-rank parameterization that represents each target weight matrix as the product of a rank-$r$ tall matrix and a rank-$r$ wide matrix as defined in equation 2. In this paper, we apply low-rank parameterization to all weights in attention functions (e.g. query-, key-, value-, and out-projection) and all linear projection layers (e.g. down-, up-, and gate-projection) with the same rank $r$. Following the setup in (Zhao et al., 2024; Han et al., 2024), we keep the embedding layer and the output language model head to be full-size. We fix the LORO update frequency as $K = 500$ step for all pretraining experiments, and we only tune the learning rate. We refer the readers to Appendix A for implementation details of the experiments in this subsection.

**Baselines.** We compare our LORO against multiple baselines that involve low-rank structures. 'Full-size' denotes pretraining the full-size model with the Adam optimizer (Kingma & Ba, 2015). 'Low-rank' is a traditional low-rank training approach (Kamalakara et al., 2022). 'LoRA' (Hu et al.,

---
[3]Since this study primarily focuses on comparing performance using BF16 across various pertaining methods, we did not conduct experiments on larger models due to limited computational resources.

Table 1: Results of low-rank pretraining on C4 dataset, including validation perplexity (PPL), the number of parameters in millions (Param), the estimated total memory cost in GB (Mem), where 1GB contains $10^9$ bytes. $r$, $d_1$, and $d_2$ denote the rank, the hidden- and intermediate-dimension of the LLaMA model. Results of the baselines are reported from (Zhao et al., 2024; Han et al., 2024).

| Model (# Token) | LLaMA-60M (1.1B) | | | LLaMA-130M (2.2B) | | | LLaMA-350M (6.4B) | | | LLaMA-1B (13.1B) | | |
| $r$ / $d_1$ / $d_2$ | 128 / 512 / 1376 | | | 256 / 768 / 2048 | | | 256 / 1024 / 2736 | | | 512 / 2048 / 5461 | | |
| Method | PPL (↓) | Param | Mem (↓) | PPL (↓) | Param | Mem (↓) | PPL (↓) | Param | Mem (↓) | PPL (↓) | Param | Mem (↓) |
| Full-size | 34.06 | 58 | 0.35 | 24.36 | 134 | 0.81 | 18.80 | 368 | 2.21 | 15.56 | 1339 | 8.04 |
| Low-rank | 78.18 | 43 | 0.24 | 45.51 | 94 | 0.57 | 37.41 | 185 | 1.11 | 142.53 | 609 | 3.66 |
| LoRA | 34.99 | 58 | 0.36 | 33.92 | 134 | 0.84 | 25.58 | 368 | 1.85 | 19.21 | 1339 | 6.34 |
| ReLoRA | 37.04 | 58 | 0.36 | 29.37 | 134 | 0.84 | 29.08 | 368 | 1.85 | 18.33 | 1339 | 6.34 |
| GaLore | 34.88 | 58 | 0.28 | 25.36 | 134 | 0.61 | 18.95 | 368 | 1.59 | 15.64 | 1339 | 4.76 |
| SLTrain | 34.15 | 44 | 0.26 | 26.04 | 97 | 0.60 | 19.42 | 194 | 1.24 | 16.14 | 646 | 4.16 |
| LORO | **33.96** | 43 | 0.24 | **24.59** | 94 | 0.57 | **18.84** | 185 | 1.11 | **15.19** | 609 | 3.66 |

Table 2: Training efficiency metrics evaluated on $1\times40$G A100 GPU, including token batch size (TBS), ratio of memory cost reduction (MemRd), and throughput (Tokens / sec).

| Model | Method | PPL (↓) | TBS | Mem (↓) | MemRd (↑) | Tokens / sec (↑) | Speedup (↑) |
| --- | --- | --- | --- | --- | --- | --- | --- |
| | Full-size | 18.80 | 16K | 2.21 | 0% | 35264 | ×1 |
| LLaMA 350M | GaLore | 18.95 | 16K | 1.59 | 28% | 35427 | ×1 |
| | LORO | 18.84 | 16K | 1.11 | **50%** | 40640 | **×1.15** |
| | Full-size | 15.56 | 8K | 8.04 | 0% | 7890 | ×1 |
| LLaMA 1B | GaLore | 15.64 | 8K | 4.76 | 41% | 7886 | ×0.98 |
| | LORO | 15.19 | 8K | 3.66 | **54%** | 14351 | **×1.82** |

2022) is a low-rank fine-tuning method that reparameterizes each weight as $\mathbf{W} + \mathbf{BA}$, where $\mathbf{W}$ is a full-size matrix that remains fixed during training, and $\mathbf{B}$ and $\mathbf{A}$ are trainable low-rank factors. In pretraining scenarios, the full-size matrix $\mathbf{W}$ is randomly initialized. 'ReLoRA' (Lialin et al., 2024) is a variant of LoRA that are adapted to pretraining by periodically absorbing the updated $\mathbf{BA}$ into $\mathbf{W}$. 'GaLore' (Zhao et al., 2024) leverages a low-rank gradient to update the full-size weights. 'SLTrain' (Han et al., 2024) parameterizes each weight as the sum of low-rank factors and a sparse matrix, with both components being jointly optimized. However, all baselines, except for 'Low-rank', work with full-size models and do not engage in the challenging low-rank pretraining.

**LORO finds high-performing low-rank Models.** Table 1 shows that, our LORO consistently outperforms all other memory-efficient baselines in terms of both perplexity and total memory cost (including the memory usage of model parameters and optimizer states, as estimated in (Han et al., 2024, Appendix C)) across all settings. For LLaMA-1B model, our LORO achieves a perplexity score that is 2% lower than the full-size baseline. These results validate the existence of competitive low-rank models within the low-rank matrix manifold, which exhibit comparable or superior performance compared to the full-size baselines. Moreover, our LORO offers a practical approach for training low-rank models from scratch without sacrificing performance.

**LORO enhances training efficiency.** Section 1 shows that a low-rank model with $d_1 = d_2 = d$ is expected to achieve a $(2r/d)$-times reduction in memory and a $(d/2r)$-times speedup in inference. In practice, we set $r = d_1/4$ across all runs to pursue an approximate $\times 0.5$ FLOPS reduction and about $\times 0.5$ model size reduction. As shown in Table 2, for a LLaMA-1B model, LORO achieves $\times 1.82$ training speedup and a 54% reduction in memory compared to the full-size baseline, while GaLore does not exhibit practical speedup as it still works on the full-size model. We observe that the training speedup of LORO is less evident on LLaMA-350M model. This is because the data loading process introduces additional overhead and execution time, which overshadow the efficiency gains provided by LORO on smaller models.

**LORO enhances inference efficiency.** In Table 3, we empirically evaluate the inference efficiency of the low-rank LLaMA-1B model obtained by LORO with varying batch sizes and sequence lengths. Compared to the full-size baseline, our LORO achieves a $\times 2.08 \sim \times 4.04$ inference speedup and a $5\% \sim 17\%$ reduction in maximum memory allocation, across all settings. The speedup of LORO is evident for small batch sizes, and it converges to $\times 2$ as the batch size increases. This occurs as full-size baselines benefit more from GPU parallelization at larger batch sizes.

Table 3: Inference efficiency metrics on $1 \times 40$G A100 GPU, including maximum memory allocation in GB (MaxMem), memory reduction (MemRd), and total time of processing 500 batches.

| LLaMA 1B | | Batch size = 512 | | | | Batch size = 256 | | | |
|---|---|---|---|---|---|---|---|---|---|
| | Seq len | MaxMem ($\downarrow$) | MemRd ($\uparrow$) | Time ($\downarrow$) | Speedup ($\uparrow$) | Seq len | MaxMem ($\downarrow$) | MemRd ($\uparrow$) | Time ($\downarrow$) | Speedup ($\uparrow$) |
| Full-size | 128 | 26.47 | - | 4.31 | - | 256 | 26.50 | - | 6.67 | - |
| LORO | 128 | 25.07 | 5% | 1.57 | ×2.14 | 256 | 25.10 | 5% | 3.07 | ×2.17 |
| Full-size | 64 | 14.49 | - | 1.79 | - | 128 | 14.52 | - | 4.69 | - |
| LORO | 64 | 13.08 | 10% | 0.79 | ×2.27 | 128 | 13.11 | 10% | 1.54 | ×3.04 |
| Full-size | 32 | 8.49 | - | 1.41 | - | 64 | 8.52 | - | 3.28 | - |
| LORO | 32 | 7.09 | 17% | 0.43 | ×3.32 | 64 | 7.12 | 16% | 0.81 | ×4.04 |

| LLaMA 1B | | Batch size = 128 | | | | Batch size = 64 | | | |
|---|---|---|---|---|---|---|---|---|---|
| | Seq len | MaxMem ($\downarrow$) | MemRd ($\uparrow$) | Time ($\downarrow$) | Speedup ($\uparrow$) | Seq len | MaxMem ($\downarrow$) | MemRd ($\uparrow$) | Time ($\downarrow$) | Speedup ($\uparrow$) |
| Full-size | 512 | 26.52 | - | 13.21 | - | 1024 | 26.53 | - | 26.63 | - |
| LORO | 512 | 25.12 | 5% | 6.33 | ×2.09 | 1024 | 25.12 | 5% | 12.79 | ×2.08 |
| Full-size | 256 | 14.53 | - | 6.67 | - | 512 | 14.54 | - | 14.68 | - |
| LORO | 256 | 13.13 | 10% | 3.1 | ×2.15 | 512 | 13.14 | 10% | 6.31 | ×2.33 |
| Full-size | 128 | 8.54 | - | 3.73 | - | 256 | 8.55 | - | 7.59 | - |
| LORO | 128 | 7.14 | 16% | 1.61 | ×2.32 | 256 | 7.15 | 16% | 3.23 | ×2.35 |

Figure 3: Linear interpolation between LORO pretrained models and full-size baselines.

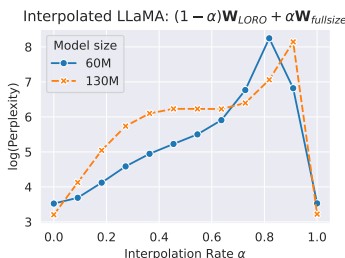

Interpolated LLaMA: $(1 - \alpha)\mathbf{W}_{LORO} + \alpha\mathbf{W}_{fullsize}$

Table 4: Comparison between LORO and other pretrained full-size architectures within the same parameter budget.

| Model # Token $r / d_1 / d_2$ | LLaMA-130M 2.2 B 256 / 768 / 2048 | | | LLaMA-350M 6.4 B 256 / 1024 / 2736 | | |
|---|---|---|---|---|---|---|
| Method | PPL ($\downarrow$) | Param | Mem ($\downarrow$) | PPL ($\downarrow$) | Param | Mem ($\downarrow$) |
| Full-size | 24.36 | 134 | 0.81 | 18.80 | 368 | 2.21 |
| Low-rank | 45.51 | 94 | 0.57 | 37.41 | 185 | 1.11 |
| Full-size Shallow | 28.49 | 94 | 0.57 | 21.67 | 185 | 1.11 |
| Full-size Slim | 25.34 | 94 | 0.57 | 21.40 | 185 | 1.11 |
| LORO + Slim MLP | 25.16 | 94 | 0.57 | 18.95 | 185 | 1.11 |
| LORO | **24.59** | 94 | 0.57 | **18.84** | 185 | 1.11 |

**LORO finds flatter minima.** As Table 1 shows that LORO is able to find minima within the low-rank matrix manifold with high performance comparable to the full-size models, we aim to compare between the low-rank minima found by LORO, denoted by $(\mathbf{B}_i^* \mathbf{A}_i^*)$, and the full-size minima, denoted by $(\mathbf{W}_i^*)$. To this end, we probe the loss landscape of these minima by via linear model interpolation (Lucas et al., 2021; Gueta et al., 2023). Specifically, for $\alpha \in \{0, 0.1, ..., 1\}$, we visualize the perplexity score of the model with weights $(1 - \alpha)\mathbf{B}_i^* \mathbf{A}_i^* + \alpha\mathbf{W}_i^*$. As shown in Figure 3, a steep loss barrier is observed between the low-rank LORO-pretrained models and the full-size pretrained models, indicating that the LORO minima do not reside in the same basin as the full-size minima. Furthermore, the loss barrier exhibits an asymmetry, with the side closer to the full-size models being significantly steeper than the side closer to the low-rank models. This implies that LORO can find flatter low-rank minima, which may explain why it sometimes outperforms the full-size baselines as shown in Table 2.

**LORO explores effective low-rank structures.** We compare the performance of LORO low-rank pretrained models against various full-size counterparts within the same parameter budget. Specifically, we evaluate three full-size variants of the original model: 1) 'Full-size Shallow,' with fewer layers than the full model, 2) 'Full-size Slim,' with a reduced hidden dimension but the same number of layers, and 3) 'LORO + Slim MLP,' where LORO is applied only to the attention weights, while the MLP layers are slimmer but retain full-size matrices. Their configurations are detailed in Appendix A. As reported in Table 4, the LORO low-rank pretrained models outperform full-size variants, demonstrating that LORO unlocks the potential of low-rank structures as a more effective parameterization scheme. We attribute this to the fact that, within the same parameter budget, low-rank structures benefit from a larger hidden dimension, which enhances their ability to learn high-dimensional features.

## 5.2 ABLATION STUDIES

**Ablation on the LORO rank $r$.** In Table 5, we evaluate the perplexity scores of LORO pretrained LLaMA-60M models with varying ranks. As anticipated, the performance decays as $r$ decreases. In practice, setting $r = d_1/4$ is preferable, as it achieves a 50% reduction in both memory and FLOPS while maintaining high performance.

Table 5: Ablation on LORO rank $r$.

| Model | | LLaMA-60M | | |
|---|---|---|---|---|
| # Token | | 1.1 B | | |
| $d_1 / d_2$ | | 512 / 1376 | | |
| Method | Rank | PPL ($\downarrow$) | Param | Mem ($\downarrow$) |
| Full-size | 512 | 34.06 | 58 | 0.35 |
| Low-rank | 128 | 78.18 | 43 | 0.24 |
| LORO | 256 | 31.54 | 53 | 0.31 |
| LORO | 128 | 33.96 | 43 | 0.24 |
| LORO | 64 | 39.40 | 38 | 0.22 |
| LORO | 32 | 49.88 | 35 | 0.21 |

Figure 4: Ablation on LORO exact update frequency $K$.

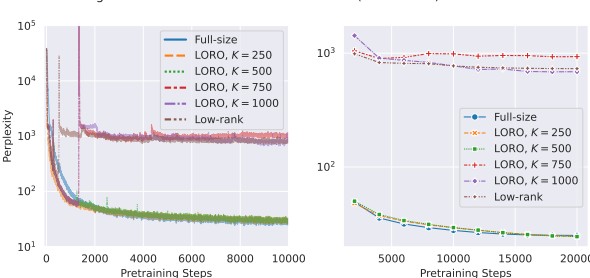

Table 6: Extending LORO to fine-tuning RoBERTa-base models on GLUE benchmark. Results of the baselines are reported from (Zhao et al., 2024; Han et al., 2024).

| RoBERTa-base | Memory | CoLA | STS-B | MRPC | RTE | SST2 | MNLI | QNLI | QQP | Avg |
|---|---|---|---|---|---|---|---|---|---|---|
| Full-size | 747M | 62.24 | 90.92 | 91.3 | 79.42 | 94.57 | 87.18 | 92.33 | 92.28 | 86.28 |
| LoRA, $r = 4$ | 257M | 61.38 | 90.57 | 91.07 | 78.70 | 92.89 | 86.82 | 92.18 | **91.29** | 85.61 |
| Galore, $r = 4$ | 253M | 60.35 | 90.73 | 92.25 | **79.42** | 94.04 | **87.00** | 92.24 | 91.06 | **85.89** |
| LORO, $r = 4$ | 257M | **61.58** | **90.75** | **92.60** | 78.69 | **94.14** | 86.15 | **92.58** | 90.64 | **85.89** |
| LoRA, $r = 8$ | 264M | 61.83 | 90.80 | 91.90 | 79.06 | 93.46 | 86.94 | 92.25 | 91.22 | 85.93 |
| Galore, $r = 8$ | 257M | 60.06 | 90.82 | 92.01 | **79.78** | 94.38 | **87.17** | 92.2 | 91.11 | 85.94 |
| SLTrain, $r = 8$ | - | 60.35 | 90.74 | 92.38 | 79.42 | 94.15 | 86.53 | 92.40 | **91.27** | 85.93 |
| LORO, $r = 8$ | 257M | **63.10** | **90.97** | **93.12** | 78.92 | **94.61** | 86.65 | **92.75** | 91.00 | **86.39** |

**Ablation on the LORO exact update frequency $K$.** In Figure 4, we compare the convergence of different runs of LORO with varying exact update frequencies. It is observed that for $K \geqslant 750$ when the LORO exact update is overly lazy, the training curve tends to explode and fail to converge to performing minima. In practice, we recommend setting $K = 500$, as it significantly reduces training overhead and results in competitive low-rank models with performance comparable to full-size baselines across LLaMA models of varying sizes, ranging from 60M to 1B parameters.

### 5.3 Efficient Low-rank Fine-tuning

Following the experiment setup in (Zhao et al., 2024, Section 5.4), we extend our LORO to fine-tune the pretrained RoBERTa-base model (Liu et al., 2019) on GLUE datasets (Wang et al., 2019). Specifically, we reparameterized each weight in the query- and key-projections as $\mathbf{W} + \mathbf{BA}$, where $\mathbf{W}$ is the full-size pretrained weight and $\mathbf{B}$ and $\mathbf{A}$ are low-rank factors. Then, we apply LORO to $\mathbf{B}$ and $\mathbf{A}$ exclusively and freeze the remaining parameters. The hyperparameters are detailed in Appendix A.2. In Table 6, we compare the fine-tuning performance of LORO with three baselines, including 'LoRA' (Hu et al., 2022), 'GaLore' (Zhao et al., 2024), and 'SLTrain' (Han et al., 2024). Although LORO achieves decent performance, its outperformance over other baselines in fine-tuning is less significant than in pretraining. We argue that, with full-size pretrained weights provided, fine-tuning requires learning only a small amount of domain-specific knowledge, which can be easily captured through low-rank adjustments in each weight matrix. As a result, the potential for improvement with different low-rank techniques is limited, and the performance gains become less noticeable.

## 6 Conclusion

In this paper, we proposed LORO for efficient yet effective pretraining of low-rank parameterized language models. Specifically, LORO optimizes the full-size product of the low-rank factors toward the steepest gradient descent direction on the low-rank matrix manifold, without computing any expensive full-size weight or gradient. Extensive experiments demonstrate that LORO can discover competitive low-rank models with performance comparable to full-size baselines, while providing significant memory reduction and acceleration in both training and inference. This work demonstrates the potential of low-rank structures in language model pretraining, inspiring us to explore efficient low-rank pretraining methods for vision and multi-modal generative models.

ACKNOWLEDGEMENT

The research work described in this paper was conducted in the JC STEM Lab of Machine Learning and Symbolic Reasoning funded by The Hong Kong Jockey Club Charities Trust. Sinno J. Pan also thanks the support of the Microsoft Research Asia collaborative research grant.

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

# A    IMPLEMENTATION DETAILS

## A.1    PRETRAINING LLAMA MODELS ON C4 DATASET

The implementation details introduced in this subsection apply to the experiments in Table 1, Table 2, Table 3, Table 4, Table 5, Figure 3, and Figure 4. To achieve a fair comparison with other pretraining baselines, we follow the same model architecture and pretraining hyperparameters as in (Zhao et al., 2024, Appendix C.1). For all runs, we set the max data sequence length as 256 with a batch size of 512 (i.e., a token batch size of 131K). For a fair comparison in terms of the pretraining efficiency, we run all the experiments on $1\times$ NVIDIA 40G A100 GPU, without using model parallelization or data parallelization. To achieve the 512 training batch size, we adjust the per-device batch size and vary the number of gradient accumulation steps according to the size of the LLaMA models. Based on (Zhao et al., 2024, Appendix C.1), we employ a learning rate warmup starting from 0 during the first 10% of the pretraining steps, followed by a cosine annealing scheduler that decays to 10% of the maximum learning rate. We initialize the low-rank factors with Xavier initialization (Glorot & Bengio, 2010) across all experiments.

We implement LORO following the pseudo-code attached in 2. Following (Zhao et al., 2024; Touvron et al., 2023b), we impose low-rank parameterization to all the weight matrices in the self-attention modules (including `q_proj`, `k_proj`, `v_proj`, and `o_proj`) and linear projection modules (including `down_proj`, `up_proj`, and `gate_proj`) and we update them with LORO, while we apply the standard Adam optimizer to update the `token_emb`, `lm_head`, and other layer normalization modules. All modules share the same learning rate. Specifically, for all LORO pretraining results reported in Section 5.1, we set the LORO exact update frequency to $K = 500$ and the learning rate to 0.01 for the LLaMA-60M, -130M, and -350M models, while for LLaMA-1B, we set $K = 200$ and the learning rate to 0.005. To stabilize the training process, at each LORO exact update step, we apply a 5-step linear learning rate warmup and we refresh the Adam statistics.

## A.2    FINE-TUNING ROBERTA MODELS ON C4 DATASET

For a fair comparison on fine-tuning with other baselines, we follow the same model architecture, datasets and evaluation processes as in (Zhao et al., 2024, Appendix D.1). We report Matthew's correlation score for CoLA, the Pearson correlation score for STS-B, matched and mismatched accuracy for MNLI, F1 score for MRPC, and accuracy for other tasks. Specifically, we fine-tune the pretrained RoBERTa-base checkpoint from Huggingface[4] on the GLUE benchmark. For each weight matrices in `q_proj` and `k_proj`, we adopt the standard LoRA parameterization $\mathbf{W} + \lambda\mathbf{BA}$, where $\mathbf{W}$ denotes the full-size pretrained weight, $\lambda$ is the LoRA scale hyperparameter, and $\mathbf{B}$ and $\mathbf{A}$ are the trainable low-rank factors. During fine-tuning, we apply LORO to update the low-rank factors in the same manner as in pretraining, while keeping all other parameters unchanged. The training hyperparameters are attached in Table 8.

Table 7: Model architecture and pretraining settings. $d_1$ and $d_2$ denotes the hidden- and intermediate-dimension of the LLaMA model. '# Tokens' denotes the amount of training tokens, 'PD-BSZ' denotes the per-device batch size, and '# Grad Accum' denotes the number of gradient accumulation steps.

| Model | Params | $d_1$ | $d_2$ | Heads | Layers | Steps | # Tokens | PD-BSZ | # Grad Accum |
|---|---|---|---|---|---|---|---|---|---|
| 60M | 58M | 512 | 1376 | 8 | 8 | 10K | 1.1B | 256 | 2 |
| 130M | 134M | 768 | 2048 | 12 | 12 | 20K | 2.2B | 128 | 4 |
| 94M Shallow | 94M | 820 | 1720 | 12 | 6 | 20K | 2.2B | 128 | 4 |
| 94M Slim | 94M | 640 | 1250 | 12 | 12 | 20K | 2.2B | 128 | 4 |
| 94M Slim MLP | 94M | 768 | 928 | 12 | 12 | 20K | 2.2B | 128 | 4 |
| 368M | 368M | 1024 | 2736 | 16 | 24 | 60K | 6.4B | 64 | 8 |
| 185M Shallow | 185M | 1024 | 1880 | 16 | 12 | 60K | 6.4B | 64 | 8 |
| 185M Slim | 185M | 736 | 1640 | 16 | 24 | 60K | 6.4B | 64 | 8 |
| 185M Slim MLP | 185M | 1024 | 940 | 16 | 24 | 60K | 6.4B | 64 | 8 |
| 1B | 1339M | 2048 | 5461 | 24 | 32 | 100K | 13.1B | 32 | 16 |

---

[4]https://huggingface.co/docs/transformers/model_doc/roberta

Table 8: Hyperparameters of LORO in fine-tuning RoBERTa experiments.

|  |  | CoLA | STS-B | MRPC | RTE | SST-2 | MNLI | QNLI | QQP |
|---|---|---|---|---|---|---|---|---|---|
| Rank $r = 4$ | Batch Size | 64 | 16 | 16 | 16 | 16 | 16 | 16 | 16 |
|  | # Epochs | 30 | 40 | 20 | 30 | 40 | 30 | 30 | 30 |
|  | Learning Rate | 4e-4 | 4e-4 | 2e-4 | 5e-4 | 5e-5 | 5e-5 | 4e-4 | 5e-5 |
|  | LoRA $\lambda$ | 32 | 8 | 32 | 8 | 8 | 8 | 8 | 8 |
|  | LORO Freq $K$ | 100 | 200 | 100 | 100 | 100 | 100 | 100 | 100 |
| Rank $r = 8$ | Batch Size | 64 | 16 | 16 | 16 | 16 | 16 | 16 | 16 |
|  | # Epochs | 30 | 40 | 20 | 30 | 40 | 30 | 30 | 30 |
|  | Learning Rate | 4e-4 | 4e-4 | 2e-4 | 5e-4 | 5e-5 | 5e-5 | 4e-4 | 5e-5 |
|  | LoRA $\lambda$ | 16 | 16 | 8 | 8 | 8 | 16 | 8 | 16 |
|  | LORO Freq $K$ | 20 | 100 | 100 | 100 | 500 | 200 | 100 | 200 |

### A.3 PYTORCH IMPLEMENTATION OF LORO

**Algorithm 2** **Lo**w-rank **R**iemannian **O**ptimizer (**LORO**) in PyTorch (Paszke et al., 2019)

```python
def LORO_step(B,A,lr,n_step,K):
    # B: [d, r], A: [d, r]
    if (n_step + 1) % K != 0:                # Approximate Riemannian update
        d, r = B.shape
        B = B - lr * (r / d) * B.grad
        A = A - lr * (r / d) * A.grad
    else:
        dB, dA = B.grad, A.grad # dB: [d, r], dA: [r, d]
        Qb, Rb = torch.qr(B) # Qb: [d, r], Rb: [r, r]
        Qa, Ra = torch.qr(A) # Qa: [d, r], Ra: [r, r]
        # dB_Ra_inv: [d, r], dA_Rb_inv: [d, r]
        dB_Ra_inv = torch.linalg.solve(Ra.T, dB.T).T
        dA_Rb_inv = torch.linalg.solve(Rb.T, dA).T
        # Qb_: [d, r], Rb_: [r, r], Qa_: [d, r], Ra_: [r, r]
        Qb_, Rb_ = torch.qr(dB_Ra_inv - Qb @ Qb.T @ dB_Ra_inv)
        Qa_, Ra_ = torch.qr(dA_Rb_inv - Qa @ Qa.T @ dA_Rb_inv)
        S = Rb @ Ra.T - lr * dA_Rb_inv.T @ Qa # [r, r]
        # SVD on [2r, 2r], U: [2r, r], Sig: [r,r], V: [2r,r]
        U, Sig, V = torch.svd(mat([[S, - lr * Ra_.T], [- lr * Rb, zeros]]))

        B = mat([Qb, Qb_]) @ U @ Sig.sqrt()
        A = Sig.sqrt() @ V.T @ mat([Qa, Qa_]).T  # Exact Riemannian update

    return B, A
```

## B PROOF DETAILS

**Proposition 3** (Riemannian Gradient under Low-rank Parameterization). *Following the previous notation, for any low-rank parameterized weight* $\mathbf{W} = \mathbf{BA} \in \mathcal{M}_r$, *where* $\mathbf{B} \in \mathbb{R}^{d \times r}$ *and* $\mathbf{A} \in \mathbb{R}^{r \times d}$, *the Riemannian gradient of any smooth function* $L(\cdot) : \mathbb{R}^{d \times d} \mapsto \mathbb{R}$ *evaluated at* $\mathbf{W}$ *is*

$$\text{grad } L(\mathbf{W}) = \mathbf{B}(\mathbf{B}^\top \mathbf{B})^{-1} \cdot \nabla_\mathbf{A} L + \nabla_\mathbf{B} L \cdot (\mathbf{AA}^\top)^{-1} \mathbf{A} - \mathbf{B}(\mathbf{B}^\top \mathbf{B})^{-1} \cdot \nabla_\mathbf{A} L \cdot \mathbf{A}^\top (\mathbf{AA}^\top)^{-1} \mathbf{A}. \quad (12)$$

*Proof of Proposition 3.* We first need to show that, for any ambient point $\mathbf{G} \in \mathbb{R}^{d \times d}$, the orthogonal projection to the tangent space $T_{(\mathbf{BA})} \mathcal{M}_r$ is given by

$$\mathcal{P}(\mathbf{G}) = \mathbf{B}(\mathbf{B}^\top \mathbf{B})^{-1} \mathbf{B}^\top \cdot \mathbf{G} + \mathbf{G} \cdot \mathbf{A}^\top (\mathbf{AA}^\top)^{-1} \mathbf{A} - \mathbf{B}(\mathbf{B}^\top \mathbf{B})^{-1} \mathbf{B}^\top \cdot \mathbf{G} \cdot \mathbf{A}^\top (\mathbf{AA}^\top)^{-1} \mathbf{A}.$$

Then, the proof can be completed by taking $\mathbf{G} = \nabla_\mathbf{W} L$, and showing that $\nabla_\mathbf{W} L \cdot \mathbf{A}^\top = \nabla_\mathbf{B} L$ and $\mathbf{B} \cdot \nabla_\mathbf{W} L = \nabla_\mathbf{A} L$.

According to Proposition 1 ((Lee, 2019, Example 8.14) and (Absil et al., 2009, Equation 3.37)), the othorgonal operator is $\mathcal{P}(\mathbf{G}) \triangleq \mathbf{UU}^\top \cdot \mathbf{G} + \mathbf{G} \cdot \mathbf{VV}^\top - \mathbf{UU}^\top \cdot \mathbf{G} \cdot \mathbf{VV}^\top$, where $\mathbf{W} = \mathbf{U\Sigma V}^\top$ is the singular value decomposition (SVD) of $\mathbf{W}$. Suppose the QR decompositions of $\mathbf{B}$ and $\mathbf{A}^\top$ are

$\mathbf{Q}_B \mathbf{R}_B = \mathbf{B}$ and $\mathbf{Q}_A \mathbf{R}_A = \mathbf{A}^\top$, and the SVD of the full-rank $r$-by-$r$ matrix $\mathbf{R}_B \mathbf{R}_A^\top$ is $\mathbf{U}_R \boldsymbol{\Sigma}_R \mathbf{V}_R^\top$. Then, it holds that $(\mathbf{Q}_B \mathbf{U}_R) \boldsymbol{\Sigma}_R (\mathbf{V}_R^\top \mathbf{Q}_A^\top) = \mathbf{U} \boldsymbol{\Sigma} \mathbf{V}^\top$, where $(\mathbf{Q}_B \mathbf{U}_R), (\mathbf{V}_R^\top \mathbf{Q}_A^\top) \in \mathbb{R}^{d \times r}$ are column-wise orthogonal and $\boldsymbol{\Sigma}_R \in \mathbb{R}^{r \times r}$ is diagonal. By the uniqueness of SVD, it holds that $\mathbf{Q}_B \mathbf{U}_R = \mathbf{U}$ and $\mathbf{Q}_A \mathbf{V}_R = \mathbf{V}$, which implies

$$\mathbf{U}\mathbf{U}^\top = \mathbf{Q}_B \mathbf{U}_R \mathbf{U}_R^\top \mathbf{Q}_B^\top = \mathbf{Q}_B \mathbf{Q}_B^\top = (\mathbf{Q}_B \mathbf{R}_B)(\mathbf{R}_B^{-1} \mathbf{Q}_B^\top \mathbf{Q}_B \mathbf{R}_B^{-\top})(\mathbf{R}_B^\top \mathbf{Q}_B^\top) \tag{13}$$

$$= (\mathbf{Q}_B \mathbf{R}_B)(\mathbf{R}_B^\top \mathbf{Q}_B^\top \mathbf{Q}_B \mathbf{R}_B)^{-1}(\mathbf{R}_B^\top \mathbf{Q}_B^\top) = \mathbf{B}(\mathbf{B}^\top \mathbf{B})^{-1} \mathbf{B}^\top. \tag{14}$$

Similarly, we have $\mathbf{V}\mathbf{V}^\top = \mathbf{A}^\top(\mathbf{A}\mathbf{A}^\top)^{-1}\mathbf{A}$, which completes the proof. In intuition, the derivation above proves that the row-space of $\mathbf{B}$ coincides with that of $\mathbf{U}$, and the column-space of $\mathbf{A}^\top$ coincides with that of $\mathbf{V}$. Therefore, the orthogonal projection to the tangent space can be reparameterized under the basis of $\mathbf{B}$ and $\mathbf{A}$.

$\square$

**Proposition 4** (Riemannian Update under Low-rank Parameterization). *Following the previous notations, for any low-rank parameterized weight* $\mathbf{W} = \mathbf{B}\mathbf{A} \in \mathcal{M}_r$, *let* $\mathbf{B} = \mathbf{Q}_B \mathbf{R}_B$ *and* $\mathbf{A}^\top = \mathbf{Q}_A \mathbf{R}_A$ *be the QR-decomposition of* $\mathbf{B}$ *and* $\mathbf{A}^\top$, *such that* $\mathbf{Q}_B, \mathbf{Q}_A \in \mathbb{R}^{d \times r}$ *are column-wise orthogonal, and* $\mathbf{R}_B, \mathbf{R}_A \in \mathbb{R}^{r \times r}$ *are upper-triangular. Then the Riemannian updated weight, defined by* $\mathbf{W}' \triangleq \mathrm{SVD}_r(\mathbf{W} - \eta \operatorname{grad} L(\mathbf{W}))$, *admits a low-rank factorization* $\mathbf{W}' = \mathbf{B}'\mathbf{A}'$, *with* $\mathbf{B}' = (\mathbf{Q}_B, \overline{\mathbf{Q}}_B) \mathbf{U}_* \boldsymbol{\Sigma}_*^{1/2}$ *and* $\mathbf{A}' = \boldsymbol{\Sigma}_*^{1/2} \mathbf{V}_*^\top (\mathbf{Q}_A, \overline{\mathbf{Q}}_A)^\top$. *Specifically,* $\mathbf{U}_* \boldsymbol{\Sigma}_* \mathbf{V}_*^\top$ *is determined by the rank-$r$ SVD of the following $2r$-by-$2r$ matrix, i.e.,*

$$\mathbf{U}_* \boldsymbol{\Sigma}_* \mathbf{V}_*^\top = \mathrm{SVD}_r \begin{pmatrix} \mathbf{S} & -\eta\, \overline{\mathbf{R}}_A^\top \\ -\eta\, \overline{\mathbf{R}}_B & \mathbf{0} \end{pmatrix} \in \mathbb{R}^{2r \times 2r}, \quad \mathbf{S} \triangleq \mathbf{R}_B \mathbf{R}_A^\top - \eta\, \mathbf{R}_B^{-\top} \nabla_{\mathbf{A}} L \cdot \mathbf{Q}_A \in \mathbb{R}^{r \times r}, \tag{15}$$

*and* $\overline{\mathbf{Q}}_B \overline{\mathbf{R}}_B$ *is the QR decomposition of the $d$-by-$r$ matrix* $(\nabla_{\mathbf{B}} L \cdot \mathbf{R}_A^{-1} - \mathbf{Q}_B \mathbf{Q}_B^\top \nabla_{\mathbf{B}} L \cdot \mathbf{R}_A^{-1})$, *while* $\overline{\mathbf{Q}}_A \overline{\mathbf{R}}_A$ *is the QR decomposition of the $d$-by-$r$ matrix* $(\nabla_{\mathbf{A}} L^\top \mathbf{R}_B^{-1} - \mathbf{Q}_A \mathbf{Q}_A^\top \nabla_{\mathbf{A}} L^\top \mathbf{R}_B^{-1})$.

*Proof of Proposition 4.* This proof follows a similar spirit of orthogonal tangent vector decomposition as in (Vandereycken, 2013, Section 3) and (Bian et al., 2023, Section 2.2.3). To complete the proof, we need to perform such decomposition under the unnormalized basis spanned by $\mathbf{B}$ and $\mathbf{A}$, such that the full-size weight and gradient terms can be canceled out using the relation $\nabla_{\mathbf{W}} L \cdot \mathbf{A}^\top = \nabla_{\mathbf{B}} L$ and $\mathbf{B} \cdot \nabla_{\mathbf{W}} L = \nabla_{\mathbf{A}} L$.

As the low-rank representation of the Riemannian gradient is provided in Proposition 3, we only need to manipulate the terms in the updated weight $\mathbf{W} - \eta \operatorname{grad} L(\mathbf{W})$ such that it can be decomposed as $\widetilde{\mathbf{U}} \widetilde{\boldsymbol{\Sigma}} \widetilde{\mathbf{V}}^\top$, where $\widetilde{\mathbf{U}}, \widetilde{\mathbf{V}} \in \mathbb{R}^{d \times 2r}$ are column-wise othorgonal matrices. Thereby, the Riemannian SVD retraction can be performed on a smaller $2r$-by-$2r$ matrix instead, since

$$\mathrm{SVD}_r(\widetilde{\mathbf{U}} \widetilde{\boldsymbol{\Sigma}} \widetilde{\mathbf{V}}^\top) = \widetilde{\mathbf{U}} \cdot \mathrm{SVD}_r(\widetilde{\boldsymbol{\Sigma}}) \cdot \widetilde{\mathbf{V}}^\top. \tag{16}$$

To complete the proof, we only need to identify the representation of $\widetilde{\mathbf{U}}, \widetilde{\boldsymbol{\Sigma}}, \widetilde{\mathbf{V}}$ by orthogonal tangent vector decomposition, under low-rank parameterization. Based on Proposition 3, it holds that

$\operatorname{grad} L(\mathbf{W})$
$= \mathbf{B}(\mathbf{B}^\top \mathbf{B})^{-1} \mathbf{B}^\top \cdot \nabla_{\mathbf{W}} L + \nabla_{\mathbf{W}} L \cdot \mathbf{A}^\top (\mathbf{A}\mathbf{A}^\top)^{-1} \mathbf{A} - \mathbf{B}(\mathbf{B}^\top \mathbf{B})^{-1} \mathbf{B}^\top \cdot \nabla_{\mathbf{W}} L \cdot \mathbf{A}^\top (\mathbf{A}\mathbf{A}^\top)^{-1} \mathbf{A}$
$= \mathbf{Q}_B \mathbf{Q}_B^\top \cdot \nabla_{\mathbf{W}} L + \nabla_{\mathbf{W}} L \cdot \mathbf{Q}_A \mathbf{Q}_A^\top - \mathbf{Q}_B \mathbf{Q}_B^\top \cdot \nabla_{\mathbf{W}} L \cdot \mathbf{Q}_A \mathbf{Q}_A^\top$
$= \mathbf{Q}_B \mathbf{Q}_B^\top \cdot \nabla_{\mathbf{W}} L \cdot \mathbf{Q}_A \mathbf{Q}_A^\top + (\mathbf{I} - \mathbf{Q}_B \mathbf{Q}_B^\top) \cdot \nabla_{\mathbf{W}} L \cdot \mathbf{Q}_A \mathbf{Q}_A^\top + \mathbf{Q}_B \mathbf{Q}_B^\top \cdot \nabla_{\mathbf{W}} L \cdot (\mathbf{I} - \mathbf{Q}_A \mathbf{Q}_A^\top),$

where the latter two terms are orthogonal to the column space of $\mathbf{B}$ and $\mathbf{A}^\top$ respectively. To cancel out the full-size gradient terms, we leverage the relations $\mathbf{Q}_B^\top \cdot \nabla_{\mathbf{W}} L = \mathbf{R}_B^{-\top} \nabla_{\mathbf{A}} L$ and $\nabla_{\mathbf{W}} L \cdot \mathbf{Q}_A = \nabla_{\mathbf{B}} L \cdot \mathbf{R}_A^{-1}$, which shows that

$\operatorname{grad} L(\mathbf{W})$
$= \mathbf{Q}_B \mathbf{R}_B^{-\top} \nabla_{\mathbf{A}} L \cdot \mathbf{Q}_A \mathbf{Q}_A^\top + (\mathbf{I} - \mathbf{Q}_B \mathbf{Q}_B^\top) \cdot \nabla_{\mathbf{B}} L \cdot \mathbf{R}_A^{-1} \cdot \mathbf{Q}_A^\top + \mathbf{Q}_B \mathbf{R}_B^{-\top} \nabla_{\mathbf{A}} L \cdot (\mathbf{I} - \mathbf{Q}_A \mathbf{Q}_A^\top).$

Therefore, the updated weight can be represented by

$$
\begin{aligned}
&\mathbf{W} - \eta \operatorname{grad} L(\mathbf{W}) \\
=& \mathbf{Q}_B (\mathbf{R}_B \mathbf{R}_A^\top) \mathbf{Q}_A^\top - \eta \left( \mathbf{Q}_B \mathbf{R}_B^{-\top} \nabla_{\mathbf{A}} L \cdot \mathbf{Q}_A \mathbf{Q}_A^\top \right. \\
& \left. \quad + (\mathbf{I} - \mathbf{Q}_B \mathbf{Q}_B^\top) \cdot \nabla_{\mathbf{B}} L \cdot \mathbf{R}_A^{-1} \cdot \mathbf{Q}_A^\top + \mathbf{Q}_B \mathbf{R}_B^{-\top} \nabla_{\mathbf{A}} L \cdot (\mathbf{I} - \mathbf{Q}_A \mathbf{Q}_A^\top) \right) \\
=& \mathbf{Q}_B \left( \mathbf{R}_B \mathbf{R}_A^\top - \eta \, \mathbf{R}_B^{-\top} \nabla_{\mathbf{A}} L \cdot \mathbf{Q}_A \right) \mathbf{Q}_A^\top - \eta \, \overline{\mathbf{Q}}_B \cdot \overline{\mathbf{R}}_B \cdot \mathbf{Q}_A^\top - \eta \, \mathbf{Q}_B \cdot \overline{\mathbf{R}}_A^\top \cdot \overline{\mathbf{Q}}_A^\top \\
=& \begin{pmatrix} \mathbf{Q}_B & \overline{\mathbf{Q}}_B \end{pmatrix} \begin{pmatrix} \mathbf{R}_B \mathbf{R}_A^\top - \eta \, \mathbf{R}_B^{-\top} \nabla_{\mathbf{A}} L \cdot \mathbf{Q}_A & -\eta \, \overline{\mathbf{R}}_A^\top \\ -\eta \, \overline{\mathbf{R}}_B & \mathbf{0} \end{pmatrix} \begin{pmatrix} \mathbf{Q}_A^\top \\ \overline{\mathbf{Q}}_B^\top \end{pmatrix}.
\end{aligned}
$$

Notice that $(\mathbf{Q}_B, \overline{\mathbf{Q}}_B)$ and $(\mathbf{Q}_A, \overline{\mathbf{Q}}_A)$ are column-wise orthogonal matrices, making them a desirable choice of the aforementioned $\widetilde{\mathbf{U}}$ and $\widetilde{\mathbf{V}}$, and hence completes the proof.

$\square$

## C  FURTHER DISCUSSIONS ON THE MECHANISM OF LORO

In this section, we provide a supplementary analysis and discussion of the LORO mechanism. Specifically, Section C.1 analyzes the limitations of standard low-rank pretraining, while Section C.2 discusses how LORO effectively overcomes these limitations, leading to significant improvements.

### C.1  LIMITATIONS IN STANDARD LOW-RANK OPTIMIZER

In this subsection, we present additional theoretical and empirical evidence demonstrating that the standard low-rank optimizer (i.e. independently optimizing low-rank factors) is inefficient for achieving satisfactory training performance.

In Section C.1.1, our empirical evidence demonstrates that: 1. the standard low-rank optimizer produces spiky training curves and high perplexity across various initialization schemes and learning rates, 2. it learns low-rank weights with high condition numbers (i.e., the ratio of the largest singular value to the smallest nonzero singular value), and 3. it continues to yield poor performance even when combined with optimizer state refreshing.

In Section C.1.2, our theoretical analysis indicates that: 1. the standard low-rank gradient in (3) fails to preserve adequate information from the full-size gradient within the tangent space, and 2. a nearly-stationary point obtained through standard low-rank gradient descent with a near-zero gradient norm may be significantly far away from a true stationary point on the manifold.

Based on these observations and analysis, we conjecture that the failure of the standard low-rank optimizer stems from the following factors: 1. the standard low-rank gradient contains redundant full-size gradient information, preventing it from accurately approximating the steepest descent direction on the manifold, 2. the lack of proper scaling in the standard low-rank gradient causes deviations from the steepest descent direction on the manifold, and 3. the standard low-rank gradient biases towards ill-conditioned weights, leading to instability during training.

### C.1.1  EMPIRICAL EVIDENCE

**Standard low-rank optimizer generally leads to poor performance.** To validate that pretraining a low-rank language model is generally hard, we train a rank-128 LLaMA-60M model with a standard low-rank optimizer under different initialization schemes and learning rates. Specifically, we initialize the low-rank factors with Xavier initialization (Glorot & Bengio, 2010), or standard Gaussian initialization with standard deviation varying in $\{0.1, 0.01, 0.001\}$, and we tried different learning rates in $\{0.01, 0.005, 0.001\}$. We report the training curve and the final perplexity of the models.

As shown in Figure 5, in all settings, the training curve of the vanilla low-rank optimizer is spiky and unstable, leading to unsatisfactory evaluation perplexity. This suggests that independent updates of the low-rank factors are insufficient for good performance across different configurations. Training a low-rank language model to match the full-size baseline is generally challenging.

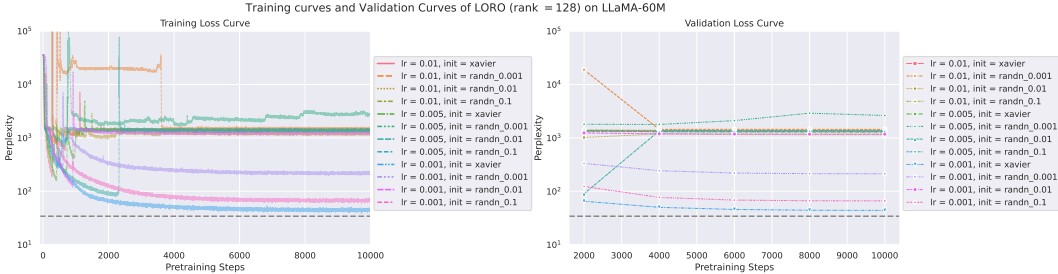

Figure 5: Ablation study on the impact of learning rate and initialization schemes on standard low-rank training for a rank-$128$ LLaMA-60M model. The y-axis (log-scale) represents perplexity, with the gray horizontal dashed line indicating the performance of full-size training.

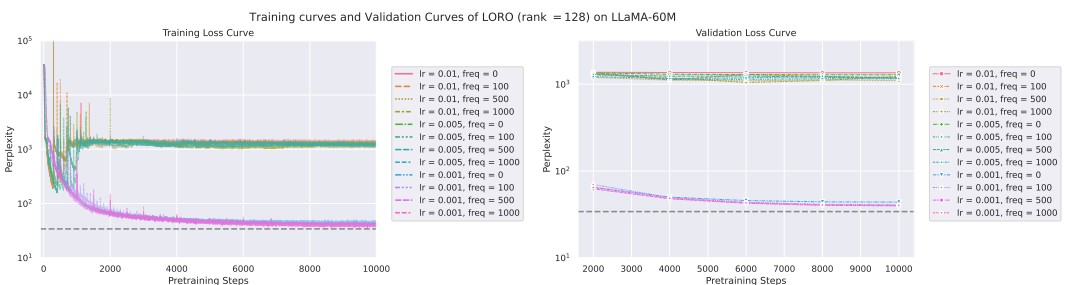

Figure 6: Ablation study on the impact of optimizer state refreshing on standard low-rank training for a rank-$128$ LLaMA-60M model. The y-axis (log-scale) represents perplexity, with the gray horizontal dashed line indicating the performance of full-size training.

**Standard low-rank optimizer leads to poor performance even when combined with optimizer state refreshing.** To assess whether optimizer state refreshing can enhance the standard low-rank optimizer, we perform an ablation study by varying the refreshing frequency in the standard low-rank optimizer. Following the setup in Section 5.1, we trained rank-$128$ LLaMA-60M models using vanilla low-rank gradient descent with different learning rates (chosen from $\{0.01, 0.005, 0.001\}$) and optimizer state refreshing frequencies (chosen from $\{0, 100, 500, 1000\}$).

As shown in Figure 6, all trials exhibit spiky training curves and unsatisfactory performance. This validates that applying optimizer state refreshing to standard low-rank gradient descent still leads to bad training performance. We conjecture that this is because the standard low-rank gradient does not actively explore new optimization subspace, and the training process biases towards low-rank weights with increasing condition numbers. In this case, refreshing the past momentum statistics resided does not help in exploring beneficial optimization subspace.

**Standard low-rank optimizer learns ill-conditioned weights.** To understand the cause of the instability and spiky training curves in the standard low-rank optimizer, we inspect the condition number and norm of the low-rank factors throughout the training process. Specifically, we train rank-$128$ LLaMA-60M models with Xavier initialization and a learning rate of $0.01$, using different optimizers: the standard low-rank optimizer, the full-size Adam optimizer (Kingma & Ba, 2015), and LORO. We visualize the evolution of the singular values, Frobenius norm, and condition numbers of the learned weights at various layers throughout the training process.

As shown in Figure 13, Figure 14, and Figure 15, the standard low-rank gradient results in model weights with significantly higher condition numbers compared to LORO or full-size Adam. We conjecture that the standard low-rank gradient update introduces imbalances among the singular values of the low-rank factors, thereby impairing the stability of the feature learning process.

### C.1.2 THEORETICAL EVIDENCE

**Standard low-rank gradient fails to approximate the full-size gradient well.** To understand the difference between the training dynamic of the standard low-rank optimizer and its full-size

counterpart, we examine the gap between the standard low-rank gradient and the full-size gradient. Following (3), we define the standard low-rank gradient as $\mathbf{Z} \triangleq \mathbf{B}\mathbf{B}^\top \nabla_\mathbf{W} L + \nabla_\mathbf{W} L \mathbf{A}^\top \mathbf{A}$, where $\nabla_\mathbf{W} L$ is the full-size gradient and $\mathbf{W} = \mathbf{B}\mathbf{A}$ is the low-rank weight. We show that $\mathbf{Z}$ fails to adequately preserve the full-size gradient information. In contrast, LORO adopts the Riemannian gradient, which provides the best approximation of the full-size gradient within the tangent space $T_\mathbf{W} \mathcal{M}_r$, thereby preserving the full-size gradient information more effectively than the standard low-rank gradient.

According to Proposition 1 and Proposition 3, the orthogonal projection onto the tangent space is

$$\mathcal{P}(\mathbf{G}) = \mathbf{U}\mathbf{U}^\top \mathbf{G} + \mathbf{G}\mathbf{V}\mathbf{V}^\top - \mathbf{U}\mathbf{U}^\top \mathbf{G}\mathbf{V}\mathbf{V}^\top = \mathbf{P_B}\mathbf{G} + \mathbf{G}\mathbf{P_A} - \mathbf{P_B}\mathbf{G}\mathbf{P_A}, \qquad (17)$$

where $\mathbf{P_B} \triangleq \mathbf{B}(\mathbf{B}^\top \mathbf{B})^{-1}\mathbf{B}^\top$ and $\mathbf{P_A} \triangleq \mathbf{A}^\top(\mathbf{A}\mathbf{A}^\top)^{-1}\mathbf{A}$ are the orthogonal projection matrices onto the column space of $\mathbf{B}$ and the row space of $\mathbf{A}$. We first study how well the full-size gradient can be approximated within the tangent space w.r.t Frobenius norm $\|\cdot\|_F$. For any tangent vector $\mathbf{G}$ resides in the tangent space, the full-size gradient approximation error equals to

$$\|\mathbf{G} - \nabla_\mathbf{W} L\|_F = \|\mathbf{G} - \mathcal{P}(\nabla_\mathbf{W} L)\|_F + \|\mathcal{P}^\perp(\nabla_\mathbf{W} L)\|_F, \qquad (18)$$

where $\mathcal{P}^\perp(\cdot)$ is the orthogonal projection to the complemented subspace of the tangent space. This implies that the full-size gradient approximation error is minimized at $\mathbf{G} = \mathcal{P}(\nabla_\mathbf{W} L)$, which is exactly the Riemannian gradient as shown in Proposition 3. Therefore, the Riemannian gradient represents the steepest gradient descent direction on the manifold, as it provides the best approximation of the full-size gradient within the tangent space.

Now, we can study how well the standard low-rank gradient $\mathbf{Z}$ approximates the full-size gradient. Since $\mathbf{Z}$ is an element in the tangent space (can be proven by showing $\mathcal{P}(\mathbf{Z}) = \mathbf{Z}$), the full-size gradient approximation error of $\mathbf{Z}$ is determined by $\|\mathbf{Z} - \mathcal{P}(\nabla_\mathbf{W} L)\|$, which equals

$$\|\mathbf{B}((\mathbf{B}^\top \mathbf{B})^{-1} - \mathbf{I})\mathbf{B}^\top \nabla_\mathbf{W} L + \nabla_\mathbf{W} L \mathbf{A}^\top((\mathbf{A}\mathbf{A}^\top)^{-1} - \mathbf{I})\mathbf{A} + \mathbf{P_B}\nabla_\mathbf{W} L \mathbf{P_A}\|_F. \qquad (19)$$

This error arises from two main factors: 1. the scale of gradient components, $\mathbf{B}\mathbf{B}^\top \nabla_\mathbf{W} L$ and $\nabla_\mathbf{W} L \mathbf{A}^\top \mathbf{A}$, is not properly normalized to match that of $\mathbf{P_B}\mathbf{G}$ and $\mathbf{G}\mathbf{P_A}$, and 2. the redundant gradient information, $\mathbf{P_B}\nabla_\mathbf{W} L \mathbf{P_A}$—which represents the gradient component in the intersection of the column space of $\mathbf{B}$ and the row space of $\mathbf{A}$—is double-counted across the two sketched gradient terms. Our conclusions align with the findings in (Zhang et al., 2024a), which suggests that down scaling the learning rates of some specific modules (e.g., the low-rank weights) is beneficial.

Therefore, instead of updating $\mathbf{B}$ and $\mathbf{A}$ independently, we should employ LORO to update the low-rank weight $\mathbf{W} = (\mathbf{B}\mathbf{A})$ in the direction of the negative Riemannian gradient, which represents the steepest descent direction on the manifold. This approach ensures that the full-size gradient information is maximally exploited.

**Nearly-zero standard low-rank gradient norm does not indicate a true stationary point.** To understand the performance gap between the standard low-rank optimizer and its full-size counterpart, we study and compare their stationary point conditions. As mentioned in (Absil et al., 2009; Vandereycken, 2013; Olikier et al., 2023), in Riemannian optimization, a nearly-zero Riemannian gradient norm indicates that the parameter is close to a stationary point on the manifold, showing that the training process nearing convergence.

When using the standard low-rank optimizer, however, while the norm of the standard low-rank gradient is nearly zero, the learned parameter can significantly deviate from a true stationary point on the manifold. This is because

$$\|(\nabla_\mathbf{B} L, \nabla_\mathbf{A} L)\|_F^2 = \|\nabla_\mathbf{W} L \mathbf{A}^\top\|_F^2 + \|\mathbf{B}\nabla_\mathbf{W} L\|_F^2 \geqslant (\sigma_r(\mathbf{A})^2 + \sigma_r(\mathbf{B})^2)\|\nabla_\mathbf{W} L\|_F^2, \qquad (20)$$

where $\sigma_r(\cdot)$ denotes the $r$-th largest singular value (i.e. the smallest nonzero singular value). This further implies

$$\|\mathcal{P}(\nabla_\mathbf{W} L)\|_F^2 \leqslant (\sigma_r(\mathbf{A})^2 + \sigma_r(\mathbf{B})^2)^{-1}\|(\nabla_\mathbf{B} L, \nabla_\mathbf{A} L)\|_F^2. \qquad (21)$$

Consequently, when $\mathbf{B}$ and $\mathbf{A}$ are ill-conditioned (i.e., when the condition numbers $\sigma_1(\mathbf{A})/\sigma_r(\mathbf{A})$ and $\sigma_1(\mathbf{B})/\sigma_r(\mathbf{B})$ are large) or nearly singular (i.e., $\sigma_r(\mathbf{A})$ and $\sigma_r(\mathbf{B})$ are small), the Riemannian gradient norm can remain large even if the norm of the standard low-rank gradient is nearly-zero. This indicates the inferior convergence of the standard low-rank optimizer. To mitigate this issue, we adopt the Riemannian gradient to optimize the product of the low-rank factors.

## C.2 Effectiveness of LORO

In this section, we present additional theoretical and empirical evidence showing that LORO can overcome the limitations of standard low-rank training. Furthermore, we show that both the exact and approximate LORO updates are necessary and effective.

In Section C.2.1, our empirical evidence shows that: 1. the exact LORO update is necessary and effective, 2. the approximate LORO update is necessary and effective, 3. LORO benefits from optimizer state refreshing.

In Section C.2.2, our theoretical evidence shows that: 1. the exact LORO update balances the singular values of low-rank factors, and 2. the exact LORO update potentially reduces the full-size gradient approximation error in the successive approximate LORO updates.

Based on our observations and analysis, we argue that the performance improvement stems from the LORO update itself, because: 1. LORO approximates the Riemannian gradient better than the standard low-rank gradient, and it benefits from utilizing more information from the full-size gradient, 2. LORO effectively explores new optimization subspace which reduces the condition number of the low-rank weights, leading to a more stable training process, and 3. LORO minimizes the accumulative error during successive approximate LORO steps.

### C.2.1 Empirical Evidence

**The exact LORO update is necessary and effective.** To validate the effectiveness of the exact LORO update, we conduct a finer ablation study on the exact LORO update frequency. Following Section 5.1, we train rank-256 LLaMA-130m models with $K \in \{10, 50, 100, 250, 500, 650, 750\}$. We visualize the training curve and report the final perplexity.

As shown in Figure 7, as the LORO exact update is getting more frequent, the loss curve is less spiky and the evaluation perplexity exhibits a slight improvement. However, when the LORO exact update is overly lazy, i.e. $K > 650$, the learning curve explodes. Our results show that while the exact LORO update is necessary, it does not need to be applied at every update step. In practice, performing the LORO update every 500 steps achieves satisfactory performance.

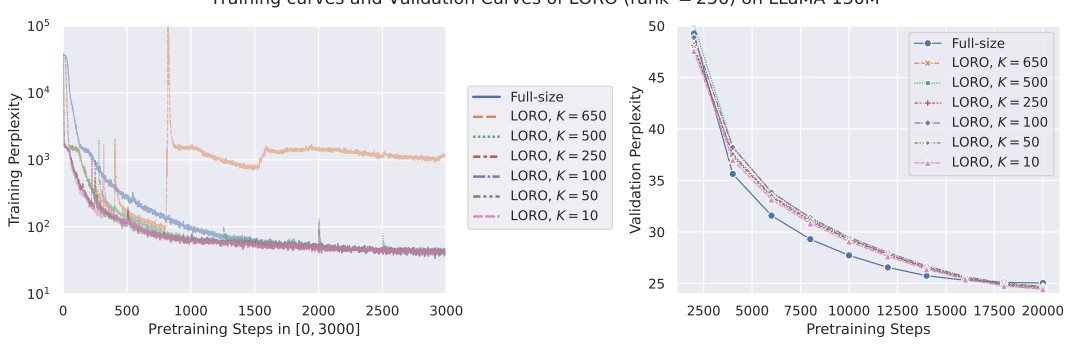

Figure 7: Ablation study on the impact of exact LORO update frequency $K$ on LORO for training rank-256 LLaMA-130M models. The y-axis (log-scale) represents perplexity, with the gray horizontal dashed line indicating the performance of full-size training.

**The approximate LORO update is necessary and effective.** To validate the necessity of using the approximate LORO steps, we train rank-256 LLaMA-130M models with LORO, both with and without learning rate down-scaling between successive exact LORO updates.

As shown in Figure 8, the training curve either fails to converge or yields poor results when the approximate LORO steps are omitted. These results imply that the exact LORO update is necessary and effective. The performance improvement stems from the exact LORO update. In practice, LORO with $K = 500$ can achieve satisfactory training performance without the need to apply the exact LORO update at every step.

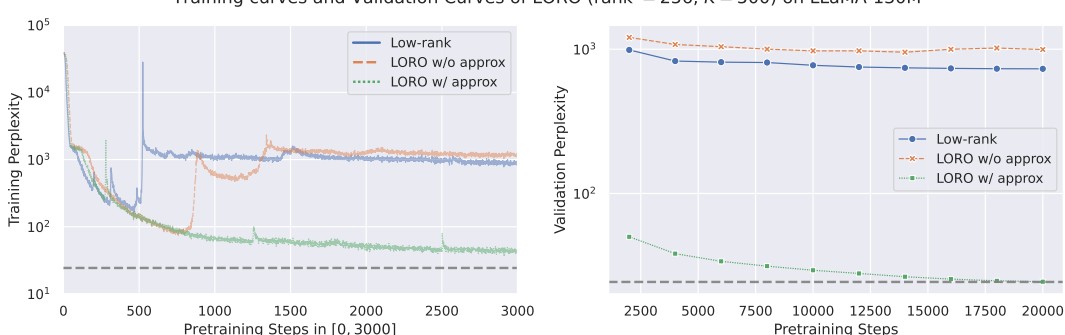

Figure 8: Ablation study on the impact of learning rate down-scaling in approximate LORO updates for training rank-256 LLaMA-130M models. The y-axis (log-scale) represents perplexity, with the gray horizontal dashed line indicating the performance of full-size training.

**LORO benefits from optimizer state refreshing.** To study how optimizer state refreshing affects LORO, we conduct an ablation study on the optimizer state refreshing. Following the setup in Section 5.1, we compare the training curve and validation perplexity of rank-256 LLaMA-130M models trained with LORO, both with and without refreshing the optimizer states periodically. As shown in Figure 9, both configurations lead to satisfactory performance. While refreshing the optimizer states introduces spikes in the early stages of training, it eventually improves the validation loss of the LORO results. This suggests that LORO benefits from optimizer state refreshing.

We conjecture that the stability of LORO against optimizer state refreshing arises because the exact LORO step shifts the low-rank factors $(\mathbf{B}, \mathbf{A})$ to a new optimization subspace with a reduced condition number. This conjecture is partially supported by the analysis in Section C.2.2. In this scenario, the optimizer state retains momentum statistics accumulated many steps ago, originating from an obsolete subspace that significantly deviates from the current one. This indicates that dropping the old momentum statistics can help reduce accumulated error during the approximate LORO steps and encourage exploration in the new subspace.

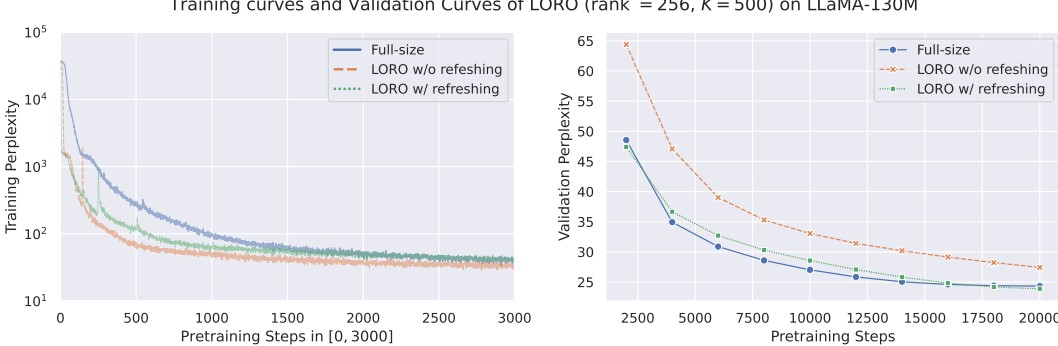

Figure 9: Ablation study on the impact of optimizer state refreshing on LORO updates for training rank-256 LLaMA-130M models. The y-axis (log scale) represents perplexity.

### C.2.2 THEORETICAL EVIDENCE

**Exact LORO update balances the singular values of low-rank factors.** As discussed in Section C.1.2, in contrast to the standard low-rank gradient, LORO adopts Riemannian gradients, which provides the best approximation of the full-size gradient within the tangent space. To better understand the mechanism of the exact LORO update, we need to inspect how the exact LORO update affects the singular values of low-rank weights.

We show that the exact LORO update implicitly balances the singular values of the low-rank factors. According to (10) and Algorithm 1, the exact LORO update ensures $\sigma_i(\mathbf{B}) = \sigma_i(\mathbf{A}) = \sqrt{\sigma_i(\mathbf{W})}$,

for $i = 1, ..., r$. Therefore, the exact LORO update minimize each $\sigma_i(\mathbf{B})^2 + \sigma_i(\mathbf{A})^2$ among all the low-rank factors $(\mathbf{B}, \mathbf{A})$ that satisfy $\sigma_i(\mathbf{B})\sigma_i(\mathbf{A}) = \sigma_i(\mathbf{W})$. This implies LORO prevents $(\sigma_r(\mathbf{A})^2 + \sigma_r(\mathbf{B})^2)$ to be overly small, and it can potentially reduce the condition number of the weights and stabilize the training process. As discussed in Section C.1.2, smaller condition numbers of $\mathbf{B}$ and $\mathbf{A}$ also indicate better convergence property.

**Exact LORO update potentially reduces error in approximate LORO updates.** As discussed in Section C.1.1, the exact LORO plays a subtle yet crucial role in successful low-rank training. To understand how a few exact LORO steps can stabilize the whole training process, we study how an exact LORO step affects the successive approximate LORO steps.

Resuming from the derivations above, we show that the singular value balancing effect in the exact LORO step minimizes an upper bound of the full-size gradient approximation error. Specifically, the full-size gradient approximation error can be bounded by

$$\|\mathbf{B}((\mathbf{B}^\top\mathbf{B})^{-1} - \mathbf{I})\mathbf{B}^\top\nabla_\mathbf{W}L + \nabla_\mathbf{W}L\mathbf{A}^\top((\mathbf{A}\mathbf{A}^\top)^{-1} - \mathbf{I})\mathbf{A} + \mathbf{P}_\mathbf{B}\nabla_\mathbf{W}L\mathbf{P}_\mathbf{A}\|_F + \|\mathcal{P}^\perp(\nabla_\mathbf{W}L)\|_F$$
$$\leqslant (\|\mathbf{I} - \Lambda_\mathbf{B}^2\|_F + \|\mathbf{I} - \Lambda_\mathbf{A}^2\|_F)\|\nabla_\mathbf{W}L\|_2 + 2\|\nabla_\mathbf{W}L\|_F, \tag{22}$$

where $\Lambda_\mathbf{B}, \Lambda_\mathbf{A}$ denote the singular value matrices of $\mathbf{B}$ and $\mathbf{A}$. Suppose $\Lambda_\mathbf{W}$ is the top-$r$ singular values matrix of the low-rank weight $\mathbf{W} = (\mathbf{BA})$, among all the choices of $(\mathbf{B}, \mathbf{A})$ satisfying $\Lambda_\mathbf{B}\Lambda_\mathbf{A} = \Lambda_\mathbf{W}$, the bound $(\|\mathbf{I} - \Lambda_\mathbf{B}^2\|_F + \|\mathbf{I} - \Lambda_\mathbf{A}^2\|_F)$ is minimized when $\Lambda_\mathbf{B} = \Lambda_\mathbf{A} = \Lambda_\mathbf{W}^{1/2}$. This condition is satisfied by the exact LORO steps, implying that the exact LORO update potentially reduces the accumulative full-size gradient approximation error in the successive approximate LORO updates.

In summary, we conjecture that LORO benefits from two key factors: 1. The exact LORO update explores the new subspace by balancing the singular values of the low-rank factors. Thereby, it reduces the condition numbers, leading to smaller accumulative gradient approximation errors in the successive approximate LORO updates. Section C.2.1 provides supportive empirical evidence for this conjecture. 2. The approximated LORO update downscales the standard low-rank gradients, preventing the rapid accumulation of redundant gradient information and avoiding undesired loss spikes before the next exact LORO update.

## D  FURTHER ABLATION STUDIES AND DISCUSSIONS

**Ablation study on the LORO scaling factor.** To study how the learning rate scaler affects the approximate LORO steps, we conducted an additional ablation study on the learning rate scaler scheme of low-rank factors in the approximated LORO steps. Following the same configuration as Section 5.1, we train the rank-128 LLaMA-60M model with LORO, using learning rate scaler choosing from $\{1, 0.5, 0.25, 0.1, 0.01\}$, where $0.25 = r/n = 128/512$ is exactly the Reimannian down-scaling rate we used in (11). We report the final perplexity of the resulting models. As shown in Figure 10, our LORO exhibits robustness across a range of learning rate scalers (i.e. from $0.1$ to $0.5$).

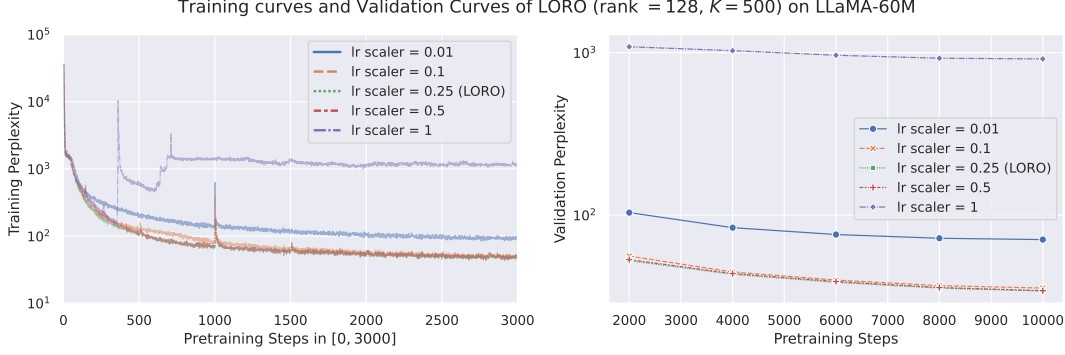

Figure 10: Ablation study on the impact of learning rate scaler on approximate LORO updates for training rank-128 LLaMA-60M models. The y-axis (log scale) represents perplexity.

**Comparing LORO against preconditioned LoRA.** We conduct further experiments to compare our approximate LORO update against the preconditioned LoRA (denoted by "PrecondLoRA") proposed in (Zhang & Pilanci, 2024). We design two experiments to study the following research questions: 1. Is PrecondLoRA, originally proposed for low-rank fine-tuning, sufficient to achieve strong performance in low-rank pretraining? 2. Is the PrecondLoRA update step more ideal than the approximate LORO update (11) in low-rank pretraining?

Specifically, we train rank-128 LLaMA-60M models under the following configurations:

- "PrecondLoRA": Following (Zhang & Pilanci, 2024), both the low-rank factors are updated by: $\mathbf{B} \leftarrow \mathbf{B} - \eta \nabla_{\mathbf{B}} L \cdot (\mathbf{A}\mathbf{A}^\top)^{-1}$ and $\mathbf{A} \leftarrow \mathbf{A} - \eta (\mathbf{B}^\top \mathbf{B})^{-1} \cdot \nabla_{\mathbf{A}} L$.

- "PrecondLoRA+ exact-LORO": We replace the approximate LORO step in Equation (12) with the PrecondLoRA update above, while we keep the exact LORO step in every $K = 500$ steps unchanged.

To ensure a fair comparison with LORO, we use the same learning rate $\eta = 0.01$, the same cosine learning decay schedule, and fix the frequency of optimizer state refreshing as 500.

In Figure 11, we visualize the training curve between PrecondLoRA, "PrecondLoRA+ exact-LORO", and our original LORO. Our observations are summarized as follows:

- LORO > PrecondLoRA: Interestingly, while PrecondLoRA achieves a faster descent in both training and validation loss during the early stages of model training, it eventually exhibits a surge in validation loss. We conjecture that PrecondLoRA bias towards ill-conditioned low-rank factors, making the matrix inversion in the PrecondLoRA update numerically unstable. In low-rank fine-tuning, where the rank $r$ is as small as $r = 8, 16, 32$, the numerical instability in the $r \times r$ matrix inversion is not severe. However, in low-rank pretraining scenarios, where the rank $r$ is as large as $r = 128$ or $256$, the instability of matrix inversion becomes evident and hinders the final performance.

- LORO > "PrecondLoRA + exact-LORO": It is observed that the validation loss of "PrecondLoRA + exact-LORO" exceeds that of LORO during the early training stages but eventually falls behind. We conjecture that performing matrix inversion at every step is overly frequent, which compromises the performance of PrecondLoRA. In contrast, our approximate LORO step demonstrates better performance, without the need to compute the expensive matrix inversion at every update step.

- "PrecondLoRA+ exact-LORO" > PrecondLoRA: It is observed that "PrecondLoRA+ exact-LORO" does not exhibit a surge in validation loss at the end of the training. This implies that our exact LORO update is effective in alleviating the instability in Precond-LoRA training.

In conclusion, for low-rank pretraining, our observation indicates that our approximate/exact LORO update is preferable to PrecondLoRA update which is initially designed for low-rank fine-tuning.

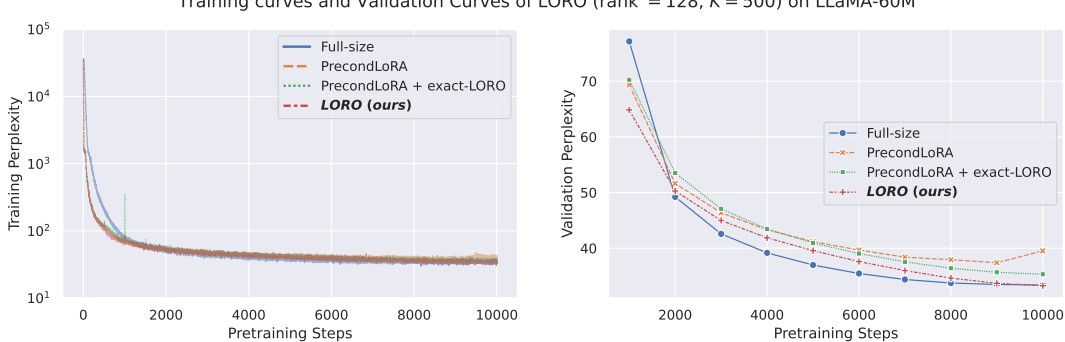

Figure 11: Comparing LORO against preconditioned LoRA (Zhang & Pilanci, 2024) on training rank-128 LLaMA-60M models. The y-axis (log scale) represents perplexity.

**Comparing exact LORO update against standard SVD projection.** To study whether replacing the exact LORO update with a simple low-rank SVD projection is sufficient for good training performance, we conduct further experiments to evaluate the effectiveness of the simple low-rank projection. We train rank-128 LLaMA-60M models, where we replace the exact LORO update with a simple SVD projection every $K$ steps (chosen from $\{500, 100, 10\}$). Specifically, the projection update comprises of $(\mathbf{U}, \mathbf{S}, \mathbf{V}) \leftarrow \text{SVD}_r(\mathbf{BA})$ and $\mathbf{B} \leftarrow \mathbf{U}\sqrt{\mathbf{S}}$, $\mathbf{A} \leftarrow \sqrt{\mathbf{S}}\mathbf{V}^\top$. To ensure a fair comparison with LORO, we use the same learning rate $\eta = 0.01$, the same cosine learning decay schedule, and fix the frequency of optimizer state refreshing as $500$.

As shown in Figure 12, for all $K \in \{10, 100, 500\}$, it is observed that the exact SVD step exhibits a less satisfactory performance. As the exact update gets more frequent ($K$ decreases from 500 to 10), the training curve and validation loss get worse. On the contrary, Figure 7 shows that our LORO enjoys improved training stability and performance under increasing $K$. This suggests that: 1. using simple SVD projection update (retraction without Riemannian update) is insufficient for good training, and 2. using Riemannian update before SVD projection leads to improved performance.

We conjecture that this is because the low-rank SVD projection plays a crucial role in exploring an effective new optimization subspace for the low-rank factors $\mathbf{B}$ and $\mathbf{A}$. Therefore, before performing the low-rank SVD projection to balance the singular values, we need to carefully determine the successive updated version of the low-rank weight ($\mathbf{BA}$). From this perspective, LORO benefits from using the Riemannian gradient, which updates the product ($\mathbf{BA}$) in the direction of the steepest descent on the low-rank manifold. In future work, we will focus on developing more advanced techniques for exploring better optimization subspaces for low-rank pretraining.

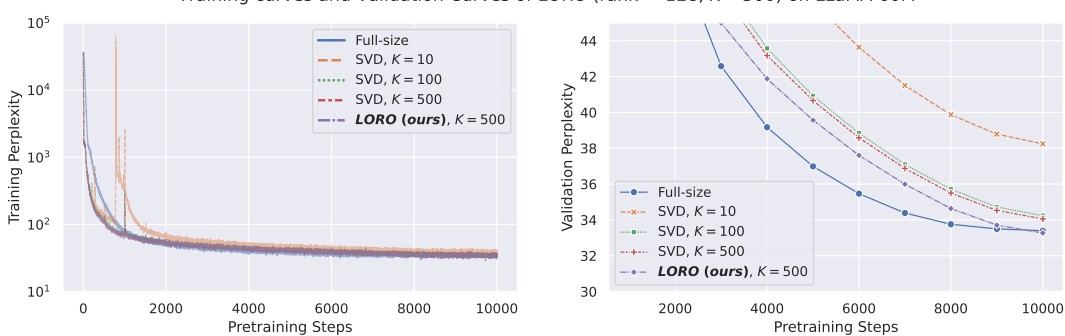

Figure 12: Comparing exact LORO update against simple SVD projection on training rank-128 LLaMA-60M models. The y-axis (log scale) represents perplexity.

**LORO finds competitive low-rank language models with comparable performance to the full-size baselines.** Based on our experimental results and observations, we empirically validate the existence of strong low-rank language models capable of achieving performance comparable to their full-size counterparts. This area remains under-explored in the existing literature. In this paper, we have demonstrated that LORO consistently achieves competitive performance that is comparable to the full-size baselines across various scenarios, indicating that its effectiveness is not merely a coincidence or the result of specific experimental setups or hyperparameter tuning.

Generally, full-size training provides satisfactory stability and strong performance, even with standard optimizers. However, these advantages come at the cost of high memory and computation demand. From an environmentally friendly and economic perspective, our LORO approach achieves a better balance between efficiency and performance.

In pretraining, the fact that a "low-rank language model can be pretrained to achieve high performance comparable to the full-size baseline" is neither entirely unexpected nor surprising. As shown in Figures 2 and Figure 3 of (Jaiswal et al., 2024), full-size pretrained LLaMA models demonstrate low-rank structures in both their weight matrices and gradients. As shown in (Chen et al., 2021) and (Wang et al., 2024), low-rank compressed language models also exhibit competitive performance against the full-size baseline. These observations suggest the feasibility of training low-rank parameterized language models to achieve performance comparable to their full-size counterparts.

In fine-tuning, the fact that "low-rank fine-tuning achieves comparable or better performance than full-parameter fine-tuning" has also been reported in (Hu et al., 2022), (Zhang et al., 2023), and (Zhao et al., 2024). In fine-tuning scenarios where the downstream dataset (e.g., GLUE) is relatively small, full-parameter fine-tuning can lead to overfitting, resulting in suboptimal performance. Moreover, in such cases, a full-size pretrained checkpoint is typically available, allowing new knowledge to be incorporated through a low-rank adaptation step. This makes it feasible to adopt a full-size pretrained model to downstream tasks within a low-rank subspace.

In summary, "low-rank pretraining" and "low-rank fine-tuning" are two distinct settings, with our work focusing on the former. Our empirical results and analysis suggest that the potential of low-rank language models was under-explored in existing literature. In future work, we aim to dedicate more effort to developing advanced low-rank pretraining methods and theoretical tools for analyzing the dynamics of low-rank pretraining.

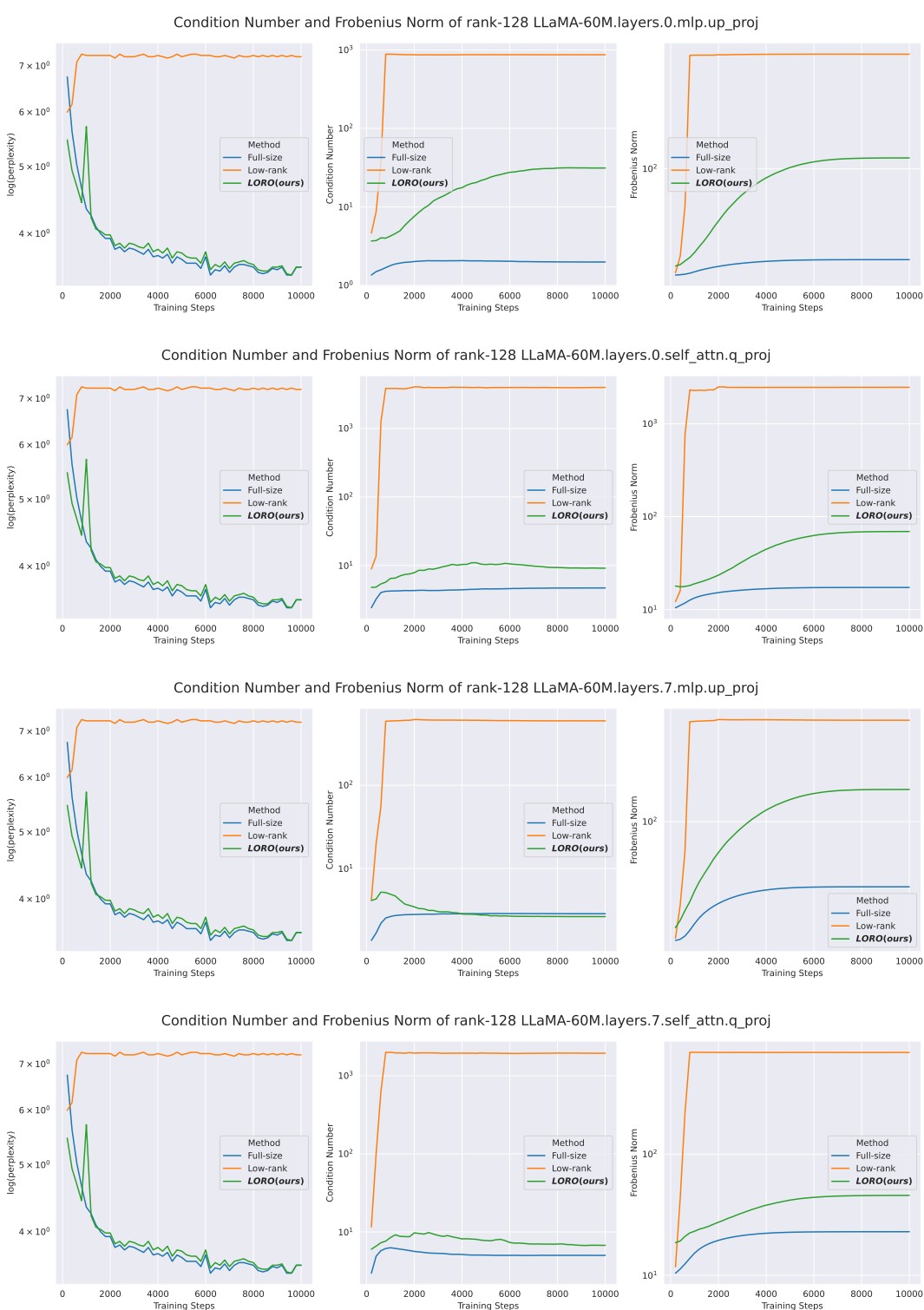

Figure 13: Visualization of the condition number and Frobenius norm for linear and self-attention layers in rank-128 LLaMA-60M model.

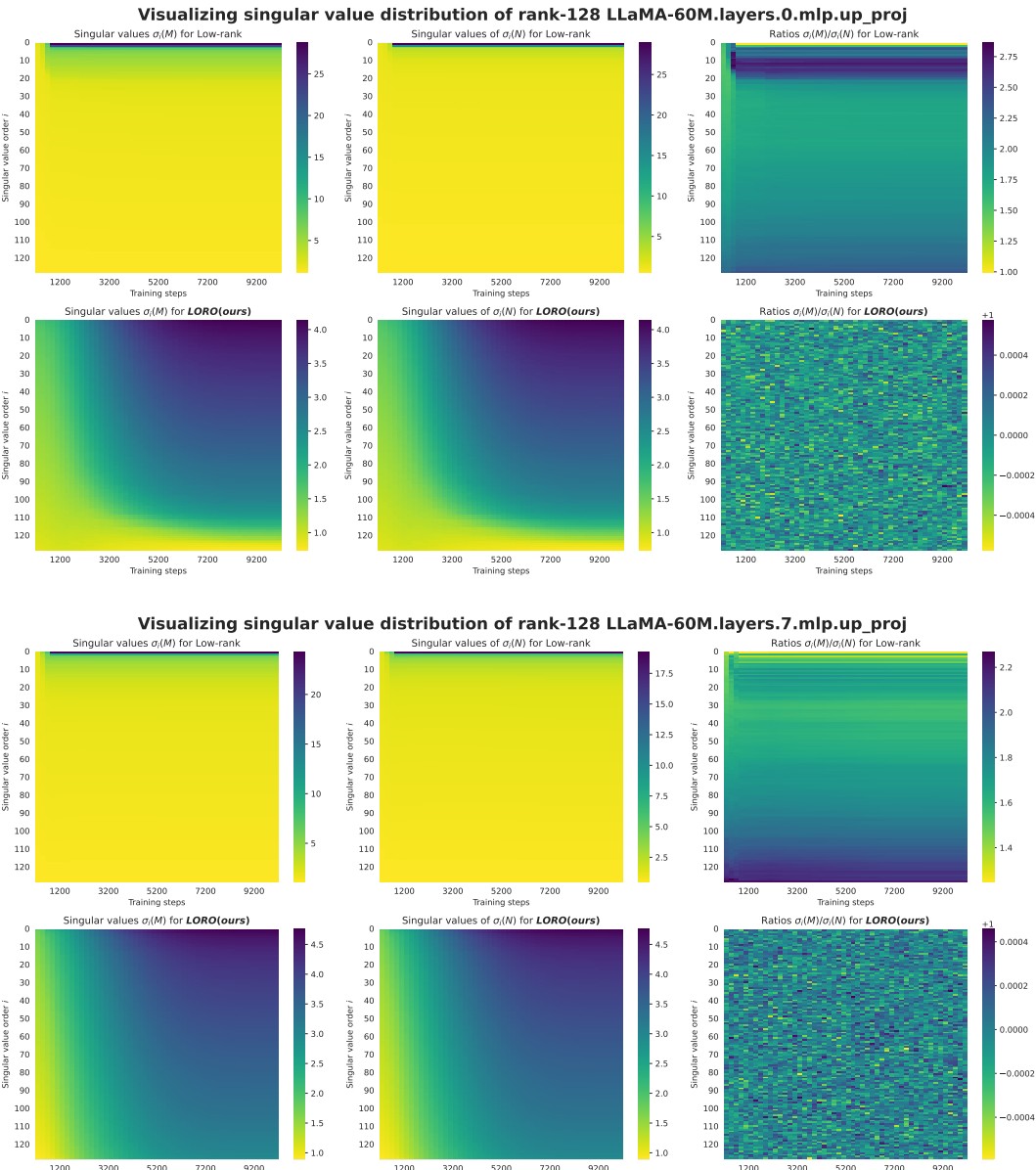

Figure 14: Visualization of the singular value distribution for the linear layers in rank-128 LLaMA-60M model. The y-axis represents the order of singular values, the x-axis represents the training steps. Deeper color indicates a larger singular value is observed.

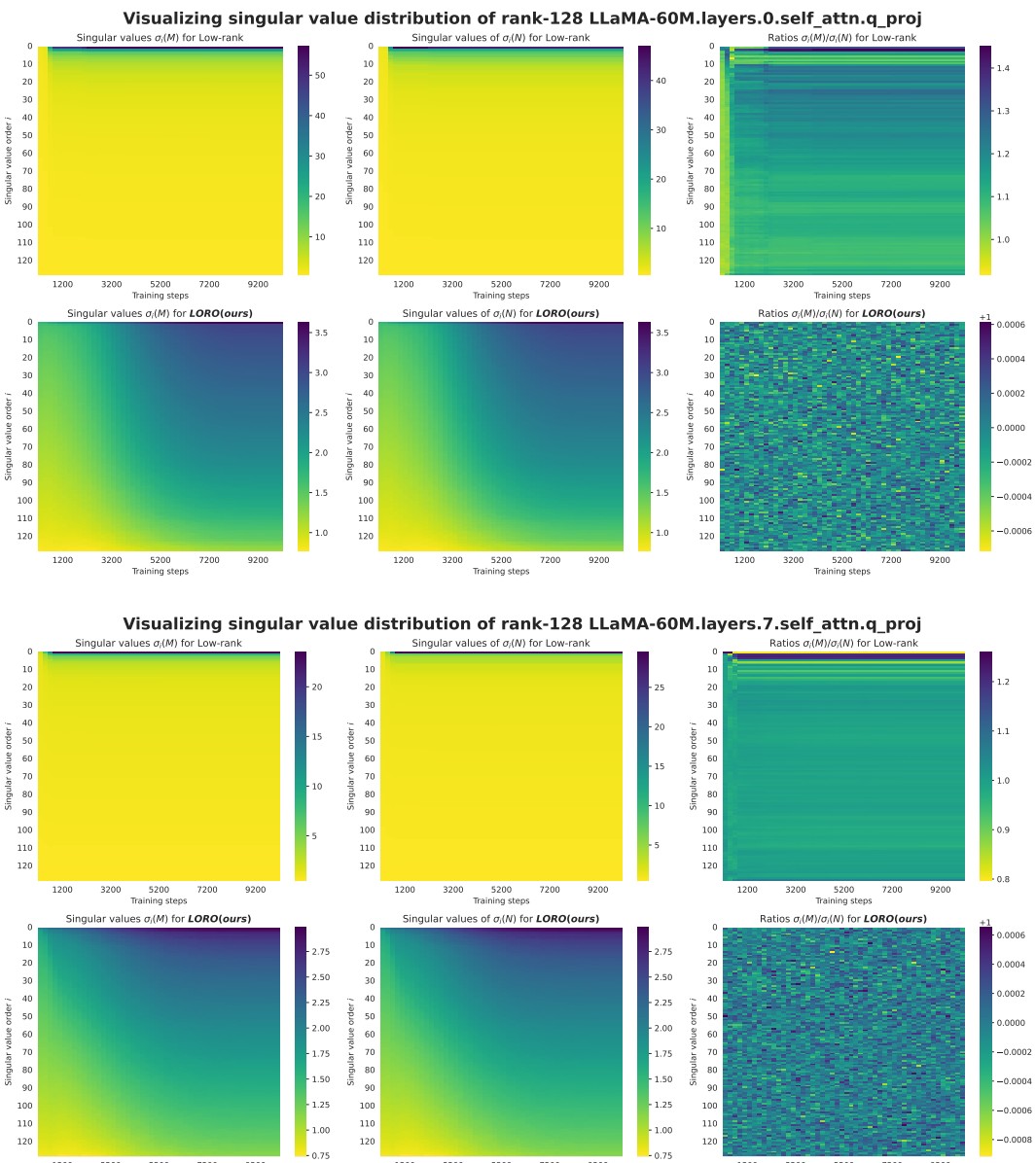

Figure 15: Visualization of the singular value distribution for the self-attention layers in rank-128 LLaMA-60M model. The y-axis represents the order of singular values, the x-axis represents the training steps. Deeper color indicates a larger singular value is observed.

