# OpenReview forum: "Parameter and Memory Efficient Pretraining via Low-rank Riemannian Optimization"
_ICLR.cc/2025/Conference — ICLR 2025 Poster_

### Official Review · Reviewer_9v9P · 2024-10-28

**Soundness:** 3
**Presentation:** 3
**Contribution:** 3
**Rating:** 6
**Confidence:** 3

**Summary:**

The paper proposes to parameterize the weights as a low-rank matrix for parameter and memory efficient pretraining of LLMs. Instead of separately optimizing two factors using (stochastic) gradient descent, the paper leverages Riemannian geometry of low-rank matrices, and propose efficient gradient computation and update strategies. The pretraining performance is evaluated on LlaMA 60M to 1B models.

**Strengths:**

1. The paper proposes a novel Riemannian optimization strategy for optimizing low-rank matrices.

2. The proposed gradient and update strategies only through the factorized form are interesting.

3. The performance is strong compared to the baselines and is comparable to the full-rank pretraining.

**Weaknesses:**

My biggest concern on this work is regarding the empirical simplification using approximated Riemannian update. The paper proposes to use a scaling factor for both the update of B and A. This results in an update which exactly matches the standard update (the same scaling factor can be merged with learning rate). In summary, it appears that for most iterations, Algorithm 1 simply perform standard gradient descent update for the low rank factors and periodically update with Riemannian gradient. This raises the question of whether the periodic Riemannian gradient update results in such a performance improvement.

**Questions:**

1. How did the experiment initialize the B and A?

2. Have you tried different scaling factors for B and A, and whether using different scaling factors can result in different performance?

---

> ### Author Response · Authors · 2024-11-23
> **Response to Reviewer-9v9P-Q1 and Reviewer-9v9P-Q2**
>
> We sincerely appreciate your time and effort in providing constructive and insightful comments. We provide further empirical and theoretical evidence in addressing the concerns raised.
>
> -----
>
> **Q1. What is the initialization scheme of low-rank factors?**
>
> We adopt the standard Xavier initialization for the low-rank factors across all the experiments.
>
>
> **Q2. Ablation study on the LORO scaling factors.**
>
> To study how the learning rate scaler effects the approximate LORO steps, we conducted additional ablation study on the learning rate scaler scheme of low-rank factors in the approximated LORO steps. Following the same configuration as Section 5.1, and we train the rank-$128$ LLaMA-60M model with LORO, using learning rate scaler choosing from $\{1,0.5,0.25,0.1,0.01\}$. We report the final perplexity of the resulting models.
>
> As shown in Figure 10, Appendix D, our LORO exhibit robustness across a range of learning rate scalers (i.e. from $0.1$ to $0.5$).

---

> ### Author Response · Authors · 2024-11-23
> **Response to Reviewer-9v9P-Weakness (summary)**
>
> **Weakness: How does the periodic LORO update achieve improvement.**
>
> Before addressing the concerns, we would like to first clarify that, **the approximate LORO step is not identical to "scaling down the learning rate of vanilla low-rank pretraining"**.
> > - We only scale down the learning rate of the low-rank factors, and we keep the learning rate of the full-size parameters (e.g. `token_emb` and `lm_head`) unchanged. The good performance of LORO aligns with the findings in [1], which suggests that down scaling the learning rates of some specific modules (e.g., the low-rank weights) is beneficial.
> > [1]. Yushun Zhang, et al. Adam-mini: Use Fewer Learning Rates To Gain More. (ICML 2024 workshop)
>
> -----
>
> We understand the importance of providing a rigorous analysis to support our central claim regarding the inefficiency of independent updates of low-rank factors. To address to your concerns, we structure our response from two apsects: Part 1 and Part 2.
> - In **Part 1**: we present additional theoretical and empirical evidence demonstrating that **the standard low-rank optimizer (i.e. independently optimizing low-rank factors ) is inefficient for achieving satisfactory training performance**.
>     - In **Part 1.1**, our empirical evidence demonstrates that:
>         1. The standard low-rank optimizer produces spiky training curves and high perplexity across various initialization schemes and learning rates.
>         2. The standard low-rank optimizer learns low-rank weights with high condition numbers (i.e., the ratio of the largest singular value to the smallest nonzero singular value).
>     - In **Part 1.2**, our theoretical analysis indicates that:
>         1. The standard low-rank gradient in (Equation (3)) fails to preserve adequate information from the full-size gradient within the tangent space.
>         2.  A nearly-stationary point from standard low-rank gradient descent with a near-zero gradient norm may be significantly far away from a true stationary point on the manifold.
>     - Based on these observations and analyses, we conjecture that **the failure of the standard low-rank optimizer stems from the following factors**:
>         1. The standard low-rank gradient contains redundant full-size gradient information, preventing it from accurately approximating the steepest descent direction on the manifold.
>         2. The lack of proper scaling in the standard low-rank gradient causes deviations from the steepest descent direction on the manifold.
>         3. The standard low-rank gradient biases towards ill-conditioned weights, leading to instability during training.
> - In **Part 2**, we present additional theoretical and empirical evidence showing that **LORO is able to overcome the limitations in standard low-rank training. Furthermore, we show that both the exact and approximate LORO updates are necessary and effective**.
>     - In **Part 2.1**, our empirical evidence shows that:
>         1. The exact LORO update is necessary and effective.
>         2. The approximate LORO update is necessary and effective.
>     - In **Part 2.2**, our theoretical evidence shows that:
>         1. The exact LORO update balances the singular values of low-rank factors.
>         2. The exact LORO update potentially reduce the full-size gradient approximation error in the successive approximate LORO updates.
>     - Based on our observations and analysis, we argue that **the performance improvement stems from the LORO update itself**, because:
>         1. LORO approximates the Riemannian gradient better than standard low-rank gradient, and it benefits from utilizing more information from the full-size gradient.
>         2. LORO effectively explore new optimization subspace which reduces the condition number of the low-rank weights, leading to a more stable training process.
>         3. LORO minimizes the accumulative error during successive approximate LORO steps.
>
> The additional theories and experiments are added to Appendix C.

---

> ### Author Response · Authors · 2024-11-23
> **Response to Reviewer-9v9P-Weakness (Part 1.1)**
>
> **Part 1.1. Empirical Evidence**
>
> **Part 1.1.a. Standard low-rank optimizer generally leads to poor performance.**
> To validate that pretraining low-rank language model is generally hard, we train a rank-$128$ LLaMA-60M models with standard low-rank optimizer under different initialization schemes and learning rates. Specifically, we initialize the low-rank factors with Xavier initialization, or standard Gaussian initialization with standard deviation varying in $\{0.1,0.01,0.001\}$, and we tried different learning rates in $\{0.01,0.005,0.001\}$. We report the training curve and the final perplexity of the models.
>
> As shown in Figure 5, Appendix C.1.1, in all settings, the training curve of the vanilla low-rank optimizer is spiky and unstable, leading to unsatisfactory evaluation perplexity. This suggests that independent updates of the low-rank factors are insufficient for good performance across different configurations. Training a low-rank language model to match the full-size baseline is generally challenging.
>
>
> **Part 1.1.b. Standard low-rank optimizer learns ill-conditioned weights.**
> To understand the cause of the instability and spiky training curves in the standard low-rank optimizer, we inspect the condition number and norm of the low-rank factors throughout the training process. Specifically, we train rank-$128$ LLaMA-60M models with Xavier initialization and a learning rate of $0.01$, using different optimizers: the standard low-rank optimizer, the full-size Adam optimizer, and LORO. We visualize the evolution of the singular values, Frobenius norm, and condition numbers of the learned weights at various layers throughout the training process.
>
> As shown in Figure 11, Figure 12, and Figure 13, Appendix C.1.1, the standard low-rank gradient results in model weights with significantly higher condition numbers compared to LORO or full-size Adam. We conjecture that the standard low-rank gradient update introduces imbalances among the singular values of the low-rank factors, thereby impairing the stability of the feature learning process.

---

> ### Author Response · Authors · 2024-11-23
> **Response to Reviewer-9v9P-Weakness (Part 1.2.a)**
>
> **Disclaimer**: The derivations below may be affected by display issues caused by OpenReview's markdown renderer. We encourage readers to refer to **Appendix C of the revised PDF** for further details.
>
> **Part 1.2. Theoretical evidence.**
>
> **Part 1.2.a. Standard low-rank gradient fails to approximate the full-size gradient well.**
>
> To understand the difference between the training dynamic of the standard low-rank optimizer and its full-size counterpart, we examine the gap between the standard low-rank gradient and the full-size gradient. Following Equation (3), we define the standard low-rank gradient as $\mathbf{Z} \triangleq \mathbf{BB}^\top \nabla_{\mathbf{W}}L + \nabla_{\mathbf{W}}L \mathbf{A}^\top\mathbf{A}$, where $\nabla_{\mathbf{W}}L$ is the full-size gradient and $\mathbf{W} = \mathbf{BA}$ is the low-rank weight. We show that $\mathbf{Z}$ fails to adequately preserve the full-size gradient information. In contrast, LORO adopts the Riemannian gradient, which provides the best approximation of the full-size gradient within the tangent space $T_{\mathbf{W}} \mathcal{M}_r$, thereby preserving the full-size gradient information more effectively than the standard low-rank gradient.
>
> According to Proposition 1 and Proposition 3, the orthogonal projection onto the tangent space is given by: $P(G) = U U^T G + G V V^T - U U^T G V V^T= P_B G + G P_A - P_B G P_A$, where $P_B = B (B^T B)^{-1} B^T$ and $P_A = A^T (A A^T)^{-1} A$ are the orthogonal projection matrices onto the column space of $B$ and the row space of $A$, respectively.
>
> Now, we can analyze how well the full-size gradient can be approximated within the tangent space with respect to the Frobenius norm $||.||_F$. For any tangent vector G that resides in the tangent space, the full-size gradient approximation error is expressed as: $||G - ∇_W L||_F = ||G - P(∇_W L)||_F + ||P^\perp(∇_W L)||_F$, where $P^\perp$ is the orthogonal projection to the complement subspace of the tangent space.
>
> This implies that the full-size gradient approximation error is minimized when $G = P(∇_W L)$, which is exactly the Riemannian gradient as stated in Proposition 3. Therefore, the Riemannian gradient represents the steepest gradient descent direction on the manifold because it provides the best approximation of the full-size gradient within the tangent space.
>
> Now, we can analyze how well the standard low-rank gradient $Z$ approximates the full-size gradient. Since $Z$ is an element of the tangent space (this can be proven by showing that $P(Z) = Z$), the full-size gradient approximation error for $Z$ is determined by $||Z - P(∇_W L)||_F$, which is given by: $||B ((B^T B)^{-1} - I) B^T ∇_W L+∇_W L A^T ((A A^T)^{-1} - I) A + P_B ∇_W L P_A||_F$.
>
> This error arises from two main factors:
>
> 1. The scale of the sketched gradient components, $B B^T ∇_W L$ and $∇_W L A^T A$, is not properly normalized to match that of $P_B G$ and $G P_A$.
> 2. The redundant gradient information, $P_B ∇_W L P_A$, which represents the gradient component in the intersection of the column space of $B$ and the row space of $A$, is double-counted across the two sketched gradient terms.
>
> Therefore, instead of updating $B$ and $A$ independently, we should use LORO (Low-Rank Optimization) to update the low-rank weight $W = B A$ in the direction of the negative Riemannian gradient. This gradient represents the steepest descent direction on the manifold and ensures that the full-size gradient information is utilized to the maximum extent.

---

> ### Author Response · Authors · 2024-11-23
> **Response to Reviewer-9v9P-Weakness (Part 1.2.b)**
>
> **Part 1.2.b. Nearly-zero standard low-rank gradient norm does not indicate a true stationary point.**
>
> To understand the performance gap between standard low-rank optimizer and its full-size counterpart, we study compare their stationary point conditions. As mentioned in [1,2,3], in Riemannian optimization, a nearly-zero Riemannian gradient norm indicates that the parameter is close to a stationary point on the manifold, showing that the training process nearing convergence.
>
> When using the standard low-rank optimizer, however, while the norm of the standard low-rank gradient is nearly-zero, the learned parameter can significantly deviate from a true stationary point on the manifold. This is because
> $$
> \begin{align}
>     \|(\nabla_{\mathbf{B}} L, \nabla_{\mathbf{A}}L)\|^2_F = \|\nabla_{\mathbf{W}}L \mathbf{A}^\top\|^2_F+\|\mathbf{B}\nabla_{\mathbf{W}}L\|^2_F\geqslant (\sigma_r(\mathbf{A})^2+\sigma_r(\mathbf{B})^2) \|\nabla_{\mathbf{W}}L\|^2_F,
> \end{align}
> $$ where $\sigma_r(\cdot)$ denotes the $r$-th largest singular value (i.e. the smallest nonzero singular value). This further implies $||P(∇_W L)||_F^2 \leq (\sigma_r(A)^2 + \sigma_r(B)^2)^{-1} ||(∇_B L, ∇_A L)||_F^2$.
>
> Consequently, when $\mathbf{B}$ and $\mathbf{A}$ are ill-conditioned (i.e., when the condition numbers $\sigma_1(\mathbf{A})/\sigma_r(\mathbf{A})$ and $\sigma_1(\mathbf{B})/\sigma_r(\mathbf{B})$ are large) or nearly singular (i.e., $\sigma_r(\mathbf{A})$ and $\sigma_r(\mathbf{B})$ are small), the Riemannian gradient norm can remain large even if the norm of the standard low-rank gradient is nearly-zero. This indicates the inferior convergence of standard low-rank optimizer. To mitigate this issue, we adopt Riemannian gradient to optimize the product of the low-rank factors.
>
>
> [1]. B. Vandereycken, Low-rank matrix completion by riemannian optimization, tech. report, ANCHP-MATHICSE, Mathematics Section, ’Ecole Polytechnique F’ed’erale de Lausanne, 2011.
>
> [2]. P.A. Absil et al. Optimization Algorithms on Matrix Manifolds. Princeton University Press, 2009
>
> [3]. Olikier, Guillaume et al. “Gauss-Southwell type descent methods for low-rank matrix optimization.” ArXiv abs/2306.00897 (2023).

---

> ### Author Response · Authors · 2024-11-23
> **Response to Reviewer-9v9P-Weakness (Part 2 summary)**
>
> **Part 2. The effectiveness of LORO update.**
>
> In this section, we present additional theoretical and empirical evidence showing that LORO is able to overcome the limitations in standard low-rank training. Furthermore, we show that both the exact and approximate LORO updates are necessary and effective.
>
> In **Part 2.1. of our response to Reviewer 9v9P**, our empirical evidence shows that:
> 1. the exact LORO update is necessary and effective.
> 2. the approximate LORO update is necessary and effective.
>
> In **Part 2.2. of our response to Reviewer 9v9P**, our theoretical evidence shows that:
> 1. The exact LORO update balances the singular values of low-rank factors.
> 2. The exact LORO update potentially reduce the full-size gradient approximation error in the successive approximate LORO updates.
>
> Based on our observations and analysis, we argue that the performance improvement stems from the LORO update itself, because:
> 1. LORO approximates the Riemannian gradient better than standard low-rank gradient, and it benefits from utilizing more information from the full-size gradient.
> 2. LORO effectively explore new optimization subspace which reduces the condition number of the low-rank weights, leading to a more stable training process.
> 3. LORO minimizes the accumulative error during successive approximate LORO steps.

---

> ### Author Response · Authors · 2024-11-23
> **Response to Reviewer-9v9P-Weakness (Part 2.1)**
>
> **Part 2.1. Empirical evidence**
>
> **Part 2.1.a. The exact LORO update is necessary and effective.**
>
> To validate the effectiveness of the exact LORO update, we conduct a finer ablation study on the exact LORO update frequency. Following Section 5.2, we train rank-$256$ LLaMA-130m models with $K\in\{10,50,100,250,500,650,750\}$. We visualize the training curve and report the final perplexity.
>
> As shown in Figure 7, Appendix C.2.1, as the LORO exact update is getting more frequent, the loss curve is less spiky and the evaluation perplexity exhibit a slight improvement. However, when the LORO exact update is overly lazy, i.e. $K>650$, the learning curve explodes. Our results show that while the exact LORO update is necessary, it does not need to be applied at every update step. In practice, performing the LORO update every $500$ steps achieves satisfactory performance.
>
>
> **Part 2.1.b. The approximate LORO update is necessary and effective.**
>
> To validate the necessity of using the approximate LORO steps, we train rank-$128$ LLaMA-60M models with LORO, both with and without learning rate downscaling between successive exact LORO updates. Specifically, we tried different learning rates chosen from $\{0.01,0.005,0.001\}$ and we fix the exact LORO update frequency as $K=500$.
>
> As shown in Figure 8, Appendix C.2.1, the training curve either fails to converge or yields poor results when the approximate LORO steps are omitted. This demonstrates that the approximate LORO step is necessary and it serves as an effective proxy for the exact LORO update.

---

> ### Author Response · Authors · 2024-11-23
> **Response to Reviewer-9v9P-Weakness (Part2.2)**
>
> **Part 2.2. Theoretical evidence**
>
> **Disclaimer**: The derivations below may be affected by display issues caused by OpenReview's markdown renderer. We encourage readers to refer to **Appendix C of the revised PDF** for further details.
>
> **Part 2.2.a. LORO update balances the singular values of low-rank factors.**
>
> As discussed in Part 1.2.a of our response to Reviewer-9v9P, in contrast to the standard low-rank gradient, LORO adopts Riemannian gradients, which provides the best approximation of the full-size gradient within the tangent space.
>
> As shown in Equation (10) and Algorithm 1, the exact LORO update ensures
> $\sigma_i(\mathbf{B}) = \sigma_i(\mathbf{A}) = \sqrt{\sigma_i(\mathbf{W})}$, for $i=1,...,r$. Therefore, the exact LORO update minimize each $\sigma_i(\mathbf{B})^2+\sigma_i(\mathbf{A})^2$ among all the low-rank factors $(\mathbf{B},\mathbf{A})$ that satisfy $\sigma_i(\mathbf{B})\sigma_i(\mathbf{A})=\sigma_i(\mathbf{W})$, $i=1,...,r$. This implies LORO prevents $(\sigma_r(\mathbf{A})^2 + \sigma_r(\mathbf{B})^2)$ to be overly small, and it can potentially reduce the condition number of the weights and stabilize the training process. As discussed in Seciton 1.2.b of our response to Reviewer-9v9P, smaller condition numbers of $\mathbf{B}$ and $\mathbf{A}$ also indicate better convergence property.
>
>
> **Part 2.2.b. Exact LORO udpate potentially reduces error in approximate LORO updates.**
>
> As discussed in Part 2.1.a. of our response to Reviewer 9v9P, the exact LORO plays a subtle yet crucial role in successful low-rank training. To understand how a few exact LORO steps can stabilize the whole training process, we study how an exact LORO step effect the successive approximate LORO steps.
>
> Resuming from Part 2.2.a of our response to Reviewer-9v9P, we show that the singular value balancing effect in the exact LORO step minimizes an upper bound of the full-size gradient approximation error. Specifically, the full-size gradient approximation error can be bounded by
>
> $||B  ((B^T  B)^{-1} - I)  B^T  ∇_W L + ∇_W L  A^T  ((A  A^T)^{-1} - I)  A + P_B  ∇_W L  P_A||_F + ||P^\perp(∇_W L)||_F\leq (||I - \Lambda_B^2||_F + ||I - \Lambda_A^2||_F)  ||∇_W L||_2 + 2  ||∇_W L||_F$. where $\Lambda_B$ and $\Lambda_A$ denote the singular value matrices of $B$ and $A$, respectively. Suppose $\Lambda_W$ is the top-$r$ singular values matrix of $W = (BA)$. Among all the choices of $(B, A)$ satisfying $\Lambda_B \Lambda_A = \Lambda_W$, the bound $(||I - \Lambda_B^2||_F + ||I - \Lambda_A^2||_F)$ is minimized when $\Lambda_B = \Lambda_A = \Lambda_W^{1/2}$. This condition is satisfied by the exact LORO steps, implying that the exact LORO update can potentially reduce the accumulative full-size gradient approximation error in successive approximate LORO updates.
>
>
>
> In summary, **we conjecture that LORO benefits from two key factors**:
> 1. The exact LORO update balances the singular values of the low-rank factors, thereby reducing their condition numbers. This leads to smaller accumulative gradient approximation error in the successive approximate LORO updates. The Part 1.1.b. of our response to Reviewer-9v9P provide supportive empirical evidence on this conjecture.
> 2. The approximated LORO update downscales the standard low-rank gradients, preventing the rapid accumulation of redundant gradient information and avoiding undesired loss spikes before the next exact LORO update.

---

> > ### Comment · Reviewer_9v9P · 2024-11-26
> >
> > Thank you for providing the detailed responses and for including additional discussions on the LORO mechanisms.
> >
> > I still have questions on the updates. Is it correct to say that approximate LORO update is equivalent to scaled/preconditioned low-rank updates, which have been studied under the Riemannian preconditioning context for LoRA finetuning [1]. It seems from the experiments that this is not ideal as it more easily experience training spikes? Also whether simply projection to the low-rank manifold for the exact LORO update is sufficient?
> >
> > [1] Zhang, F., & Pilanci, M. (2024). Riemannian Preconditioned LoRA for Fine-Tuning Foundation Models. ICML 2024.

---

> ### Author Response · Authors · 2024-11-26
> **Response to Reviewer-9v9P Additional Questions (Additional Q1)**
>
> Thank you for your thoughtful response and inspiring questions. We hope the following discussions and experiments will help address your concerns. We look forward to receiving your feedback.
>
> **Reviewer-9v9P Additional Q1. Is it correct to say that approximate LORO update is equivalent to scaled/preconditioned low-rank updates in [1]?**
>
>
> **Response to Reviewer-9v9P Additional Q1.** **NO**, it is **NOT** correct to say the approximate LORO update is preconditioned LoRA in [1]. The implementation (and the approximation objective) of the approximate LORO update is **NOT** equivalent to the preconditioned low-rank updates in [1].
>
> --------
>
> For notation simlicity, we denote $\mathbf{P_B}\triangleq \mathbf{B}(\mathbf{B}^\top\mathbf{B})^{-1}\mathbf{B}^\top$ and $\mathbf{P_A}\triangleq \mathbf{A}^\top(\mathbf{A}\mathbf{A}^\top)^{-1}\mathbf{A}$ as the orthogonal projection matrices onto the column space of $\mathbf{B}$ and the row space of $\mathbf{A}$.
>
> According to Equation (1) in [1], the preconditioned low-rank update is given by:
> $\mathbf{BA} \gets (\mathbf{B} - \eta \nabla_{\mathbf{B}}L(\mathbf{A}\mathbf{A}^\top)^{-1})(\mathbf{A} - \eta (\mathbf{B}^\top\mathbf{B})^{-1}\nabla_{\mathbf{A}}L)=\mathbf{BA} - \eta (\nabla_{\mathbf{W}}L \mathbf{P_A} + \mathbf{P_B}\nabla_{\mathbf{W}}L) + O(\eta^2).$
>
> In contrast, as shown in Lines 306-307 and **our Response to Reviewer-9v9P-Weakness (Part 1.2.a)**, the approximation objective of approximate LORO steps is
> $\mathbf{BA} \gets \mathbf{BA} - \eta (\nabla_{\mathbf{W}}L \mathbf{P_A} + \mathbf{P_B}\nabla_{\mathbf{W}}L - \mathbf{P_B}\nabla_{\mathbf{W}} L\mathbf{P_A}).$
>
> The key difference between LORO update and the preconditioned update in [1] is that, our LORO update excludes the term $\mathbf{P_B}\nabla_{\mathbf{W}} L\mathbf{P_A}$ from the update direction. As discussed in **our Response to Reviewer-9v9P-Weakness (Part 1.2.a)**, this term represents the redundant full-size gradient information that lies in the intersection of both the column-space of $\mathbf{B}$ and the row-space of $\mathbf{A}$, and hinders approximating the full-size gradient within the tangent space.
>
> Thus, it is **NOT** correct to say that (the approximation objective) of our approximate LORO update is equivalent to the scaled/preconditioned updates proposed in [1].
>
>
> >[1] Zhang, F., & Pilanci, M. (2024). Riemannian Preconditioned LoRA for Fine-Tuning Foundation Models. ICML 2024.

---

> > ### Author Response · Authors · 2024-11-26
> > **Response to Reviewer-9v9P Additional Questions (Additional Q3)**
> >
> > **Reviewer-9v9P Additional Q3. Whether replacing exact LORO update with simple low-rank projection (SVD) is sufficient?**
> >
> > **Response to Reviewer-9v9P Additional Q3.** To address this concern, we conduct further experiments to evaluate the effectiveness of the simple low-rank projection.
> >
> > Our results shows that:
> > - Using simple SVD projection (retraction w/o Riemannian gradient step) is **NOT** sufficient for good training performance.
> > - **Exact LORO > simple SVD**, using Riemannian update before SVD projection leads to improved performance.
> >
> > -------
> >
> > We train a rank-$128$ LLaMA-60M model, where we replace the exact LORO update with a simple SVD projection every $K$ steps (chosen from $\{500,100,10\}$). Specifically, the projection update comprises of $(\mathbf{U},\mathbf{S},\mathbf{V})\gets \text{SVD}_r(\mathbf{BA})$ and $\mathbf{B}\gets \mathbf{U}\sqrt{\mathbf{S}}$, $\mathbf{A}\gets \sqrt{\mathbf{S}}\mathbf{V}^\top$. To ensure a fair comparison with LORO, we use the same learning rate $\eta=0.01$, the same cosine learning decay schedule, and fix the frequency of optimizer state refreshing as $500$.
> >
> > As shown in **Figure 12, Appendix D of the latest rebuttal revision**, for all $K\in\{10,100,500\}$, it is observed that the exact SVD step exhibit a less satisfactory performance. As the exact update getting more frequently ($K$ decreases from $500$ to $10$), the training curve and validation loss gets worse. On the contrary, **Figure 7, Appendix C.2.1 of the latest rebuttal revision** shows that our LORO enjoys improved training stability and performance under increasing $K$.
> >
> > This suggests that:
> > - Using exact SVD update (projection w/o Riemannian update) is insufficient for good training.
> > - Using Riemannian update before SVD projection leads to improved performance.
> >
> > We conjecture that this is because the low-rank SVD projection plays a crucial role in exploring an effective new optimization subspace for the low-rank factors $\mathbf{B}$ and $\mathbf{A}$. Therefore, before performing the low-rank SVD projection to balance the singular values, we need to carefully determine the successive updated version of the low-rank weight $(\mathbf{BA})$. From this perspective, LORO benefits from using the Riemannian gradient, which updates the product $(\mathbf{BA})$ in the direction of steepest descent on the low-rank manifold.
> >
> > In future work, we will focus on developing more advanced techniques for exploring better optimization subspaces for low-rank pretraining.
> >
> > >[1] Zhang, F., & Pilanci, M. (2024). Riemannian Preconditioned LoRA for Fine-Tuning Foundation Models. ICML 2024.

---

> > > ### Comment · Reviewer_9v9P · 2024-11-26
> > > **Thank you for the response**
> > >
> > > Thank you for the detailed responses. I maintain my score leaning towards acceptance.

---

> > > > ### Author Response · Authors · 2024-11-27
> > > >
> > > > Dear Reviewer 9v9P,
> > > >
> > > > Thank you once again for your time and effort. We sincerely appreciate your valuable comments, which have enhanced the comprehensiveness of our manuscript.
> > > >
> > > > Best regards,
> > > > The Authors

---

> ### Author Response · Authors · 2024-11-26
> **Response to Reviewer-9v9P Additional Questions (Additional Q2)**
>
> **Reviewer-9v9P Additional Q2. Is the preconditioned low-rank updates in [1] more ideal than approximate LORO?**
>
> **Response to Reviewer-9v9P Additional Q2.**
>
> We conduct further experiments to compare our approximate LORO update against the preconditioned low-rank updates in [1].
>
> Our experiment results show that, for low-rank pretraining, our approximate / exact LORO update is more preferrable than `PrecondLoRA` update which are initially designed for low-rank fine-tuning.
>
> -------
>
> We design two experiments to study the following research questions:
> 1. Is `PrecondLoRA`, originally proposed in [1] for low-rank fine-tuning, sufficient to achieve strong performance in low-rank pretraining?
> 2. Is the `PrecondLoRA ` update step more ideal than the approximate LORO update Equation (12) in low-rank pretraining?
>
> Specifically, we train rank-$128$ LLaMA-60M models with learning rate $\eta$ under the follow configurations:
> >1. `PrecondLoRA `: Following [1], both the low-rank factors are updated by: $\mathbf{B} \gets \mathbf{B} - \eta \nabla_{\mathbf{B}}L\cdot (\mathbf{A}\mathbf{A}^\top)^{-1}$ and $\mathbf{A} \gets \mathbf{A} - \eta (\mathbf{B}^\top\mathbf{B})^{-1}\cdot \nabla_{\mathbf{A}}L$ at every step.
> >2. `PrecondLoRA + exact-LORO`: We replace the approximate LORO step in Equation (12) with the `PrecondLoRA` update above, while we keep the exact LORO step in every $K=500$ steps unchanged.
>
> To ensure a fair comparison with LORO, we use the same learning rate $\eta=0.01$, the same cosine learning decay schedule, and fix the frequency of optimizer state refreshing as $500$.
>
> In **Figure 11, Appendix D of the latest rebuttal revision**, we visualize the training curve of between `PrecondLoRA `, `PrecondLoRA + exact-LORO`, and our original `LORO`. Our observations are summarized as follows:
> >1. `LORO` > `PrecondLoRA`: Interestingly, while `PrecondLoRA` achieves a faster descent in both training and validation loss during the early stages of model training, it eventually exhibits a surge in validation loss. We conjecture that, `Precond` bias towards ill-conditioned low-rank factors, making the matrix inversion in the `Precond` update numerically unstable. In low-rank fine-tuning, where the rank $r$ is as small as $r = 8, 16, 32$, the numerical instability in the $r \times r$ matrix inversion is not severe. However, in low-rank pretraining scenarios, where the rank $r$ is as large as $r = 128$ or $256$, the instability of matrix inversion becomes evident and hinders the final performance.
> >2. `LORO` > `PrecondLoRA + exact-LORO`: It is observed that the validation loss of `PrecondLoRA + exact-LORO` exceeds that of LORO during the early training stages but eventually falls behind. We conjecture that performing matrix inversion at every step is overly frequent, which compromises the performance of `Precond`. In contrast, our approximate LORO step demonstrates better performance, without the need of computing the expensive matrix inversion at every update step.
> >3. `PrecondLoRA + exact-LORO` > `PrecondLoRA `: It is observed that `PrecondLoRA + exact-LORO` does not exhibit a surge in validation loss at the end of the training. This implies that our exact LORO update is effective in alleviating the unstability in `PrecondLoRA ` training.
>
> In conclusion, for low-rank pretraining, our observation indicates that our approximate / exact LORO update is more preferrable than `PrecondLoRA ` update which are intially designed for low-rank fine-tuning.
>
> >[1] Zhang, F., & Pilanci, M. (2024). Riemannian Preconditioned LoRA for Fine-Tuning Foundation Models. ICML 2024.

---

### Official Review · Reviewer_37oB · 2024-11-02

**Soundness:** 2
**Presentation:** 3
**Contribution:** 3
**Rating:** 6
**Confidence:** 4

**Summary:**

This paper propose a Riemannian update scheme for low-rank pretraining of large laguage models(LLM). The proposed scheme utilize a Riemannian gradient step (for every $K$ iterations) on the low-rank manifold (represented by SVD) for efficient Riemannian optimization. Numerical experiments are provided to show the efficiency of the proposed approach comparing to the existing low-rank pretraining approaches.

**Strengths:**

The paper is well-organized. The proposed algorithm is theoretically motivatived by Riemannian optimization and does not require much more computational overhead by updating the Riemannian step every $K$ steps. The proposed approach yields good performance in the experiments in the paper.

**Weaknesses:**

(Please respond to the "Questions" section directly)

The effecitveness of the proposed approach is still under-explored both theoretically and numerically; Comparing the low-rank approach, I still have some doubts on the reliability of this work.

**Questions:**

I have following comments and concerns regarding the paper:

1. The effecitveness of the proposed approach is still under-explored: The authors claim that the goal of the proposed approach is to (Line 82) "train a low-rank parameterized language model from scratch to achieve performance comparable to the full-size baseline". According to the scaling law (which might not be 100\% applicable here), the primary indicator of the final model performance should be the model size, and this approach is using **exactly** same number of parameters as the low-rank approach. This makes me really wonder if the low-rank approach is highly underestimated in Figure 1 and Table 1. Maybe it's due to initialization/random seed in the Galore paper. The authors might want to comment on this, especially consider that their update is identical to low-rank except for the Riemannian step in every $K$ steps
2. Continuing above point, since the algorithm only update Riemannian step for every $K$ steps, I would expect to see a clear improvement at the Riemannian step (since other steps are same as low-rank, and low-rank is performing badly), say in Figure 4 (and the curve of low-rank should also be included in Figure 4). However seems that the lose is just decreasing steadily for $K=250$ or $500$. Again, the authors might want to comment on this.
3. One additional experiment I would recommend doing is that, updating the Riemannian step for every iteration or very frequently, and see if it clearly outperforms low-rank/sltrain. And then say that due to the commputational cost, oen could jsut update the Riemannian step for every $K$ iterations.
4. The inference time of the proposed method should also be the same as the low-rank approach?

I think overall this is an interesting work. But it's still hard for me to believe that the every-$K$-steps Riemannian update could be such a miracle to turn the badly-performing low-rank approach to SOTA, considering no additional model parameters introduced. I'm open to discussion and would like to hear the authors' comment on this.

---

> ### Author Response · Authors · 2024-11-23
> **Response to Reviewer 37oB-W1 (summary)**
>
> We sincerely appreciate your time and effort in providing constructive and insightful comments. We provide further empirical and theoretical evidence in addressing the concerns raised.
>
>
> ------
>
>
> **W1. Is the standard low-rank optimizer underestimated?**
>
> Before addressing the concerns, we would like to first clarify that, **the approximate LORO step is not identical to "scaling down the learning rate of vanilla low-rank pretraining"**.
> > - We only scale down the learning rate of the low-rank factors, and we keep the learning rate of the full-size parameters (e.g. `token_emb` and `lm_head`) unchanged. The good performance of LORO aligns with the findings in [1], which suggests that down scaling the learning rates of some specific modules (e.g., the low-rank weights) is beneficial.
> > [1]. Yushun Zhang, et al. Adam-mini: Use Fewer Learning Rates To Gain More. (ICML 2024 workshop)
>
> -----
>
> We then present additional empirical (**in Part 1.1**) and theoretical (**in Part 1.2**) evidence demonstrating that the standard low-rank optimizer (i.e. independently optimizing low-rank factors ) is inefficient for achieving satisfactory training performance.
> - In **Part 1.1**, we present the empirical evidence to demonstrate that:
>     1. The standard low-rank optimizer produces spiky training curves and high perplexity across various initialization schemes and learning rates.
>     2. The standard low-rank optimizer learns low-rank weights with high condition numbers (i.e., the ratio of the largest singular value to the smallest nonzero singular value).
> - In **Part 1.2**, we present the theoretical analysis, which indicates that:
>     1. The standard low-rank gradient in (Equation (3)) fails to preserve adequate information from the full-size gradient within the tangent space.
>     2.  A nearly-stationary point from standard low-rank gradient descent with a near-zero gradient norm may be significantly far away from a true stationary point on the manifold.
>
> Based on these observations and analyses, we conjecture that the failure of the standard low-rank optimizer stems from the following factors:
> 1. The standard low-rank gradient contains redundant full-size gradient information, preventing it from accurately approximating the steepest descent direction on the manifold.
> 2. The lack of proper scaling in the standard low-rank gradient causes deviations from the steepest descent direction on the manifold.
> 3. The standard low-rank gradient biases towards ill-conditioned weights, leading to instability during training.

---

> ### Author Response · Authors · 2024-11-23
> **Response to Reviewer 37oB-W1 (Part 1.1)**
>
> **Part 1.1. Empirical Evidence**
>
>
> **Part 1.1.a. Standard low-rank optimizer generally leads to poor performance.**
> > To validate that pretraining low-rank language model is generally hard, we train a rank-$128$ LLaMA-60M models with standard low-rank optimizer under different initialization schemes and learning rates. Specifically, we initialize the low-rank factors with Xavier initialization, or standard Gaussian initialization with standard deviation varying in $\{0.1,0.01,0.001\}$, and we tried different learning rates in $\{0.01,0.005,0.001\}$. We report the training curve and the final perplexity of the models.
>
> > As shown in Figure 5, Appendix C.1.1, in all settings, the training curve of the vanilla low-rank optimizer is spiky and unstable, leading to unsatisfactory evaluation perplexity. This suggests that independent updates of the low-rank factors are insufficient for good performance across different configurations. Training a low-rank language model to match the full-size baseline is generally challenging.
>
> **Part 1.1.b. Standard low-rank optimizer learns ill-conditioned weights.**
> > To understand the cause of the instability and spiky training curves in the standard low-rank optimizer, we inspect the condition number and norm of the low-rank factors throughout the training process. Specifically, we train rank-$128$ LLaMA-60M models with Xavier initialization and a learning rate of $0.01$, using different optimizers: the standard low-rank optimizer, the full-size Adam optimizer, and LORO. We visualize the evolution of the singular values, Frobenius norm, and condition numbers of the learned weights at various layers throughout the training process.
>
> > As shown in Figure 11, Figure 12, and Figure 13, Appendix C.1.1, the standard low-rank gradient results in model weights with significantly higher condition numbers compared to LORO or full-size Adam. We conjecture that the standard low-rank gradient update introduces imbalances among the singular values of the low-rank factors, thereby impairing the stability of the feature learning process.

---

> ### Author Response · Authors · 2024-11-23
> **Response to Reviewer 37oB-W1 (Part 1.2.a)**
>
> **Part 1.2.: Theoretical evidence.**
>
> **Disclaimer**: The derivations below may be affected by display issues caused by OpenReview's markdown renderer. We encourage readers to refer to **Appendix C of the revised PDF** for further details.
>
> **Part 1.2.a. Standard low-rank gradient fails to approximate the full-size gradient well.**
> > To understand the difference between the training dynamic of the standard low-rank optimizer and its full-size counterpart, we examine the gap between the standard low-rank gradient and the full-size gradient. Following Equation (3), we define the standard low-rank gradient as $\mathbf{Z} \triangleq \mathbf{BB}^\top \nabla_{\mathbf{W}}L + \nabla_{\mathbf{W}}L \mathbf{A}^\top\mathbf{A}$, where $\nabla_{\mathbf{W}}L$ is the full-size gradient and $\mathbf{W} = \mathbf{BA}$ is the low-rank weight. We show that $\mathbf{Z}$ fails to adequately preserve the full-size gradient information. In contrast, LORO adopts the Riemannian gradient, which provides the best approximation of the full-size gradient within the tangent space $T_{\mathbf{W}} \mathcal{M}_r$, thereby preserving the full-size gradient information more effectively than the standard low-rank gradient.
>
> > According to Proposition 1 and Proposition 3, the orthogonal projection onto the tangent space is given by: $P(G) = UU^TG + GVV^T - UU^TGVV^T= P_BG + GP_A - P_BGP_A$, where $P_B = B(B^TB)^{-1}B^T$ and $P_A = A^T(AA^T)^{-1}A$ are the orthogonal projection matrices onto the column space of $B$ and the row space of $A$, respectively.
>
> > Now, we can analyze how well the full-size gradient can be approximated within the tangent space with respect to the Frobenius norm $||.||_F$. For any tangent vector G that resides in the tangent space, the full-size gradient approximation error is expressed as: $||G - ∇_W L||_F = ||G - P(∇_W L)||_F + ||P^\perp(∇_W L)||_F$, where $P^\perp$ is the orthogonal projection to the complement subspace of the tangent space.
>
> > This implies that the full-size gradient approximation error is minimized when $G = P(∇_W L)$, which is exactly the Riemannian gradient as stated in Proposition 3. Therefore, the Riemannian gradient represents the steepest gradient descent direction on the manifold because it provides the best approximation of the full-size gradient within the tangent space.
>
> > Now, we can analyze how well the standard low-rank gradient $Z$ approximates the full-size gradient. Since $Z$ is an element of the tangent space (this can be proven by showing that $P(Z) = Z$), the full-size gradient approximation error for $Z$ is determined by $||Z - P(∇_W L)||_F$, which is given by: $||B((B^TB)^{-1} - I)B^T∇_W L+∇_W LA^T((AA^T)^{-1} - I)A + P_B∇_W LP_A||_F$.
>
> > This error arises from two main factors:
> > 1. The scale of the sketched gradient components, $BB^T∇_W L$ and $∇_W LA^TA$, is not properly normalized to match that of $P_BG$ and $GP_A$.
> > 2. The redundant gradient information, $P_B∇_W LP_A$, which represents the gradient component in the intersection of the column space of $B$ and the row space of $A$, is double-counted across the two sketched gradient terms.
>
> > Therefore, instead of updating $B$ and $A$ independently, we should use LORO (Low-Rank Optimization) to update the low-rank weight $W = BA$ in the direction of the negative Riemannian gradient. This gradient represents the steepest descent direction on the manifold and ensures that the full-size gradient information is utilized to the maximum extent.

---

> ### Author Response · Authors · 2024-11-23
> **Response to Reviewer 37oB-W1 (Part 1.2.b)**
>
> **Part 1.2.b. Nearly-zero standard low-rank gradient norm does not indicate a true stationary point.**
>
>
> **Disclaimer**: The derivations below may be affected by display issues caused by OpenReview's markdown renderer. We encourage readers to refer to **Appendix C of the revised PDF** for further details.
>
>
>
> > To understand the performance gap between standard low-rank optimizer and its full-size counterpart, we study compare their stationary point conditions. As mentioned in [1,2,3], in Riemannian optimization, a nearly-zero Riemannian gradient norm indicates that the parameter is close to a stationary point on the manifold, showing that the training process nearing convergence.
> >
> > When using the standard low-rank optimizer, however, while the norm of the standard low-rank gradient is nearly-zero, the learned parameter can significantly deviate from a true stationary point on the manifold. This is because
> $$
> \begin{align}
>     \|(\nabla_{\mathbf{B}} L, \nabla_{\mathbf{A}}L)\|^2_F = \|\nabla_{\mathbf{W}}L \mathbf{A}^\top\|^2_F+\|\mathbf{B}\nabla_{\mathbf{W}}L\|^2_F\geqslant (\sigma_r(\mathbf{A})^2+\sigma_r(\mathbf{B})^2) \|\nabla_{\mathbf{W}}L\|^2_F,
> \end{align}
> $$ where $\sigma_r(\cdot)$ denotes the $r$-th largest singular value (i.e. the smallest nonzero singular value). This further implies $||P(∇_W L)||_F^2 \leq (\sigma_r(A)^2 + \sigma_r(B)^2)^{-1}  ||(∇_B L, ∇_A L)||_F^2$.
>
> > Consequently, when $\mathbf{B}$ and $\mathbf{A}$ are ill-conditioned (i.e., when the condition numbers $\sigma_1(\mathbf{A})/\sigma_r(\mathbf{A})$ and $\sigma_1(\mathbf{B})/\sigma_r(\mathbf{B})$ are large) or nearly singular (i.e., $\sigma_r(\mathbf{A})$ and $\sigma_r(\mathbf{B})$ are small), the Riemannian gradient norm can remain large even if the norm of the standard low-rank gradient is nearly-zero. This indicates the inferior convergence of standard low-rank optimizer. To mitigate this issue, we adopt Riemannian gradient to optimize the product of the low-rank factors.
>
>
> > [1]. B. Vandereycken, Low-rank matrix completion by riemannian optimization, tech. report, ANCHP-MATHICSE, Mathematics Section, ’Ecole Polytechnique F’ed’erale de Lausanne, 2011.
> > [2]. P.A. Absil et al. Optimization Algorithms on Matrix Manifolds. Princeton University Press, 2009
> > [3]. Olikier, Guillaume et al. “Gauss-Southwell type descent methods for low-rank matrix optimization.” ArXiv abs/2306.00897 (2023).

---

> ### Author Response · Authors · 2024-11-23
> **Response to Reviewer 37oB-W2 (summary)**
>
> **W2. “I would expect to see a clear improvement at the Riemannian step”**
>
> We understand the concern of the reviewer on the **effectiveness of LORO** and we thank Reviewer-37oB for reminding us to include the training curve of the vanilla low-rank training in Figure 4.
>
> To demonstrate that **LORO is able to overcome the limitations in standard low-rank training**, we present additional empirical and theoretical evidences in Part 2.1 and Part 2.2, respectively. Furthermore, we highlight that **both the exact and approximate LORO updates are necessary and effective**.
> - In **Part 2.1**, our empirical evidence shows that:
>     1. the exact LORO update is necessary and effective.
>     2. the approximate LORO update is necessary and effective.
> - In **Part 2.2**, our theoretical evidence shows that:
>     1. The exact LORO update balances the singular values of low-rank factors.
>     2. The exact LORO update potentially reduce the full-size gradient approximation error in the successive approximate LORO updates.
>
> Based on our observations and analysis, we argue that the performance improvement stems from the LORO update itself, because:
> 1. LORO approximates the Riemannian gradient better than standard low-rank gradient, and it benefits from utilizing more information from the full-size gradient.
> 2. LORO effectively explore new optimization subspace which reduces the condition number of the low-rank weights, leading to a more stable training process.
> 3. LORO minimizes the accumulative error during successive approximate LORO steps.

---

> ### Author Response · Authors · 2024-11-23
> **Response to Reviewer 37oB-W2 (Part 2.1)**
>
> **Part 2.1. Empirical evidence**
>
> **Part 2.1.a. The exact LORO update is necessary and effective.**
> > To validate the effectiveness of the exact LORO update, we conduct a finer ablation study on the exact LORO update frequency. Following Section 5.2, we train rank-$256$ LLaMA-130m models with $K\in\{10,50,100,250,500,650,750\}$. We visualize the training curve and report the final perplexity.
>
> > As shown in Figure 7, Appendix C.2.1, as the LORO exact update is getting more frequent, the loss curve is less spiky and the evaluation perplexity exhibit a slight improvement. However, when the LORO exact update is overly lazy, i.e. $K>650$, the learning curve explodes. Our results show that while the exact LORO update is necessary, it does not need to be applied at every update step. In practice, performing the LORO update every $500$ steps achieves satisfactory performance.
>
> **Part 2.1.b. The approximate LORO update is necessary and effective.**
> > To validate the necessity of using the approximate LORO steps, we train rank-$128$ LLaMA-60M models with LORO, both with and without learning rate downscaling between successive exact LORO updates. Specifically, we tried different learning rates chosen from $\{0.01,0.005,0.001\}$ and we fix the exact LORO update frequency as $K=500$.
>
> > As shown in Figure 8, Appendix C.2.1, the training curve either fails to converge or yields poor results when the approximate LORO steps are omitted. This demonstrates that the approximate LORO step is necessary and it serves as an effective proxy for the exact LORO update.

---

> ### Author Response · Authors · 2024-11-23
> **Response to Reviewer 37oB-W2 (Part 2.2)**
>
> **Part 2.2. Theoretical evidence**
>
> **Disclaimer**: The derivations below may be affected by display issues caused by OpenReview's markdown renderer. We encourage readers to refer to **Appendix C of the revised PDF** for further details.
>
> **Part 2.2.a. LORO update balances the singular values of low-rank factors.**
>
> >As discussed in **Part 1.2.a. of our response to Reviewer-37oB**, in contrast to the standard low-rank gradient, LORO adopts Riemannian gradients, which provides the best approximation of the full-size gradient within the tangent space.
>
> >As shown in Equation (10) and Algorithm 1, the exact LORO update ensures
> > $\sigma_i(\mathbf{B}) = \sigma_i(\mathbf{A}) = \sqrt{\sigma_i(\mathbf{W})}$, for $i=1,...,r$. Therefore, the exact LORO update minimize each $\sigma_i(\mathbf{B})^2+\sigma_i(\mathbf{A})^2$ among all the low-rank factors $(\mathbf{B},\mathbf{A})$ that satisfy $\sigma_i(\mathbf{B})\sigma_i(\mathbf{A})=\sigma_i(\mathbf{W})$, $i=1,...,r$. This implies LORO prevents $(\sigma_r(\mathbf{A})^2 + \sigma_r(\mathbf{B})^2)$ to be overly small, and it can potentially reduce the condition number of the weights and stabilize the training process. As discussed in **Part 1.2.b of our response to Reviewer-fYLm**, smaller condition numbers of $\mathbf{B}$ and $\mathbf{A}$ also indicate better convergence property.
>
> **Part 2.2.b. Exact LORO udpate potentially reduces error in approximate LORO updates.**
>
> > As discussed in **Part 2.1.a of our response to Reviewer 37oB**, the exact LORO plays a subtle yet crucial role in successful low-rank training. To understand how a few exact LORO steps can stabilize the whole training process, we study how an exact LORO step effect the successive approximate LORO steps.
>
> > Resuming from **Part 2.2.a of our response to Reviewer 37oB**, we show that the singular value balancing effect in the exact LORO step minimizes an upper bound of the full-size gradient approximation error. Specifically, the full-size gradient approximation error can be bounded by $||B((B^TB)^{-1} - I)B^T∇_W L + ∇_W LA^T((AA^T)^{-1} - I)A + P_B∇_W LP_A||_F + ||P^\perp(∇_W L)||_F\leq (||I - \Lambda_B^2||_F + ||I - \Lambda_A^2||_F)||∇_W L||_2 + 2||∇_W L||_F$, where $\Lambda_B$ and $\Lambda_A$ denote the singular value matrices of $B$ and $A$, respectively. Suppose $\Lambda_W$ is the top-$r$ singular values matrix of $W = (BA)$. Among all the choices of $(B, A)$ satisfying $\Lambda_B \Lambda_A = \Lambda_W$, the bound $(||I - \Lambda_B^2||_F + ||I - \Lambda_A^2||_F)$ is minimized when $\Lambda_B = \Lambda_A = \Lambda_W^{1/2}$. This condition is satisfied by the exact LORO steps, implying that the exact LORO update can potentially reduce the accumulative full-size gradient approximation error in successive approximate LORO updates.
>
> **In summary, we conjecture that LORO benefits from two key factors**:
> 1. The exact LORO update balances the singular values of the low-rank factors, thereby reducing their condition numbers. This leads to smaller accumulative gradient approximation error in the successive approximate LORO updates. The **Part 1.1.b. of our response to Reviewer 37oB** provide supportive empirical evidence on this conjecture.
> 2. The approximated LORO update downscales the standard low-rank gradients, preventing the rapid accumulation of redundant gradient information and avoiding undesired loss spikes before the next exact LORO update.

---

> ### Author Response · Authors · 2024-11-23
> **Response to Reviewer-37oB-W3, Reviewer-37oB-W4**
>
> **W3. Finer ablation on the exact LORO update frequency.**
>
> We refer the readers to **Part 2.1.a of our response to Reviewer 37oB in W2** for the ablation study on the frequency $K$ of exact LORO update.
>
>
> **W4. Is the inference time of standard low-rank equals to that of LORO?**
>
> Yes. The inference time of the resultant models of vanilla low-rank and LORO are the same.

---

> > ### Comment · Reviewer_37oB · 2024-11-25
> >
> > Thanks for the detailed reply.
> >
> > From Fig. 7 of the updated pdf, with a higher LORO update frequency, the LORO algorithm can evetually outperform full-rank training. Fig. 9 also shows that LORO with refreshing outperforms full rank. Also Tab. 6 already shows that LORO outperforms full-rank in quite a number of GLUE tasks. Do you think that this indicates a very strong potential of low-rank training, or that this is just an experimental or hyper-parameter tuning issue and that in general full rank should still be the best-performing one?

---

> ### Author Response · Authors · 2024-11-26
> **Response to Reviewer 37oB: Additional Question 1**
>
> **Response to Reviewer 37oB: Additional Question 1**
>
> Thanks for your feedback and insightful questions.
>
> Based on our experimental results and observations, we empirically validate the existence of strong low-rank language models capable of achieving performance comparable to their full-size counterparts. This area remains underexplored in the existing literature.
>
> We have demonstrated that LORO consistently achieves competitive performance that is comparable to the full-size baselines across various scenarios, indicating that its effectiveness is not merely a coincidence or the result of specific experimental setups or hyperparameter tuning.
>
> Generally, full-size training provides satisfactory stability and strong performance, even with standard optimizers. However, these advantages come at the cost of high memory and computation demand. From an environmentally friendly and economic perspective, our LORO approach achieves a better balance between efficiency and performance.
>
> In pretraining, the fact that a "low-rank language model can be pretrained to achieve high performance comparable to the full-size baseline" is neither entirely unexpected nor surprising. As shown in Figures 2 and Figure 3 of [1], full-size pretrained LLaMA models demonstrate low-rank structures in both their weight matrices and gradients. As shown in [2] and [3], low-rank compressed language models also exhibit competitive performance against the full-size baseline. These observations suggest the feasibility of training low-rank parameterized language models to achieve performance comparable to their full-size counterparts.
>
> In fine-tuning, the fact that "low-rank fine-tuning achieves comparable or better performance than full-parameter fine-tuning" has also been reported in [4], [5], and [6]. In fine-tuning scenarios where the downstream dataset (e.g., GLUE) is relatively small, full-parameter fine-tuning can lead to overfitting, resulting in suboptimal performance. Moreover, in such cases, a full-size pretrained checkpoint is typically available, allowing new knowledge to be incorporated through a low-rank adaptation step. This makes it feasible to adapt a full-size pretrained model to downstream tasks within a low-rank subspace.
>
> In summary, "low-rank pretraining" and "low-rank fine-tuning" are two distinct settings, with our work focusing on the former. Our empirical results and analysis suggest that the potential of low-rank language models was underexplored in existing literatures. In future work, we aim to dedicate more effort to developing advanced low-rank pretraining methods and theoretical tools for analyzing the dynamics of low-rank pretraining.
>
>
>
> >[1]. Ajay Jaiswal, Lu Yin, et al. From GaLore to WeLore: How Low-Rank Weights Non-uniformly Emerge from Low-Rank Gradients. (arXiv:2407.11239)
> >
> >[2]. Patrick Chen, Hsiang-Fu Yu, et al. DRONE: Data-aware Low-rank Compression for Large NLP Models. (NeurlPS 2021)
> >
> >[3]. Xin Wang, Yu Zheng, et al. SVD-LLM: Truncation-aware Singular Value Decomposition for Large Language Model Compression. (arXiv:2403.07378)
> >
> >[4]. Jiawei Zhao, Zhenyu Zhang, et al. Galore: Memory-efficient llm training by gradient low-rank projection. (ICML 2024)
> >
> >[5]. Edward J. Hu, Yelong Shen, et al. LoRA: Low-Rank Adaptation of Large Language Models. (arXiv:2106.09685)
> >
> >[6]. Qingru Zhang, Minshuo Chen, et al. ADALORA: ADAPTIVE BUDGET ALLOCATION FOR
> PARAMETER-EFFICIENT FINE-TUNING. (ICLR 2023)

---

> > ### Comment · Reviewer_37oB · 2024-11-27
> >
> > I sincerely appreciate the detailed reply from the authors. I'd encourage the author to incorporate further discussion into the paper.
> >
> > Two things that I'm still a bit reserved about: 1. I'm still worried if a simple low rank modeling could work for larger models such as 7B or even larger, since with a larger model, low rank may not catch up with the expressibility of the full rank model; 2. I would appreciate if the authors could later provide the codes for experiment settings.
> >
> > Nevertheless, I'd like to increase my score for all the discussions. I think this work has merits that should be included in the venue of ICLR and lean toward accept this work.

---

> > > ### Author Response · Authors · 2024-11-27
> > >
> > > Dear Reviewer 37oB,
> > >
> > > Thank you once again for your constructive suggestions. In response to your concerns, we would like to clarify the following:
> > >
> > > - Due to constraints in time and computational resources, we have deferred the low-rank pretraining experiment on 7B and larger models to future work.
> > > - We plan to release our code and model checkpoints in the final version.
> > >
> > > We sincerely appreciate your time and effort in providing such high-quality reviews.
> > >
> > > Best regards,
> > >
> > > The Authors

---

### Official Review · Reviewer_4NrY · 2024-11-04

**Soundness:** 3
**Presentation:** 3
**Contribution:** 3
**Rating:** 6
**Confidence:** 4

**Summary:**

This paper proposes a novel pretraining method that combines Riemannian optimization with Low-Rank Adaptation (LoRA). The authors aim to enhance the update process of the low-rank factors by integrating Riemannian optimization, which purportedly allows for more effective use of information during parameter updates and reduces redundancy caused by subspace overlap. The paper also introduces a well-designed mechanism to avoid full-size weight and gradient computation, reducing the computational cost during pretraining and resulting in inference-time acceleration for the proposed LORO method.

**Strengths:**

Overall, the paper presents a clear theoretical framework and demonstrates impressive experimental results. However, I remain skeptical about the results.

**Weaknesses:**

1. As part of the motivation, the authors assert that "separately optimizing the low-rank factors
overlooks the intricate structure of the low-rank parameterization and fails to guide their product
toward the steepest loss descent on the low-rank matrix manifold." This statement appears to be
largely intuitive. I believe that this claim requires a clearer analytical breakdown and more
substantial theoretical justifications.

2. The paper mentions that to stabilize training, they apply a 5-step linear learning rate warmup and
refresh the Adam statistics at each exact LORO update step. In prior work, changing the
parameters of the Adam optimizer required careful handling, as restarting the optimizer state
often led to loss spikes. However, the proposed method seems to avoid this issue. It would be
beneficial if the authors could provide a more detailed explanation or proof regarding why LORO
exhibits robustness to this situation, particularly in comparison to previous methods where loss
spikes were problematic.

3. The authors indicate that not every step in training involves a Riemannian update. From the
ablation study provided, it appears that using K= 500 and K= 250 and yields similar performance.
This raises questions about the necessity of the Riemannian update step. Specifically, I am
concerned that the performance improvement may not stem from Riemannian optimization itself,
but rather from the fact that the Riemannian update step refreshes the optimization subspace. It
is possible that standard low-rank steps are already sufficient for good training performance. A
more detailed ablation study isolating the effects of the Riemannian updates would help clarify
their contribution to the overall performance of the method.

4. In the non-Riemannian update steps, the authors adopt the update rule in (10). The authors claim that this is a reasonable approximation of the Riemannian update step. However, intuitively, this appears to simply scale down the learning rate for low-rank pretraining,
which would be equivalent to standard low-rank pretraining. Previous work has shown that vanilla
low-rank pretraining often underperforms, so it is unclear why LORO’s approximate update step
achieves strong empirical results. I would appreciate further clarification or experiments that
explain why this approximation works well in practice.

**Questions:**

See above.

---

> ### Author Response · Authors · 2024-11-23
> **Response to 4NrY-W1**
>
> We sincerely appreciate your time and effort in providing constructive and insightful comments. We detail our response below point by point and add the discussions and additional experiments to Appendix C. Please kindly let us know whether you have any further concerns.
>
>
> **W1. Clarifying the issues of optimizing low-rank factors separately.**
>
> **Disclaimer**: The derivations below may be affected by display issues caused by OpenReview's markdown renderer. We encourage readers to refer to **Appendix C of the revised PDF** for further details.
>
> We have added additional theoretical evidence to support the claim that "separately optimizing the low-rank factors fails to guide their product towards the steepest descent direction."
>
> Specifically, we show that **the standard low-rank gradient (Equation (3)) fails to provide the best approximation of the full-size gradient among the tangent space**. This occurs for the following reasons:
>
> 1. The vanilla low-rank gradient lacks proper normalization.
> 2. It double-counts gradient components that lie in the intersection of the linear spaces spanned by both of the low-rank factors, resulting in redundant gradient information.
>
> >We first examine the gap between the standard low-rank gradient and the full-size gradient. Following (Equation (3)), we define the standard low-rank gradient as $\mathbf{Z} \triangleq \mathbf{BB}^\top \nabla_{\mathbf{W}}L + \nabla_{\mathbf{W}}L \mathbf{A}^\top\mathbf{A}$, where $\nabla_{\mathbf{W}}L$ is the full-size gradient and $\mathbf{W} = \mathbf{BA}$ is the low-rank weight. We show that $\mathbf{Z}$ fails to adequately preserve the full-size gradient information. In contrast, LORO adopts the Riemannian gradient, which provides the best approximation of the full-size gradient within the tangent space $T_{\mathbf{W}} \mathcal{M}_r$, thereby preserving the full-size gradient information more effectively than the standard low-rank gradient.
> >
> >According to Proposition 1 and Proposition 3, the orthogonal projection onto the tangent space is given by: $P(G) = UU^TG + GVV^T - UU^TGVV^T = P_BG + GP_A - P_BGP_A$, where $P_B = B(B^TB)^{-1}B^T$ and $P_A = A^T(A A^T)^{-1}A$ are the orthogonal projection matrices onto the column space of $B$ and the row space of $A$, respectively.
>
> >Now, we can analyze how well the full-size gradient can be approximated within the tangent space with respect to the Frobenius norm $||.||_F$. For any tangent vector G that resides in the tangent space, the full-size gradient approximation error is expressed as: $||G - ∇_W L||_F = ||G - P(∇_W L)||_F + ||P^\perp(∇_W L)||_F$, where $P^\perp$ is the orthogonal projection to the complement subspace of the tangent space.
>
> >This implies that the full-size gradient approximation error is minimized when $G = P(∇_W L)$, which is exactly the Riemannian gradient as stated in Proposition 3. Therefore, the Riemannian gradient represents the steepest gradient descent direction on the manifold because it provides the best approximation of the full-size gradient within the tangent space.
>
> >Now, we can analyze how well the standard low-rank gradient $Z$ approximates the full-size gradient. Since $Z$ is an element of the tangent space (this can be proven by showing that $P(Z) = Z$), the full-size gradient approximation error for $Z$ is determined by $||Z - P(∇_W L)||_F$, which is given by: $||B((B^TB)^{-1} - I)B^T∇_W L+∇_W LA^T((AA^T)^{-1} - I)A + P_B∇_W LP_A||_F$.
>
> >This error arises from two main factors:
> >1. The scale of the sketched gradient components, $BB^T∇_W L$ and $∇_W LA^TA$, is not properly normalized to match that of $P_BG$ and $GP_A$.
> >2. The redundant gradient information, $P_B∇_W LP_A$, which represents the gradient component in the intersection of the column space of $B$ and the row space of $A$, is double-counted across the two sketched gradient terms.
>
> Therefore, instead of updating $B$ and $A$ independently, we should use LORO (Low-Rank Optimization) to update the low-rank weight $W = BA$ in the direction of the negative Riemannian gradient. This gradient represents the steepest descent direction on the manifold and ensures that the full-size gradient information is utilized to the maximum extent.

---

> ### Author Response · Authors · 2024-11-23
> **Response to Reviewer 4NrY-W2 (summary)**
>
> **W2. Why LORO remains stable under optimizer state refreshing?**
>
>
> To address this concern, we add additional discussions on why LORO exhibits robustness against optimizer state refreshing.
>
> - In **Part 2.1**, we provide empirical evidence to show that:
>     1. While optimization state refreshing introduces spikes to the LORO training at the early stage, it eventually improve the LORO's performance.
> - In **Part 2.2**, we present theoretical evidence showing that:
>     1. The exact LORO update balances the singular values of the low-rank factors.
>     2. The exact LORO update explores new optimization subspaces, which potentially reduces the condition numbers of the low-rank factors.
>
> We conjecture that LORO's robustness against optimizer state refreshing stems from the exact LORO step, which moves the low-rank factors $(\mathbf{B}, \mathbf{A})$ to a new optimization subspace with reduced condition numbers. In this scenario, the optimizer state contains momentum statistics that is accumulated many steps ago from obsolete subspaces, which deviate from the current subspace. **As a result, these outdated statistics become misleading and incompatible with the training dynamics of the new subspace.** This indicates that discarding the old momentum statistics can potentially reduce the accumulated error during the approximate LORO steps and encourage exploration in the new subspace.

---

> ### Author Response · Authors · 2024-11-23
> **Response to Reviewer 4NrY-W2 (Part 2.1)**
>
> **Part 2.1. Refreshing optimizer states improves LORO's performance.**
>
> > To study how optimizer state refreshing effects LORO, we conduct ablation study on the optimizer state refreshing. Following the setup in Section 5.2, we compare the training curve and validation perplexity of both rank-$128$ LLaMA-60M and rank-$256$ LLaMA-130M models trained with LORO, both with and without refreshing the optimizer states periodically.
>
> > As shown in Figure 9, Appendix 2.1, both configurations lead to satisfactory performance. While refreshing the optimizer states introduces spikes in the early stages of training, it eventually improves the validation loss of the LORO results. This suggests that LORO benefits from optimizer state refreshing.
>
> > We conjecture that the stability of LORO against optimizer state refreshing arises because the exact LORO step shifts the low-rank factors $(\mathbf{B}, \mathbf{A})$ to a new optimization subspace with a reduced condition number. In **Part 2.2. of our response to Reviewer 4NrY-W2**, we provide further supportive theoretical evidence on this conjecture.
>
> > In this scenario, the optimizer state retains momentum statistics accumulated many steps ago, originating from an obsolete subspace that significantly deviates from the current one. This indicates that dropping the old momentum statistics can help reduce accumulated error during the approximate LORO steps and encourage exploration in the new subspace.

---

> ### Author Response · Authors · 2024-11-23
> **Response to 4NrY-W2 (Part 2.2)**
>
> **Part 2.2. Exact LORO update explores new subspace via singular value balancing, and it potentially learns well-conditioned weights.** To understand why LORO benefits from optimizer state refreshing, we analyze the mechanism of the exact LORO update step.
>
> **Disclaimer**: The derivations below may be affected by display issues caused by OpenReview's markdown renderer. We encourage readers to refer to **Appendix C of the revised PDF** for further details.
>
>
> > We show that the exact LORO update implicitly balances the singular values of the low-rank factors. Accordig to Equation (10) and Algorithm 1, the exact LORO update ensures
> > $\sigma_i(\mathbf{B}) = \sigma_i(\mathbf{A}) = \sqrt{\sigma_i(\mathbf{W})}$, for $i=1,...,r$. Therefore, the exact LORO update minimize each $\sigma_i(\mathbf{B})^2+\sigma_i(\mathbf{A})^2$ among all the low-rank factors $(\mathbf{B},\mathbf{A})$ that satisfy $\sigma_i(\mathbf{B})\sigma_i(\mathbf{A})=\sigma_i(\mathbf{W})$, $i=1,...,r$. This implies LORO prevents $(\sigma_r(\mathbf{A})^2 + \sigma_r(\mathbf{B})^2)$ to be overly small, and it can potentially reduce the condition number of the weights and stabilize the training process. In optimization theory, smaller condition numbers of $\mathbf{B}$ and $\mathbf{A}$ generally indicate better convergence property and a more stable training process.
>
> >Furthermore, we show that the singular value balancing effect in the exact LORO step minimizes an upper bound of the full-size gradient approximation error. In particular, the full-size gradient approximation error can be bounded by: $||B((B^TB)^{-1} - I)B^T∇_W L + ∇_W LA^T((AA^T)^{-1} - I)A + P_B∇_W LP_A||_F + ||P^\perp(∇_W L)||_F \leq (||I - \Lambda_B^2||_F + ||I - \Lambda_A^2||_F)||∇_W L||_2 + 2||∇_W L||_F$, where $\Lambda_B$ and $\Lambda_A$ denote the singular value matrices of $B$ and $A$.
>
> > Suppose $\Lambda_W$ is the top-$r$ singular values matrix of $W = (BA)$, among all the choices of $(B, A)$ satisfying $\Lambda_B \Lambda_A = \Lambda_W$, the bound $(||I - \Lambda_B^2||_F + ||I - \Lambda_A^2||_F)$ is minimized when $\Lambda_B = \Lambda_A = \Lambda_W^{1/2}$. This condition is satisfied by the exact LORO steps, implying that the exact LORO update can potentially reduce the accumulative full-size gradient approximation error in successive approximate LORO updates.
>
>
> In summary, **we conjecture that LORO benefits from two key factors**:
> 1. The exact LORO update explores new subspace by balancing the singular values of the low-rank factors. Thereby, it reduces the condition numbers, leading to smaller accumulative gradient approximation error in the successive approximate LORO updates.
> 2. The approximated LORO update down scales the standard low-rank gradients, preventing the rapid accumulation of redundant gradient information and avoiding undesired loss spikes before the next exact LORO update.
>
>
> To support our conjecture, **we inspect the condition number and norm of the low-rank factors throughout the training process**. Specifically, we train rank-$128$ LLaMA-60M models with Xavier initialization and a learning rate of $0.01$, using different optimizers: the standard low-rank optimizer, the full-size Adam optimizer, and LORO. We visualize the evolution of the singular values, Frobenius norm, and condition numbers of the learned weights at various layers throughout the training process.
>
> > As shown in Figure 11, Figure 12, and Figure 13, Appendix C.1.1, the standard low-rank gradient results in model weights with significantly higher condition numbers compared to LORO or full-size Adam. We conjecture that the standard low-rank gradient update introduces imbalances among the singular values of the low-rank factors, thereby impairing the stability of the feature learning process.
>
> >In contrast, our LORO stabilize training by exploring new optimization subspace which potentially reduces the condition number of the low-rank factors.

---

> ### Author Response · Authors · 2024-11-23
> **Response to Reviewer 4NrY-W3**
>
> **W3. Ablation on optimizer state refreshing and LORO update.**
>
> In this section, we present additional experiments demonstrating that the performance improvement stems from the LORO update itself, rather than from optimizer state refreshing.
>
> 1. Standard low-rank steps fail to achieve stable training, regardless of the learning rate or initialization used.
> 2. Standard low-rank steps combined with optimization state refreshing still fail to achieve stable training.
> 3. The performance improvement is due to the LORO update itself: the exact LORO update is necessary for good training performance.
> 4. Applying the exact LORO update more frequently can slightly improve performance and reduce spikes.
>
> The following results and discussions help clarify the contribution of optimizer state refreshing and exact LORO update to the overall performance of the method.
>
>
> **Part 3.1. Standard low-rank gradient leads to poor performance.**
>
> > To show that standard low-rank method is insufficient for good training performance, we train rank-$128$ LLaMA-60M models with varying initialization schemes and learning rates.
>
> > Following the same configuration as Section 5.1, we initialize the low-rank factors with Xavier initialization, or standard Gaussian initialization with standard deviation varying in $\{0.1,0.01,0.001\}$, and we tried different learning rates in $\{0.01,0.005,0.001\}$. We report the training curve and final perplexity of the models.
>
> > As shown in Figure 5, Appendix C.1.1, the training curve of the vanilla low-rank optimizer is spiky and unstable, resulting in unsatisfactory evaluation perplexity. This indicates that independent updates of the low-rank factors are insufficient for good training performance across different configurations.
>
>
> **Part 3.2. Standard low-rank gradient descent + optimizer state refreshing leads to poor performance.**
> > To assess whether optimizer state refreshing can enhance the standard low-rank optimizer, we perform an ablation study by varying the refreshing frequency in the standard low-rank optimizer. Following the setup in Section 5.2, we trained rank-$128$ LLaMA-60M models using vanilla low-rank gradient descent with different learning rates (chosen from $\{0.01,0.005,0.001\}$) and optimizer state refreshing frequencies (chosen from $\{100, 500, 1000\}$).
>
> > As shown in Figure 6, Appendix C.1.1, all trials exhibit spiky training curves and unsatisfactory performance. This validates that applying optimizer state refreshing to standard low-rank gradient descent still leads to bad training performance. We conjecture that, this is because the vanilla low-rank gradient does not actively explore new optimization subspace, and the training process biases towards low-rank weights with increasing condition numbers. In this case, refreshing the past momentum statistics resided does not help exploring beneficial optimization subspace.
>
>
> **Part 3.3. The exact LORO update is necessary and effective.**
>
> > To further validate that the performance improvement stems from LORO update, we conduct a finer ablation study on the LORO update frequency. Following the setting in Section 5.2, we train rank-$256$ LLaMA-130m models with different LORO exact update frequency, chosen in $K\in\{10,50,100,250,500,650,750\}$. The frequency of optimizer refreshing is fixed to be $500$. We visualize the training curve and report the final perplexity.
>
> > As shown in Figure 7 and Appendix C.2.1, as the LORO exact update is getting more frequent, the loss curve is less spiky and the evaluation perplexity exhibit a slight improvement. However, when the LORO exact update is overly lazy, i.e. $K>650$, the learning curve explodes.
>
> > These results imply that the exact LORO update is necessary and effective. The performance improvement stems from the exact LORO update. In practice, LORO with $K=500$ can achieve satisfactory training performance without the need to apply the exact LORO update at every step.

---

> ### Author Response · Authors · 2024-11-23
> **Response to 4NrY-W4**
>
> **W4. How the approximate LORO steps achieve good performance.**
>
> Before addressing the concerns, we would like to first clarify that, **the approximate LORO step is not identical to "scaling down the learning rate of vanilla low-rank pretraining"**.
> > - We only scale down the learning rate of the low-rank factors, and we keep the learning rate of the full-size parameters (e.g. `token_emb` and `lm_head`) unchanged. The good performance of LORO aligns with the findings in [1], which suggests that down scaling the learning rates of some specific modules (e.g., the low-rank weights) is beneficial.
> > [1]. Yushun Zhang, et al. Adam-mini: Use Fewer Learning Rates To Gain More. (ICML 2024 workshop)
>
> -----
>
> As outlined in Section 4.2 of the manuscript and **our response to Reviewer-4NrY-W1**, the approximate LORO step is primarily designed
> to approximate the scale of the Riemannian gradient by normalizing the standard low-rank gradient terms. Intuitively, this normalization helps control the accumulated Riemannian gradient approximation error $\|\mathbf{Z} - \mathcal{P}(\nabla_{\mathbf{W}}L)\|$,
> thereby mitigating spikes in training performance between successive exact LORO updates.
>
> As outlined in **Part 2.2. of our response to Reivewer-4NrY-W2**, the singular value balancing effect of the exact LORO step potentially alleviate accumulative full-size gradient approximation error in the successive approximate LORO steps. This implicitly enhances the stability of approximate LORO training.
>
> To study whether using the approximate LORO steps is necessary for goo training performance, we train rank-$128$ LLaMA-60M models with LORO, both with and without learning rate downscaling between successive exact LORO updates. Specifically, we tried different learning rates chosen from $\{0.01,0.005,0.001\}$ and we fix the exact LORO update frequency as $K=500$.
>
> As shown in Figure 8, Appendix C.2.1, the training curve either diverges or yields poor results when the approximate LORO steps are omitted. This demonstrates that the approximate LORO step is necessary and it serves as an effective proxy for the exact LORO update.

---

> ### Author Response · Authors · 2024-11-26
>
> Dear Reviewer 4NrY,
>
> We sincerely appreciate the effort you’ve dedicated to providing constructive and insightful comments. We are eager to hear your feedback and to know whether our rebuttal properly addresses your concerns.
>
> Thank you once again for your time and valuable response.
>
> Best,
>
> Authors

---

### Official Review · Reviewer_fYLm · 2024-11-04

**Soundness:** 3
**Presentation:** 4
**Contribution:** 3
**Rating:** 6
**Confidence:** 4

**Summary:**

The paper focuses on memory-efficient pretraining for decoder-only transformer models, particularly in the pretraining setting. It introduces the Low-rank Riemannian Optimizer (LORO), which maintains a low-rank parameterization throughout training and inference. A central challenge in standard low-rank pretraining is that separately optimizing low-rank factors introduces redundant gradient information, which hinders learning; LORO addresses this by using Riemannian optimization to jointly update factors along the low-rank manifold’s steepest descent. Through periodic exact updates and low-cost approximations, LORO reduces memory usage, increases training speed, and achieves lower perplexity than baseline methods for low-rank transformer training.

**Strengths:**

I think the main formulation of the paper, which is to use Riemannian optimization to jointly update factors along the low-rank manifold’s steepest descent, is a novel and interesting approach to the problem of low-rank pretraining. The paper is generally well-written and the math is clear and easy to follow. I particularly liked the method section, where the authors devlop the solution by systematically addressing the memory and computational overheads.

The experimental results in the pre-training setting are strong and show that LORO can achieve competitive performance with reduced memory and computation.

**Weaknesses:**

The main problem I see with the paper is that the authors do not provide a rigorous analysis supporting their central claim that independent updates of the low-rank factors are inefficient. They state that separately optimizing the factors B and A introduces redundant information from the full gradient, which hinders learning, but this claim is not systematically studied or empirically validated. I think this is a major weakness of the paper and should be addressed in a revision. Specifically, it would be good to see some evidence that this phenomenon is common and that it is a significant problem in practice. Additionally, it would be good to also see how LORO is able to overcome this issue and why it is more efficient than other methods in this regard.

Minor: Some other related works that can be cited:

1. Adam-mini: Use Fewer Learning Rates To Gain More, Zhang et al
2. BlockLLM: Memory-Efficient Adaptation of LLMs by Selecting and Optimizing the Right Coordinate Blocks: Ramesh et al

**Questions:**

Refer to the weaknesses section and please share your views on this.

---

> ### Author Response · Authors · 2024-11-23
> **Response to Reviewer fYLm (summary)**
>
> We sincerely appreciate your time and effort in providing constructive and insightful comments. We have cited [1] in Appendix C.1.2. and [2] in line 38, Page 1.
> > [1]. Yushun Zhang, et al. Adam-mini: Use Fewer Learning Rates To Gain More. (ICML 2024 workshop)
> > [2]. Amrutha Varshini Ramesh, Vignesh Ganapathiraman, Issam H. Laradji, and Mark Schmidt. Block-LLM: Memory-efficient adaptation of LLMs by selecting and optimizing the right coordinate blocks. (ICLR 2025 submission)
>
>
> We understand the importance of providing a rigorous analysis to support our central claim regarding the inefficiency of independent updates of low-rank factors. To address to your concerns, we structure our response from two apsects: Part 1 and Part 2.
> - In **Part 1**: we present additional theoretical and empirical evidence demonstrating that **the standard low-rank optimizer (i.e. independently optimizing low-rank factors ) is inefficient for achieving satisfactory training performance**.
>     - In **Part 1.1**, our empirical evidence demonstrates that:
>         1. The standard low-rank optimizer produces spiky training curves and high perplexity across various initialization schemes and learning rates.
>         2. The standard low-rank optimizer learns low-rank weights with high condition numbers (i.e., the ratio of the largest singular value to the smallest nonzero singular value).
>     - In **Part 1.2**, our theoretical analysis indicates that:
>         1. The standard low-rank gradient in (Equation (3)) fails to preserve adequate information from the full-size gradient within the tangent space.
>         2.  A nearly-stationary point from standard low-rank gradient descent with a near-zero gradient norm may be significantly far away from a true stationary point on the manifold.
>     - Based on these observations and analyses, we conjecture that **the failure of the standard low-rank optimizer stems from the following factors**:
>         1. The standard low-rank gradient contains redundant full-size gradient information, preventing it from accurately approximating the steepest descent direction on the manifold.
>         2. The lack of proper scaling in the standard low-rank gradient causes deviations from the steepest descent direction on the manifold.
>         3. The standard low-rank gradient biases towards ill-conditioned weights, leading to instability during training.
> - In **Part 2**, we present additional theoretical and empirical evidence showing that **LORO is able to overcome the limitations in standard low-rank training. Furthermore, we show that both the exact and approximate LORO updates are necessary and effective**.
>     - In **Part 2.1**, our empirical evidence shows that:
>         1. The exact LORO update is necessary and effective.
>         2. The approximate LORO update is necessary and effective.
>     - In **Part 2.2**, our theoretical evidence shows that:
>         1. The exact LORO update balances the singular values of low-rank factors.
>         2. The exact LORO update potentially reduce the full-size gradient approximation error in the successive approximate LORO updates.
>     - Based on our observations and analysis, we argue that **the performance improvement stems from the LORO update itself**, because:
>         1. LORO approximates the Riemannian gradient better than standard low-rank gradient, and it benefits from utilizing more information from the full-size gradient.
>         2. LORO effectively explore new optimization subspace which reduces the condition number of the low-rank weights, leading to a more stable training process.
>         3. LORO minimizes the accumulative error during successive approximate LORO steps.
>
> The additional theories and experiments are added to Appendix C.

---

> > ### Author Response · Authors · 2024-11-23
> > **Response to Reviewer fYLm (Part 2, summary)**
> >
> > **Part 2. The effectiveness of LORO update.**
> >
> > In this section, we present additional theoretical and empirical evidence showing that LORO is able to overcome the limitations in standard low-rank training. Furthermore, we show that both the exact and approximate LORO updates are necessary and effective.
> >
> > In **Part 2.1. of our response to Reviewer fYLm**, our empirical evidence shows that:
> > 1. the exact LORO update is necessary and effective.
> > 2. the approximate LORO update is necessary and effective.
> >
> > In **Part 2.2. of our response to Reviewer fYLm**, our theoretical evidence shows that:
> > 1. The exact LORO update balances the singular values of low-rank factors.
> > 2. The exact LORO update potentially reduce the full-size gradient approximation error in the successive approximate LORO updates.
> >
> > Based on our observations and analysis, we argue that the performance improvement stems from the LORO update itself, because:
> > 1. LORO approximates the Riemannian gradient better than standard low-rank gradient, and it benefits from utilizing more information from the full-size gradient.
> > 2. LORO effectively explore new optimization subspace which reduces the condition number of the low-rank weights, leading to a more stable training process.
> > 3. LORO minimizes the accumulative error during successive approximate LORO steps.

---

> ### Author Response · Authors · 2024-11-23
> **Response to Reviewer fYLm (Part 1.1)**
>
> **Part 1.1. Empirical Evidence**
>
> **Part 1.1.a. Standard low-rank optimizer generally leads to poor performance.**
> To validate that pretraining low-rank language model is generally hard, we train a rank-$128$ LLaMA-60M models with standard low-rank optimizer under different initialization schemes and learning rates. Specifically, we initialize the low-rank factors with Xavier initialization, or standard Gaussian initialization with standard deviation varying in $\{0.1,0.01,0.001\}$, and we tried different learning rates in $\{0.01,0.005,0.001\}$. We report the training curve and the final perplexity of the models.
>
> As shown in Figure 5, Appendix C.1.1, in all settings, the training curve of the vanilla low-rank optimizer is spiky and unstable, leading to unsatisfactory evaluation perplexity. This suggests that independent updates of the low-rank factors are insufficient for good performance across different configurations. Training a low-rank language model to match the full-size baseline is generally challenging.
>
>
> **Part 1.1.b. Standard low-rank optimizer learns ill-conditioned weights.**
> To understand the cause of the instability and spiky training curves in the standard low-rank optimizer, we inspect the condition number and norm of the low-rank factors throughout the training process. Specifically, we train rank-$128$ LLaMA-60M models with Xavier initialization and a learning rate of $0.01$, using different optimizers: the standard low-rank optimizer, the full-size Adam optimizer, and LORO. We visualize the evolution of the singular values, Frobenius norm, and condition numbers of the learned weights at various layers throughout the training process.
>
> As shown in Figure 11, Figure 12, and Figure 13, Appendix C.1.1, the standard low-rank gradient results in model weights with significantly higher condition numbers compared to LORO or full-size Adam. We conjecture that the standard low-rank gradient update introduces imbalances among the singular values of the low-rank factors, thereby impairing the stability of the feature learning process.

---

> ### Author Response · Authors · 2024-11-23
> **Response to Reviewer fYLm (Part 1.2.a)**
>
> **Disclaimer**: The derivations below may be affected by display issues caused by OpenReview's markdown renderer. We encourage readers to refer to **Appendix C of the revised PDF** for further details.
>
> **Part 1.2. Theoretical evidence.**
>
> **Part 1.2.a. Standard low-rank gradient fails to approximate the full-size gradient well.**
>
> To understand the difference between the training dynamic of the standard low-rank optimizer and its full-size counterpart, we examine the gap between the standard low-rank gradient and the full-size gradient. Following Equation (3), we define the standard low-rank gradient as $\mathbf{Z} \triangleq \mathbf{BB}^\top \nabla_{\mathbf{W}}L + \nabla_{\mathbf{W}}L \mathbf{A}^\top\mathbf{A}$, where $\nabla_{\mathbf{W}}L$ is the full-size gradient and $\mathbf{W} = \mathbf{BA}$ is the low-rank weight. We show that $\mathbf{Z}$ fails to adequately preserve the full-size gradient information. In contrast, LORO adopts the Riemannian gradient, which provides the best approximation of the full-size gradient within the tangent space $T_{\mathbf{W}} \mathcal{M}_r$, thereby preserving the full-size gradient information more effectively than the standard low-rank gradient.
>
>
> According to Proposition 1 and Proposition 3, the orthogonal projection onto the tangent space is given by:
>
> $P(G) = U U^T G + G V V^T - U U^T  G  V  V^T = P_B  G + G  P_A - P_B G P_A$,
>
> where $P_B = B  (B^T  B)^{-1}  B^T$ and $P_A = A^T  (A  A^T)^{-1}  A$ are the orthogonal projection matrices onto the column space of $B$ and the row space of $A$, respectively.
>
> Now, we can analyze how well the full-size gradient can be approximated within the tangent space with respect to the Frobenius norm $||.||_F$. For any tangent vector G that resides in the tangent space, the full-size gradient approximation error is expressed as:
>
> $||G - ∇_W L||_F = ||G - P(∇_W L)||_F + ||P^\perp(∇_W L)||_F$,
>
> where $P^\perp$ is the orthogonal projection to the complement subspace of the tangent space.
>
> This implies that the full-size gradient approximation error is minimized when $G = P(∇_W L)$, which is exactly the Riemannian gradient as stated in Proposition 3. Therefore, the Riemannian gradient represents the steepest gradient descent direction on the manifold because it provides the best approximation of the full-size gradient within the tangent space.
>
> Now, we can analyze how well the standard low-rank gradient $Z$ approximates the full-size gradient. Since $Z$ is an element of the tangent space (this can be proven by showing that $P(Z) = Z$), the full-size gradient approximation error for $Z$ is determined by $||Z - P(∇_W L)||_F$, which is given by: $||B ((B^T B)^{-1} - I)  B^T  ∇_W L+∇_W L A^T  ((A A^T)^{-1} - I) A + P_B  ∇_W L P_A||_F$.
>
>
> This error arises from two main factors:
>
> 1. The scale of the sketched gradient components, $B  B^T  ∇_W L$ and $∇_W L  A^T  A$, is not properly normalized to match that of $P_B  G$ and $G P_A$.
> 2. The redundant gradient information, $P_B  ∇_W L P_A$, which represents the gradient component in the intersection of the column space of $B$ and the row space of $A$, is double-counted across the two sketched gradient terms.
>
> Therefore, instead of updating $B$ and $A$ independently, we should use LORO (Low-Rank Optimization) to update the low-rank weight $W = B A$ in the direction of the negative Riemannian gradient. This gradient represents the steepest descent direction on the manifold and ensures that the full-size gradient information is utilized to the maximum extent.

---

> ### Author Response · Authors · 2024-11-23
> **Response to Reviewer fYLm (Part 1.2.b)**
>
> **Part 1.2.b. Nearly-zero standard low-rank gradient norm does not indicate a true stationary point.**
>
> **Disclaimer**: The derivations below may be affected by display issues caused by OpenReview's markdown renderer. We encourage readers to refer to **Appendix C of the revised PDF** for further details.
>
> To understand the performance gap between standard low-rank optimizer and its full-size counterpart, we study compare their stationary point conditions. As mentioned in [1,2,3], in Riemannian optimization, a nearly-zero Riemannian gradient norm indicates that the parameter is close to a stationary point on the manifold, showing that the training process nearing convergence.
>
> When using the standard low-rank optimizer, however, while the norm of the standard low-rank gradient is nearly-zero, the learned parameter can significantly deviate from a true stationary point on the manifold. This is because
> $$
> \begin{align}
>     \|(\nabla_{\mathbf{B}} L, \nabla_{\mathbf{A}}L)\|^2_F = \|\nabla_{\mathbf{W}}L \mathbf{A}^\top\|^2_F+\|\mathbf{B}\nabla_{\mathbf{W}}L\|^2_F\geqslant (\sigma_r(\mathbf{A})^2+\sigma_r(\mathbf{B})^2) \|\nabla_{\mathbf{W}}L\|^2_F,
> \end{align}
> $$ where $\sigma_r(\cdot)$ denotes the $r$-th largest singular value (i.e. the smallest nonzero singular value). This further implies $||P(∇_W L)||_F^2 \leq (\sigma_r(A)^2 + \sigma_r(B)^2)^{-1}  ||(∇_B L, ∇_A L)||_F^2$.
>
> Consequently, when $\mathbf{B}$ and $\mathbf{A}$ are ill-conditioned (i.e., when the condition numbers $\sigma_1(\mathbf{A})/\sigma_r(\mathbf{A})$ and $\sigma_1(\mathbf{B})/\sigma_r(\mathbf{B})$ are large) or nearly singular (i.e., $\sigma_r(\mathbf{A})$ and $\sigma_r(\mathbf{B})$ are small), the Riemannian gradient norm can remain large even if the norm of the standard low-rank gradient is nearly-zero. This indicates the inferior convergence of standard low-rank optimizer. To mitigate this issue, we adopt Riemannian gradient to optimize the product of the low-rank factors.
>
>
> [1]. B. Vandereycken, Low-rank matrix completion by riemannian optimization, tech. report, ANCHP-MATHICSE, Mathematics Section, ’Ecole Polytechnique F’ed’erale de Lausanne, 2011.
>
> [2]. P.A. Absil et al. Optimization Algorithms on Matrix Manifolds. Princeton University Press, 2009
>
> [3]. Olikier, Guillaume et al. “Gauss-Southwell type descent methods for low-rank matrix optimization.” ArXiv abs/2306.00897 (2023).

---

> ### Author Response · Authors · 2024-11-23
> **Response to Reviewer fYLm (Part 2.1)**
>
> **Part 2.1. Empirical evidence**
>
> **Part 2.1.a. The exact LORO update is necessary and effective.**
>
> To validate the effectiveness of the exact LORO update, we conduct a finer ablation study on the exact LORO update frequency. Following Section 5.2, we train rank-$256$ LLaMA-130m models with $K\in\{10,50,100,250,500,650,750\}$. We visualize the training curve and report the final perplexity.
>
> As shown in Figure 7, Appendix C.2.1, as the LORO exact update is getting more frequent, the loss curve is less spiky and the evaluation perplexity exhibit a slight improvement. However, when the LORO exact update is overly lazy, i.e. $K>650$, the learning curve explodes. Our results show that while the exact LORO update is necessary, it does not need to be applied at every update step. In practice, performing the LORO update every $500$ steps achieves satisfactory performance.
>
>
> **Part 2.1.b. The approximate LORO update is necessary and effective.**
>
> To validate the necessity of using the approximate LORO steps, we train rank-$128$ LLaMA-60M models with LORO, both with and without learning rate downscaling between successive exact LORO updates. Specifically, we tried different learning rates chosen from $\{0.01,0.005,0.001\}$ and we fix the exact LORO update frequency as $K=500$.
>
> As shown in Figure 8, Appendix C.2.1, the training curve either fails to converge or yields poor results when the approximate LORO steps are omitted. This demonstrates that the approximate LORO step is necessary and it serves as an effective proxy for the exact LORO update.

---

> ### Author Response · Authors · 2024-11-23
> **Response to Reviewer fYLm (Part 2.2)**
>
> **Part 2.2. Theoretical evidence**
>
> **Disclaimer**: The derivations below may be affected by display issues caused by OpenReview's markdown renderer. We encourage readers to refer to **Appendix C of the revised PDF** for further details.
>
> **Part 2.2.a. LORO update balances the singular values of low-rank factors.**
>
> As discussed in **Part 1.2.a of our response to Reviewer-fYLm**, in contrast to the standard low-rank gradient, LORO adopts Riemannian gradients, which provides the best approximation of the full-size gradient within the tangent space.
>
> As shown in Equation (10) and Algorithm 1, the exact LORO update ensures
> $\sigma_i(\mathbf{B}) = \sigma_i(\mathbf{A}) = \sqrt{\sigma_i(\mathbf{W})}$, for $i=1,...,r$. Therefore, the exact LORO update minimize each $\sigma_i(\mathbf{B})^2+\sigma_i(\mathbf{A})^2$ among all the low-rank factors $(\mathbf{B},\mathbf{A})$ that satisfy $\sigma_i(\mathbf{B})\sigma_i(\mathbf{A})=\sigma_i(\mathbf{W})$, $i=1,...,r$. This implies LORO prevents $(\sigma_r(\mathbf{A})^2 + \sigma_r(\mathbf{B})^2)$ to be overly small, and it can potentially reduce the condition number of the weights and stabilize the training process. As discussed in **Seciton 1.2.b of our response to Reviewer-fYLm**, smaller condition numbers of $\mathbf{B}$ and $\mathbf{A}$ also indicate better convergence property.
>
>
> **Part 2.2.b. Exact LORO udpate potentially reduces error in approximate LORO updates.**
>
> As discussed in **Part 2.1.a. of our response to Reviewer fYLm**, the exact LORO plays a subtle yet crucial role in successful low-rank training. To understand how a few exact LORO steps can stabilize the whole training process, we study how an exact LORO step effect the successive approximate LORO steps.
>
> Resuming from **Part 2.2.a of our response to Reviewer-fYLm**, we show that the singular value balancing effect in the exact LORO step minimizes an upper bound of the full-size gradient approximation error. Specifically, the full-size gradient approximation error can be bounded by
>
> $||B((B^TB)^{-1} - I)B^T∇_W L + ∇_W LA^T((AA^T)^{-1} - I)A + P_B∇_W LP_A||_F + ||P^\perp(∇_W L)||_F\leq (||I - \Lambda_B^2||_F + ||I - \Lambda_A^2||_F)||∇_W L||_2 + 2||∇_W L||_F$.
>
> where $\Lambda_B$ and $\Lambda_A$ denote the singular value matrices of $B$ and $A$, respectively. Suppose $\Lambda_W$ is the top-$r$ singular values matrix of $W = (BA)$. Among all the choices of $(B, A)$ satisfying $\Lambda_B \Lambda_A = \Lambda_W$, the bound $(||I - \Lambda_B^2||_F + ||I - \Lambda_A^2||_F)$ is minimized when $\Lambda_B = \Lambda_A = \Lambda_W^{1/2}$. This condition is satisfied by the exact LORO steps, implying that the exact LORO update can potentially reduce the accumulative full-size gradient approximation error in successive approximate LORO updates.
>
>
>
> In summary, **we conjecture that LORO benefits from two key factors**:
> 1. The exact LORO update balances the singular values of the low-rank factors, thereby reducing their condition numbers. This leads to smaller accumulative gradient approximation error in the successive approximate LORO updates. The **Part 1.1.b. of our response to Reviewer-fYLm** provide supportive empirical evidence on this conjecture.
> 2. The approximated LORO update downscales the standard low-rank gradients, preventing the rapid accumulation of redundant gradient information and avoiding undesired loss spikes before the next exact LORO update.

---

> ### Author Response · Authors · 2024-11-26
>
> Dear Reviewer fYLm,
>
> We sincerely appreciate the effort you’ve dedicated to providing constructive and insightful comments. We are eager to hear your feedback and to know whether our rebuttal properly addresses your concerns.
>
> Thank you once again for your time and valuable response.
>
> Best,
>
> Authors

---

> > ### Comment · Reviewer_fYLm · 2024-12-02
> >
> > Thank you for the detailed response and updates to the paper. After reading the rebuttal and discussions from other reviewers, I'm inclined to accept this work.

---

### Meta-Review · Area_Chair_VMAP · 2024-12-18

**Metareview:**

The paper claims that pretraining of LLMs using low-rank factorization of the weights with a direct optimization on the factors can be problematic. It is shown that issues arise due to redundant information in the full gradient. An approach to solve this problem using Riemannian optimization is introduced and it is claimed that the approach works better for pretraining LLaMA models (up to 1B) than baselines such as GaLore.

The main strength of the paper, as also recognized by the reviewers, is the clean and principled formulation based on Riemannian optimization, and the fact that it works well also on larger 1B models.

Regarding weaknesses, the main one is that the claim of direct low-rank pretraining being inefficient is not fully supported by experiments, and the paper interleaves standard updates with an occasional Riemannian one intertwined. The claims could perhaps be toned down, and a full theoretical investigation is out of scope for this paper and could be a direction for future investigations.

In summary, mathematically clean and principled methods that work well in practice are of high relevance to both theory and practitioners in the ICLR community, and efficient pretraining of LLMs is a relevant direction. I recommend to accept the paper. I encourage the authors to take the feedback of the reviewers into account when preparing a final version of the paper.

**Additional Comments On Reviewer Discussion:**

The main issues raised by the reviewers were:
* reviewers questioned whether the main claim of the paper (direct optimization of the low rank factors being inefficient) is really true. This was partially addressed by theoretical and empirical studies added to the appendix.
* the work only uses Riemannian steps once in a while. This point was addressed by additional ablation studies, showing that more frequent updates are more effective and without them the approach doesn't work.
* some issues about initialization, scaling factors, hyper parameters were raised, which were addressed by the authors adding additional ablation studies to the appendix.

My acceptance decision was mostly influenced by the fact that most criticisms raised by the reviewers were addressed, and that a full investigation of the reasons behind inefficiencies in direct optimization are out of scope. Moreover, the numerical results are interesting and run on decently sized 1b models.

---

### Decision · Program_Chairs · 2025-01-22

Accept (Poster)